# Entropic repulsion of cholesterol-containing layers counteracts bioadhesion

Jens Friedrichs[1], Ralf Helbig[1], Julia Hilsenbeck[1], Prithvi Raj Pandey[2], Jens-Uwe Sommer[2,3], Lars David Renner[1], Tilo Pompe[1,4] & Carsten Werner[1,3 ✉]

Control of adhesion is a striking feature of living matter that is of particular interest regarding technological translation[1–3]. We discovered that entropic repulsion caused by interfacial orientational fluctuations of cholesterol layers restricts protein adsorption and bacterial adhesion. Moreover, we found that intrinsically adhesive wax ester layers become similarly antibioadhesive when containing small quantities (under 10 wt%) of cholesterol. Wetting, adsorption and adhesion experiments, as well as atomistic simulations, showed that repulsive characteristics depend on the specific molecular structure of cholesterol that encodes a finely balanced fluctuating reorientation at the interface of unconstrained supramolecular assemblies: layers of cholesterol analogues differing only in minute molecular variations showed markedly different interfacial mobility and no antiadhesive effects. Also, orientationally fixed cholesterol layers did not resist bioadhesion. Our insights provide a conceptually new physicochemical perspective on biointerfaces and may guide future material design in regulation of adhesion.

Life has evolved a plethora of powerful principles to control adhesion, some of which have been recapitulated in engineered materials. Prominent examples include the superhydrophobic leaves of the sacred lotus[1] and the omniphobic surfaces of *Nepenthes* pitcher plants[2]. Whereas interfacial phenomena in nature are widely studied, the physical mechanisms underlying the control of bioadhesion—the interfacial accumulation of biopolymers and cells (including bacteria)—are not yet thoroughly understood. We previously explored the omniphobic, antibioadhesive cuticula of *Collembola* (Fig. 1a) and found that it consists of nanoscopic structures with overhanging cross-sectional profiles (Fig. 1b,c) preventing wetting and bacterial colonization[3–6]. Later the lipid-rich envelope of the *Collembola* cuticula (Fig. 1c)—considered to serve as another 'line of defence' against bioadhesion—was shown to contain aliphatic hydrocarbons, in particular steroids, fatty acids and wax esters (Fig. 1d and Extended Data Fig. 1)[7]. Whereas wax esters can be reasonably assumed to support the non-wetting properties of the cuticula[8], the role of steroids and fatty acids remains elusive. Free fatty acids have been reported to kill or inhibit the growth of bacteria and fungi[9,10], and steroids have been found to reduce bioadhesion on sponges and sea stars[11]; however, no mechanistic explanation for this effect of steroids is available. Amphiphilic lipid components of the *Collembola* cuticula are also contained in the membranes of animal and bacterial cells that play key roles in compartmentalization and the functional alignment of molecular machineries[12]. Cholesterol, in particular, has been comprehensively studied and its presence is considered crucial for the regulation of functional lipid domains and the interaction between proteins and lipids[13]. However, the functional relevance of cholesterol at interfaces of living structures other than cellular membranes is underexplored.

Using lipid layers without the cuticula's morphological features, we dissect the role of molecular assemblies in the control of bioadhesion from the antiadhesive topography effects of the *Collembola* cuticula[5,14]. Cholesterol-containing lipid layers were found to counteract bioadhesion by a previously unknown entropic repulsion mechanism that is—unlike earlier reported interfacial effects[15–17]—caused by orientational fluctuations.

## Cholesterol layers exibit low bioadhesion

Lipids of the *Collembola* cuticula (Fig. 1d) were physiosorbed in multilayers on solid supports by spin-coating (spin-coated lipid multilayers, SCLs; Fig. 1e). For cholesterol SCLs, attenuated total reflection Fourier-transform infrared spectroscopy (ATR–FTIR) showed a multilayer structure with the majority of molecules oriented perpendicular to the interface (Supplementary Fig. 2)[18,19]. Self-assembled monolayers (SAMs; Fig. 1e) chemisorbed on gold served as a comparative analysis of the investigated lipidic cuticle components in geometrically constrained molecular orientations (Extended Data Figs. 1 and 2a–c).

A hallmark of bioadhesion is the formation of a molecular conditioning layer of proteins, nucleic acids, lipids and other biomolecules[20]. This process was mimicked by adsorption experiments with lysozyme and albumin, the results showing significantly lower adsorbed amounts of proteins on cholesterol SCLs than on all other studied SCLs and SAMs (Fig. 1f and Extended Data Fig. 2f,g). Likewise, adhesion of Gram-positive (*Staphylococcus epidermidis*, strain PCI 1200) and Gram-negative (*Escherichia coli*, strain W3110) bacteria was found to be very low on cholesterol SCLs (Fig. 1g and Extended Data Fig. 2d,e).

[1]Institute of Biofunctional Polymer Materials, Leibniz Institute of Polymer Research Dresden, Dresden, Germany. [2]Institute of Theory of Polymers, Leibniz Institute of Polymer Research Dresden, Dresden, Germany. [3]Cluster of Excellence Physics of Life and Center of Regenerative Therapies Dresden, Technische Universität Dresden, Dresden, Germany. [4]Institute for Biochemistry, Leipzig University, Leipzig, Germany. ✉e-mail: werner@ipfdd.de

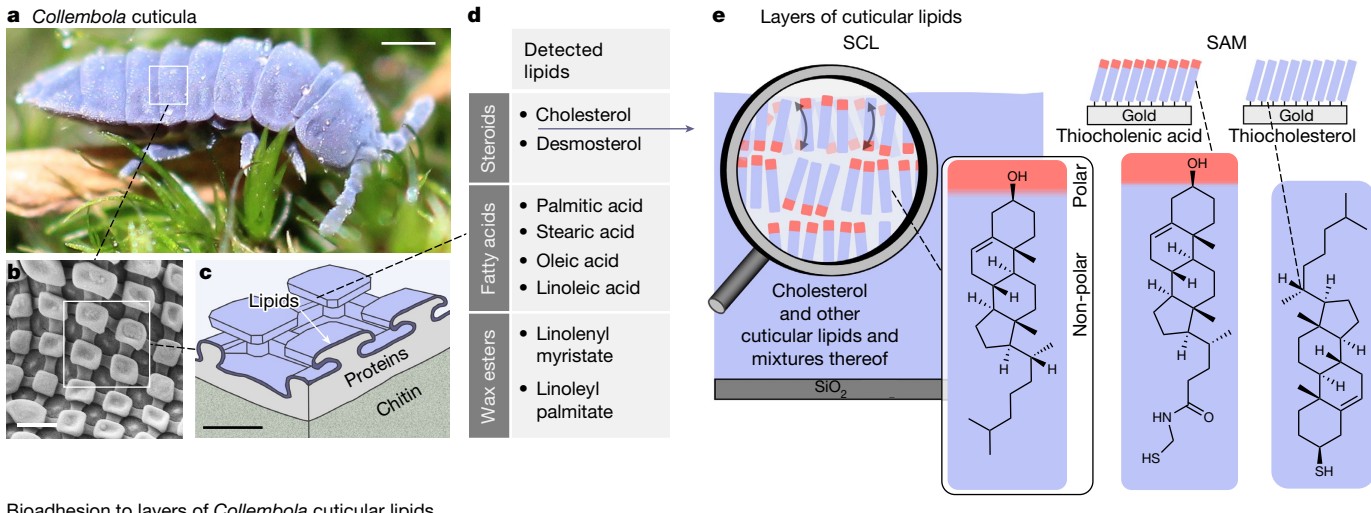

**a** *Collembola* cuticula

**b**

**c**
Lipids
Proteins
Chitin

**d**
Detected lipids

| | |
|---|---|
| Steroids | • Cholesterol<br>• Desmosterol |
| Fatty acids | • Palmitic acid<br>• Stearic acid<br>• Oleic acid<br>• Linoleic acid |
| Wax esters | • Linolenyl myristate<br>• Linoleyl palmitate |

**e** Layers of cuticular lipids

SCL          SAM

Thiocholenic acid      Thiocholesterol

Cholesterol and other cuticular lipids and mixtures thereof

SiO₂

Bioadhesion to layers of *Collembola* cuticular lipids
Monocomponent

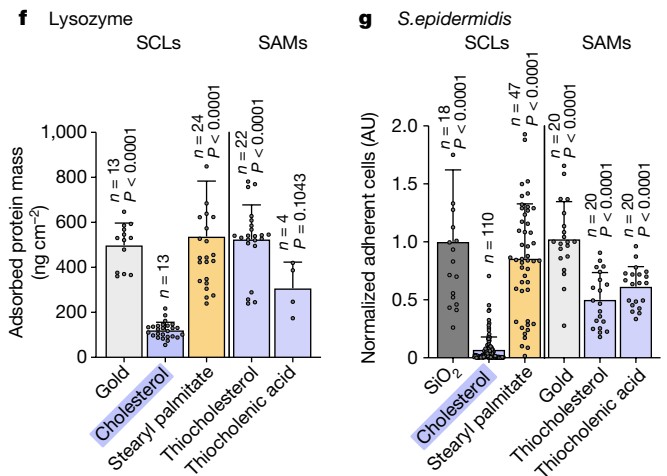

**f** Lysozyme

**g** *S. epidermidis*

Multicomponent

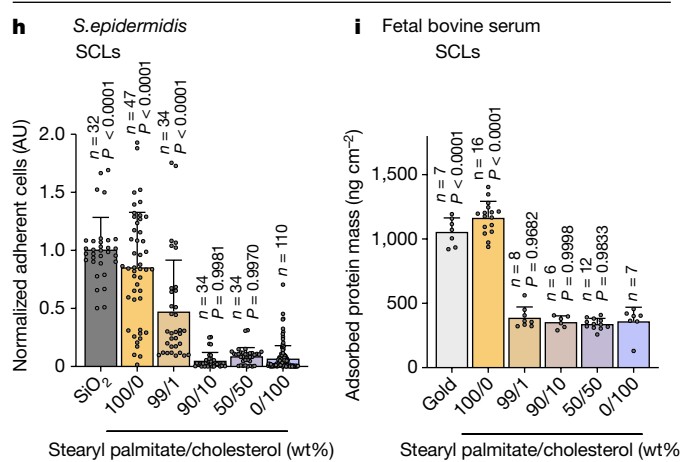

**h** *S. epidermidis*
SCLs

**i** Fetal bovine serum
SCLs

**Fig. 1 | Layers of *Collembola* cuticular lipids and their bioadhesion properties. a**, Image of *Tetrodontophora bielanensis*, an exemplary *Collembola* sp. Scale bar, 1 mm. **b**, Scanning electron microscopy image of a *T. bielanensis* cuticula. Scale bar, 500 nm. **c**, Cross-sectional schematic of the cuticula, showing a layered structure consisting of a chitin-rich inner skeleton covered by a protein-rich layer. A thin, lipid-rich envelope covers the protein-rich layer. Scale bar, 200 nm. **d**, Summary of lipids detected in the outer cuticula layer of *T. bielanensis*[7]. **e**, Layers of *Collembola* cuticular lipids; SCLs containing cholesterol facilitate orientational adaptation of the topmost lipids to the polarity of the environment. ATR–FTIR (Supplementary Fig. 2) and dynamic contact angle measurements (Fig. 2c and Extended Data Fig. 4a) indicate highly ordered cholesterol molecules, with the hydrocarbon tail of the outer cholesterol layer initially oriented towards the interface and the hydroxyl groups oriented inward. SAMs chemisorbed to gold via thiol groups, with either the polar or non-polar side of cholesterol oriented to the interface, served

as references in selected experiments. **f–i**, Adsorbed amount of protein (**f,i**) and normalized adherent cells (**g,h**). Adsorbed amount of protein (lysozyme or fetal bovine serum) on monocomponent layers of *Collembola* cuticular lipids (**f**) and multicomponent SCLs of stearyl palmitate and cholesterol (**i**), as determined by quartz crystal microbalance measurements. Normalized adherent cells of *S. epidermidis* on monocomponent layers of *Collembola* cuticular lipids (**g**) and multicomponent SCLs of stearyl palmitate and cholesterol (**h**). Data normalized to average adherent cell density on a silica (SiO₂) substrate. **h,i**, Pure stearyl palmitate SCLs (100/0) and pure cholesterol SCLs (0/100) served as negative and positive controls, respectively. **f–i**, Mean + s.d. The number of observations (*n*) is indicated. *P* values (comparison with cholesterol SCL condition in **f,g** and with the 0/100 condition in **h,i**) were determined using one-way analysis of variance. AU, arbitrary units.

By analysis of cholesterol SCLs of varying layer thickness (Supplementary Fig. 3a), any influence of multilayer thickness on antiadhesive properties could be excluded. Notably, SAMs of thiocholesterol and *N*-(2-mercaptoethyl)-3-hydroxy-5-cholenic acid amide (thiocholenic acid)—that is, surrogates of chemically fixed cholesterol monolayers of opposing orientational molecular alignment; Fig. 1e)—accumulated higher amounts of adsorbed protein and adherent bacteria than cholesterol SCLs (Fig. 1f,g and Extended Data Fig. 2d–g). This observation suggests that restriction of molecular mobility in layered cholesterol impedes antiadhesive properties. Partial dissolution of cholesterol SCLs was excluded as a cause of low bioadhesion by total organic carbon content analysis (Supplementary Fig. 4).

Because cholesterol occurs in the *Collembola* cuticula together with several other lipid components (Fig. 1d), we further investigated its relevance regarding the adhesive characteristics of multicomponent SCLs. Multicomponent SCLs containing stearyl palmitate—a lipid that causes strong bioadhesion to its single-component SCLs (Fig. 1f,g and Extended Data Fig. 2d,g)—and cholesterol, with cholesterol content ranging from 0 to 100 wt%, were investigated using similar bioadhesion assays (Extended Data Fig. 3a–c). Multicomponent SCLs containing at least 10 wt% cholesterol showed low adherent cell densities similar to pure cholesterol SCLs (Fig. 1h and Extended Data Fig. 3d). Likewise, protein adsorption on multicomponent SCLs was significantly reduced when a minimum of only 1 wt% cholesterol was present (Extended Data

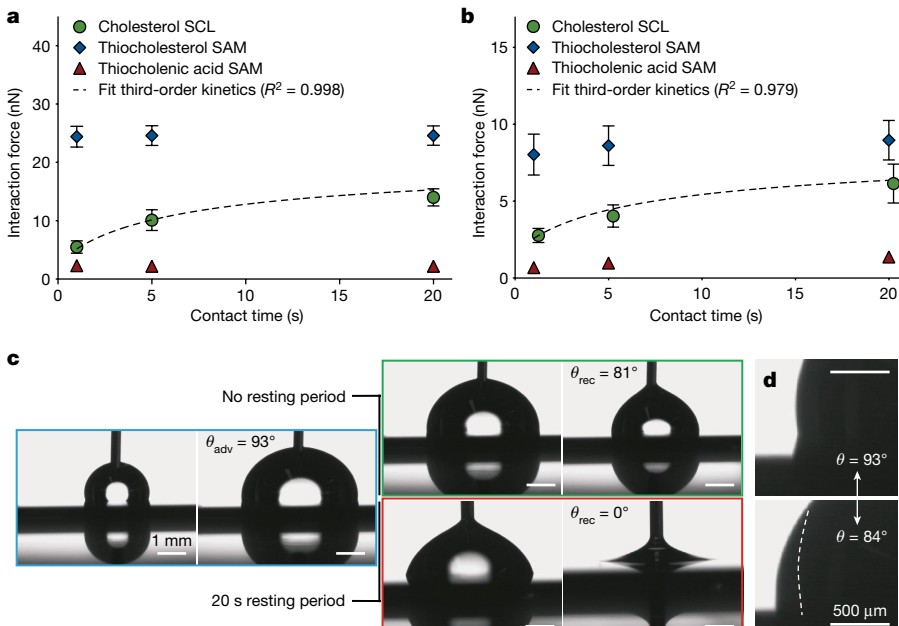

**Fig. 2 | Cholesterol SCLs dynamically adapt to the polarity of their environment. a,b**, Contact time-dependent interaction forces (determined by AFM-based force spectroscopy) between a hydrophobic colloidal probe (**a**) (∅10 μm silica bead modified with a hydrophobic silane) or individual *E. coli* cells (**b**) (attached to a ∅10 μm silica bead) and cholesterol SCLs or thiocholesterol and thiocholenic acid SAMs. Experiments were performed at 37 °C. **a,b**, Mean ± s.e.m. Data were obtained from at least three independent experiments. Regressions are the best-fit solutions for a third-order process indicating formation of bonds at the probe cell–SCL interface following $\frac{d\sigma}{dt} = -k\sigma^3$ (for details see Supplementary Note 1). The interaction force was assumed to be directly correlated to the surface concentration of formed bonds, $\sigma$. Coefficients of determination ($R^2$) for fits are shown. **c**, Representative images showing the dynamic contact angles of water on the surface of a cholesterol SCL. The high advancing contact angle ($\theta_{adv}$, blue box) reflects a hydrophobic surface. Following immediate withdrawal of the droplet (no resting period, green box), a slightly reduced but still high receding contact angle ($\theta_{rec}$) was observed. No receding of the three-phase contact line was observed when the water droplet was withdrawn after a 20 s resting period (red box). **d**, During the 20 s resting period, oscillations in the three-phase contact line between the initial advancing angle (93°) and an approximately 10° lower angle were observed.

Fig. 3e,f). The results of single-protein adsorption experiments were confirmed when applying physiologically relevant protein mixtures (that is, 10 vol% fetal bovine serum; Fig. 1i).

## Cholesterol layers adapt to ambient polarity

The results of the bioadhesion assays suggest that mobility of interfacial cholesterol is key to the antiadhesive characteristics of single and multicomponent SCLs. Atomic force microscopy (AFM)-based colloid probe force spectroscopy, AFM-based single-cell force spectroscopy and dynamic contact angle measurements provided further insight into the underlying interfacial dynamics.

AFM-based force spectroscopy (Supplementary Fig. 5a,b and Methods) showed the dynamic interfacial adaptation of immersed cholesterol SCLs. Interaction forces between hydrophobic colloidal probes (Supplementary Fig. 5c) or single *E. coli* cells and cholesterol SCLs were quantified (Supplementary Fig. 5d) and were found to increase continuously with contact time (Fig. 2a,b). Further analysis showed that third-order kinetics controlled the underlying polarity adaptation process of the SCL interface to maximize hydrophobic interactions (Fig. 2a,b and Supplementary Note 1). Control experiments with thiocholesterol and thiocholenic acid SAMs confirmed that cholesterol mobility within SCLs is key to the observed adaptation (Fig. 2a,b).

Dynamic wetting experiments on cholesterol SCLs in air showed high advancing and receding water contact angles when the droplet was immediately withdrawn after deposition (Fig. 2c, Extended Data Fig. 4a and Supplementary Video 1), indicating that the hydrocarbon chains of the outer cholesterol layers conditioned in air are preferentially oriented towards the interface. If the water droplet was kept on the surface for 20 s before receding, oscillations of the three-phase contact line occurred for 15–20 s and subsequently the static contact angle was lower by approximately 10° (Fig. 2d and Supplementary Video 2). Pinning of the three-phase contact line was observed when the droplet was withdrawn (Fig. 2c, Extended Data Fig. 4a and Supplementary Video 2). The contact angle drop was found to be fully reversible following drying in inert gas (Extended Data Fig. 4d). The resting-period-dependent decrease in the contact angle, the oscillating three-phase contact line during the resting period and the increased contact angle hysteresis confirm a dynamic polarity adaptation of cholesterol at the interface of the SCLs in response to changes in the polarity of the environment.

The time dependence of the increase in interaction force between cholesterol SCLs and AFM probes (Fig. 2a) and single bacterial cells (Fig. 2b), as well as the decrease in receding water contact angle on cholesterol SCLs (Fig. 2c), indicate that the interfacial polarity adaptation process of cholesterol SCLs is relatively slow (that is, occurring over a period of a few seconds).

Also underpinning the results of the bioadhesion assays (Fig. 1h,i and Extended Data Fig. 3d–f), low quantities of cholesterol in multicomponent SCLs of cholesterol and stearyl palmitate sufficed to decrease the receding water contact angle (Extended Data Fig. 4c) and to increase the interaction force with hydrophobic AFM probes (Extended Data Fig. 3h).

## Entropic repulsion by surface fluctuations

The antibioadhesive properties of cholesterol-containing SCLs were shown above to correlate with a dynamic adaptation of the interfacial orientation of cholesterol in response to changes in environmental polarity. We hypothesized that entropically driven orientational fluctuations of interfacial cholesterol molecules mechanistically connect

these features: any adsorption of biomolecules or attachment of (bacterial) cells requires an orientational (polarity) adaptation of the SCL interface that constrains the orientational states of interfacial cholesterol and thereby reduces the entropy of the system.

To validate this hypothesis we first examined the system for the characteristic temperature dependence of entropic effects on protein adsorption (according to the definition of the Gibbs free energy of adsorption, $\Delta G = \Delta H - T\Delta S$) using three selected proteins of increasing size and complexity: lysozyme, albumin and fibrinogen. Indeed, protein adsorption to cholesterol SCLs was observed to decrease when the temperature was increased from 15 to 40 °C (that is, in a temperature range in which significant temperature-induced changes in protein conformation can be excluded[21–23]), in contrast to minor changes in similar experiments with thiocholesterol and thiocholenic acid SAMs (orientationally fixed controls of cholesterol SCLs underneath the adsorbed protein) (Fig. 3a and Extended Data Fig. 5a,b). At thermal adsorption equilibrium, the temperature dependence of the Gibbs free energy of adsorption can be used to quantitatively estimate the repulsive entropic barrier ($\Delta S_{chol}$) to protein adsorption on cholesterol SCLs. A quantitative analysis of adsorption (based on the slopes of $\Delta G(T)$ on cholesterol SCLs and the SAM controls) was performed for lysozyme, because adsorption-induced structural changes are negligible for the small, compact protein and provided $\Delta S_{chol} = -200 \pm 60 \, \text{J} \text{mol}^{-1} \text{K}^{-1}$ (Extended Data Fig. 5c and Supplementary Note 2)[14,24]. This value of $\Delta S_{chol}$ shows a significant entropic repulsion counteracting bioadhesion to cholesterol SCLs. The plot of $\Delta G(T)$ for cholesterol SCLs exhibits a positive slope (that is, a negative $\Delta S$ of adsorption), indicating that entropic repulsion overcompensates any entropic gain of protein adsorption for that system. Whereas desorption-determined processes might show a similar temperature dependence, protein adsorption is well known to be strongly determined by adsorption[24].

Entropic repulsion effects were previously reported to occur either due to the pinned conformational flexibility of grafted oligo(ethylene glycol)[15] and poly(ethylene glycol) brushes[16] or enforced smoothening of height fluctuations of lipid bilayers[17]. Herein, entropic repulsion of cholesterol SCLs is suggested to be mechanistically very different, primarily governed by polarity-controlled orientational fluctuations: The entropic penalty for the suppression of orientational fluctuations of cholesterol molecules at the SCL interface on protein adsorption can be estimated by relating the cross-sectional area of protein-facing cholesterol at the SCL interface to the area of orientationally free cholesterol at the SCL–solution interface (Fig. 3b and Supplementary Note 3)[18,19]. The obtained value for this entropic penalty ($-160 \, \text{J} \text{mol}^{-1} \text{K}^{-1}$) is close to the experimentally determined entropic repulsion barrier derived from temperature-dependent protein adsorption experiments ($-200 \, \text{J} \text{mol}^{-1} \text{K}^{-1}$). This rather large entropy penalty per adsorbing lysozyme molecule, corresponding to about three hydrogen bonds, emphasizes the relevance of the uncovered repulsive effect. In this estimation of entropic penalty, three associated interfacial cholesterol molecules were assumed to be cooperatively involved in orientational fluctuations, which was reasoned at first by geometrical considerations of cholesterol molecular packing in SCLs. Time-dependent interaction force measurements (Fig. 2a,b) showing a third-order kinetics support this assumption because the data indicate cooperative interactions involving three cholesterol molecules (Fig. 3b).

Analysis of adsorption experiments with the larger proteins albumin and fibrinogen yielded lower values for the estimated repulsive entropic barrier (probably due to entropy effects of conformational changes; Supplementary Note 2). However, it similarly showed a negative (repulsive) entropy contribution of cholesterol SCLs indicating the general relevance of the newly identified antiadhesion mechanism.

In an entirely independent approach, cholesterol multilayers in contact with water were explored by molecular dynamics (MD) simulations and similarly showed the relevance of molecular orientational fluctuations (Supplementary Note 4). Using equilibrium multilayer systems and steered MD simulations with reverted orientation of a variable percentage of interfacial cholesterol molecules, we observed spontaneous reorientation of these molecules within 1 µs simulation time such that their hydroxyl groups aligned again toward the water layer (Fig. 3c,d and Extended Data Fig. 6b,d). Importantly, control simulations using the cholesterol analogue stigmasterol (Fig. 4) did not show this spontaneous molecular reorientation (Fig. 3c,d and Extended Data Fig. 6b,d). The orientation correlation function of cholesterol molecules in the interfacial layer indicated a rapid decay over a distance of 1 nm (Fig. 3e), confirming orientational cooperativity of a few (two or three) cholesterol molecules as discussed above. Together with the observation of high vertical fluctuation of the interfacial layer of cholesterol multilayers (Fig. 3f), the MD simulations support the view that cholesterol SCLs represent a highly dynamic and fluctuating molecular interface.

To obtain direct experimental evidence of orientational fluctuations of cholesterol at SCL interfaces, we applied AFM-based force mapping to detect the simultaneous exposure of polar and non-polar residues of immersed cholesterol SCLs (Extended Data Fig. 7). The low and high levels of interaction forces between a hydrophobic AFM tip and thiocholenic acid or thiocholesterol SAMs span the range of interaction forces detected for cholesterol SCLs, confirming that both polar and non-polar residues are exposed at the interface of immersed cholesterol SCLs (Extended Data Fig. 7c). Repeated force–distance measurements at a constant position showed substantially higher variations for cholesterol SCLs than for thiocholesterol SAMs, providing proof of orientational fluctuations of cholesterol at the SCL interface (Fig. 3g and Extended Data Fig. 8).

## Bioadhesion to layers of cholesterol analogues

The discovered entropic barrier to bioadhesion resulting from orientational fluctuations at the interface of cholesterol-containing SCLs raises the question of what key structural features of cholesterol underlie this effect.

Therefore, we compared the bioadhesion characteristics of SCLs prepared from structurally related cholesterol analogues (Fig. 4 and Extended Data Fig. 9) in terms of bacterial adhesion and protein adsorption (Extended Data Fig. 10). The results confirmed that molecular amphiphilicity is essential—but not sufficient—for low bioadhesion (Fig. 4 shows the results of different assays aggregated into a bioadhesion index) and showed that even small deviations in the molecular structure of cholesterol can strongly diminish its antiadhesive performance. Only SCLs of dehydrocholesterol, the closest chemical relative of the tested cholesterol analogues, approached the bioadhesion index of cholesterol SCLs.

Our results show that cholesterol assembles in molecular arrangements that are capable of entropically restricting bioadhesion. The combination of the spatially decoupled amphiphilicity of cholesterol and its effective alignment in multilayers, generating a slowly adaptive, cooperative interfacial orientational mobility of the assemblies, was identified to be the prerequisite of pronounced entropic repulsion.

The resting-time-dependent decrease in the receding water contact angle indicates that, among the set of cholesterol analogues compared, only cholesterol and dehydrocholesterol SCLs meet these criteria (Extended Data Fig. 9d). Fluorescence recovery after photobleaching (FRAP) measurements on cholesterol SCLs providing lateral diffusion coefficients of $0.3–0.4 \, \mu\text{m}^2 \, \text{s}^{-1}$ further support the ordered, but slowly adaptive, characteristics of the system, comparable to smectic liquid crystals and high-cholesterol-content lipid membranes (Supplementary Fig. 6)[25,26].

In agreement with the experimental results, MD simulations show that SCLs of analogues with slight structural modifications with respect

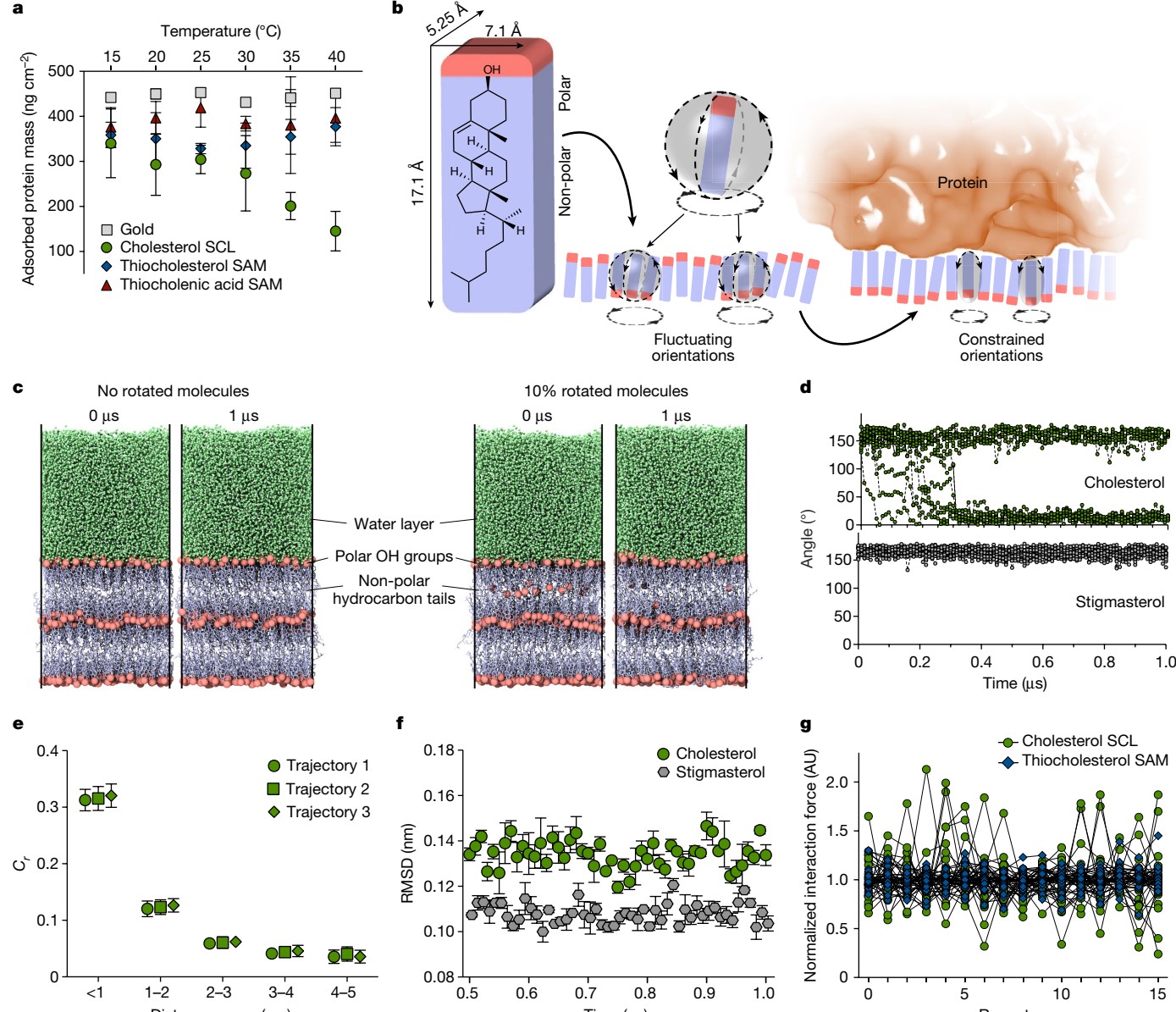

**Fig. 3 | Quantitative evidence of entropic repulsion by cholesterol SCLs and verification of the molecular mechanism. a**, Temperature-dependent adsorbed amounts of lysozyme as determined by quartz crystal microbalance measurements. For cholesterol SCLs, a pronounced temperature dependence on lysozyme adsorption was observed. Data were obtained from at least three independent experiments. Mean ± s.d. **b**, Schematic illustration of the transition of interfacial cholesterol from the orientationally fluctuating (unbound) to orientationally constrained (protein-bound) state. Three aligned cholesterol molecules are shown to highlight the cooperative reorientation of neighbouring molecules within SCLs derived from geometric considerations and evidenced by the experimentally observed third-order kinetics of interfacial adaptation and orientational correlation length in MD simulations (Supplementary Notes 1 and 4). The entropic penalty of bioadhesion can be estimated from the ratio of constraint and fluctuating interfacial areas, $\Delta S = R \times \ln(A_{constr}/A_{fluct})$, of cholesterol molecules, with $A_{fluct}$ roughly equal to the molecular dimension of cholesterol due to the cooperative orientational fluctuation and $A_{constr}$ related to the cross-sectional area of the cholesterol facing the interface in the protein-bound state (Supplementary Note 3). **c**, MD simulations of cholesterol multilayers in contact with water, with equilibrium state (left) and intentionally reverted molecular orientation of 10% of molecules in interface layer (right) (for details see Supplementary Note 4). Oxygen atoms

of hydroxyl groups are shown as orange spheres. **d**, Angles between the vector connecting atoms C3 and C17 on cholesterol and stigmasterol (Extended Data Fig. 6c and Supplementary Note 4) and the $z$ axis for interfacial molecules as a function of simulation time for one representative simulation trajectory. In cholesterol multilayer systems with 10% of reverted molecules, spontaneous back-reorientation within 1 μs of simulation time was observed. No back-orientation was observed for stigmasterol controls. **e**, Correlation function ($C_r$) as a function of the distance to a reference molecule, plotted for three simulation trajectories of cholesterol multilayer systems (with no changed orientation of molecules at the interface). Error bars represent s.d. at that point over time intervals. **f**, Root mean squared deviation (RMSD) (for cholesterol multilayers and stigmasterol controls) over three simulation trajectories of the $z$-component of the position of all O atoms at the interface layer as a function of time for frames every 0.01 μs, with error bars denoting s.e.m. **g**, Normalized interaction forces between hydrophobic AFM tips and cholesterol SCL or thiocholesterol SAM surfaces. Each curve contains 16 data points recorded sequentially at the same location on the substrate. Individual datasets were acquired at different locations on the substrate. Curves are normalized to the average of each dataset. Data were obtained from at least three independent experiments.

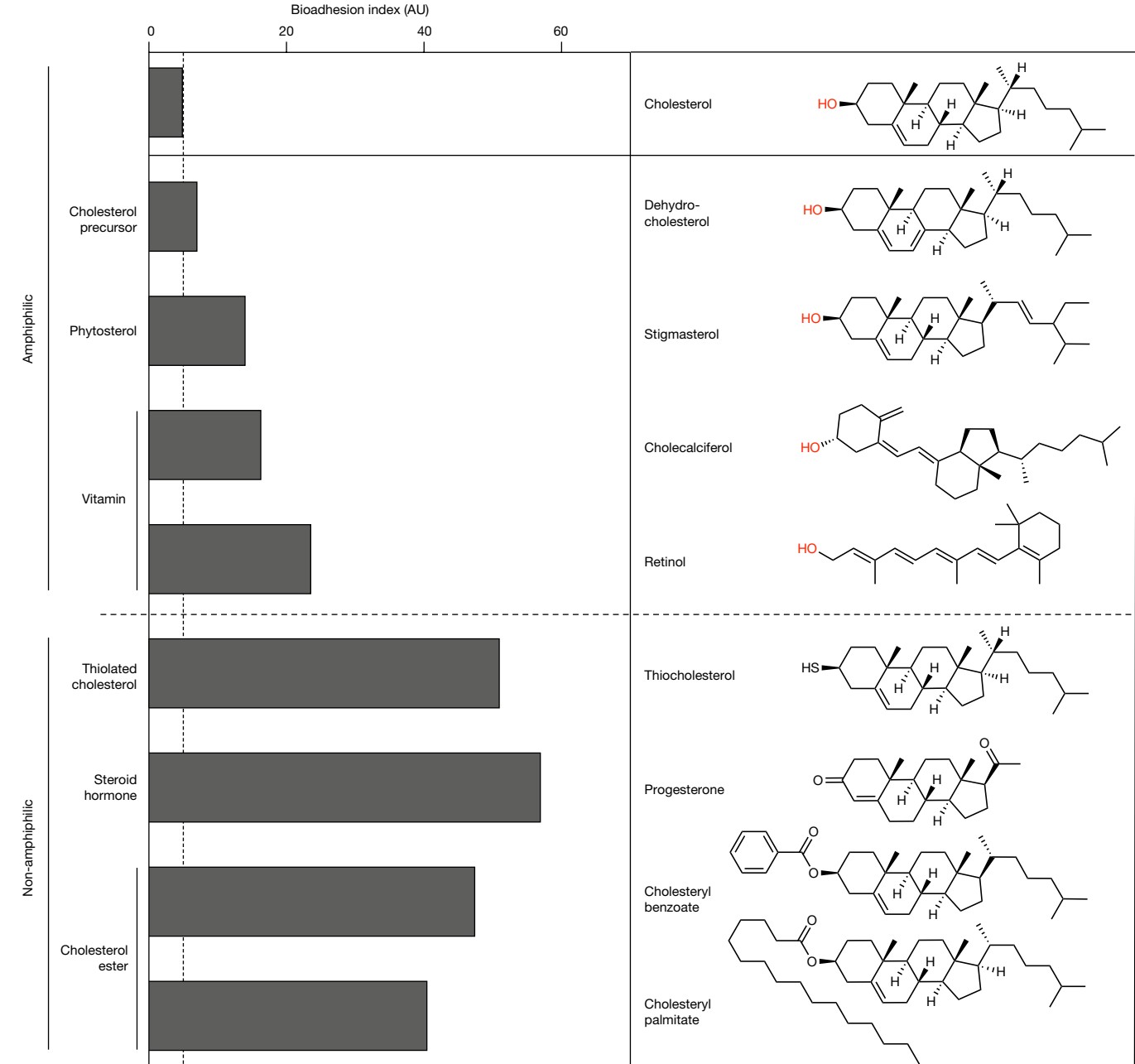

**Fig. 4 | SCLs of cholesterol analogue molecules show graded bioadhesion.** Bioadhesion to SCLs of cholesterol analogues (polar hydroxyl groups shown in red). To facilitate the comparison, a bioadhesion index was introduced by aggregating the results of bacterial adhesion experiments with *E. coli* and *S. epidermidis* cells and protein adsorption experiments with bovine serum albumin, lysozyme and fetal bovine serum (for the full set of experimental results see Extended Data Fig. 10), all normalized to the corresponding results for cholesterol SCLs. The reference value was calculated using the respective controls from bacterial adhesion ($SiO_2$) and protein adsorption tests (gold). AU, arbitrary units.

to cholesterol (Extended Data Fig. 9d) behave significantly differently, with no spontaneous back-rotations of reverted molecules within the simulation time (Fig. 3d and Extended Data Fig. 6b,d). The simulations also indicate that stigmasterol SCLs are subject to significantly lower interfacial fluctuations than cholesterol SCLs, suggesting that the specific intermolecular interactions of cholesterol provide high interfacial mobility and allow for spontaneous molecular fluctuations (Fig. 3f and Extended Data Fig. 6f). These findings strongly support the experimentally deduced criteria for amphiphilic molecules capable of generating entropically repulsive assemblies.

Exploring the compositional line of defence of the antiadhesive *Collembola* cuticula, we identified physicochemical features of cholesterol-containing layers that result in effective entropic repulsion. Given the widespread occurrence of cholesterol at tissue boundaries, our findings may shed new light on important and ubiquitous interfacial processes of living matter. The development of cholesterol-inspired synthetic and scalable materials to control bioadhesion is highly attractive but demands further analyses of the molecular requirements that govern this newly identified mechanism.

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

## Methods

### Materials

All chemicals (solvents, lipids, media, proteins and so on) used in this article were purchased from Sigma-Aldrich without further purification, except where stated otherwise.

### Lipid layer preparation

**SCLs.** SCLs were prepared on silicon wafers ($10 \times 15$ mm$^2$). The substrates were cleaned by immersion in a solution of deionized water, ammonia and hydrogen peroxide (volume fraction 5/1/1) for 15 min at 70 °C, rinsed repeatedly in Milli-Q water and then dried in a nitrogen stream. The cleaned substrates were immediately used for the preparation of SCLs by spin-coating. Compounds were dissolved in chloroform at a concentration of 2 wt% (unless stated otherwise), and spin-coating (LabSpin6, SÜSS MicroTec) was performed at 3,000 rpm s$^{-1}$ for 30 s.

**SAMs.** Silicon wafers were cleaned as described for SCL preparation. Clean substrates were first coated with a 3 nm chromium adhesion layer and then a 50 nm gold layer by chemical vapour deposition performed at $5 \times 10^{-5}$ mbar (Univex 300, Leybold). Gold substrates were cleaned by immersion in a solution of deionized water, ammonia and hydrogen peroxide (volume fraction 5/1/1) for 15 min at 70 °C, rinsed repeatedly in Milli-Q water and then dried in a nitrogen stream. To minimize defects, the gold substrates were used for SAM formation immediately after cleaning. Thiol compounds were dissolved in ethanol (analytically pure, p.a.) to a concentration of 1 mM. Before use, thiol solutions were sonicated for 5–10 min. Gold-coated substrates were additionally cleaned in ethanol (p.a.) for 30 min using an ultrasonic bath. Subsequently, samples were immersed in thiol solutions and incubated for 24 h to ensure complete assembly. Containers housing the solution and samples were filled with dry nitrogen and sealed to minimize oxygen exposure. After incubation, sample surfaces were rinsed for 15–20 s with ethanol (p.a.), dried under nitrogen and used directly for further experiments.

### Preparation of thiocholenic acid

For the reaction scheme see Supplementary Fig. 1. One equivalent of 3-acetoxy-5-cholenic acid (**A**) was dissolved in anhydrous dimethyl sulfoxide (DMSO) under an argon atmosphere. The solution was stirred on ice, and six equivalents of carbonyldiimidazole (**B**), dissolved in anhydrous DMSO, were added slowly to the solution. The reaction mixture was stored at 4 °C overnight for activation. A solution of 2.5 eq. cystamine (**D**) was added dropwise to the stirred, ice-cooled imidazole intermediate (**C**) and left to react overnight at room temperature. The product (**E**) was purified, freeze-dried, dissolved in ethanol and diluted in water before the addition of 1 M NaOH to obtain ester cleavage. After purification and lyophilization, the product (**F**) was again dissolved in ethanol and diluted in water. The disulfide bond was cleaved by the addition of a sixfold excess of tris(2-carboxyethyl)phosphine hydrochloride to cholenic acid in aqueous solution at neutral pH. The cleavage product (**G**) was purified and lyophilized (purity around 80%). For reaction/purity control and purification, reverse-phase high-performance liquid chromatography (RP–HPLC) with a linear gradient of acetonitrile in water with additive formic or trifluoroacetic acid was used. The analytical HPLC instrument (1260 Infinity II, Agilent) was equipped with a diode array detector (210 and 278 nm) and an electron spray ionization–time-of-flight (ESI–TOF) detector. The preparative HPLC instrument (1200 series, Agilent) used a diode array detector and a fraction collector in manual collection mode.

### TOC measurements

The total organic carbon (TOC) content of solutions was analysed using a Sievers 5310C laboratory TOC analyser (GE Analytical Instruments) in accordance with the manufacturer's specifications. Information on the preparation of samples is provided in Supplementary Fig. 4.

### Bacterial adhesion assays

*S. epidermidis* (strain PCI 1200, ATCC) and *E. coli* (strain W3110) were grown overnight from single colonies in lysogeny broth (LB) at 37 °C and 200 rpm. Overnight cultures were centrifuged at 4,000$g$ for 5 min. The supernatant was removed and the remaining pellet resuspended in LB. This washing step was repeated three times. Cell densities were adjusted to an optical density (OD$_{600}$) of 0.2 in fresh LB, and sample substrates were incubated in the bacterial solution for 1 h at 37 °C (no shaking). After incubation, adherent bacteria were fixed with 4% paraformaldehyde in PBS for 10 min, washed in fresh PBS and Milli-Q water and dried under nitrogen. Samples were sputter-coated with a 15 nm gold layer (SCD 050, Balzers) and imaged by scanning electron microscopy (SEM; XL30 ESEM-FEG, Philips/FEI) in high-vacuum mode at an acceleration voltage of 5 kV. For each sample, at least six images were acquired at random positions and cells were counted using the counting tool in Fiji[27]. The scale bars of SEM images were used to calibrate pixel width. The Fiji 'cell counter' plugin was used to count the number of cells in the calculated area (approximately $10^3$ µm$^2$). The numbers counted were either normalized against the median value of the silicon reference for relative comparisons or scaled up to cells per square millimetre for absolute values.

### QCM measurements

Quartz crystal microbalance (QCM) measurements were performed using a QCM-D model E4 (Biolin Scientific) equipped with a peristaltic pump system (IPC, Ismatec). Gold-coated quartz crystals (QSX301, Quantum Design) with a resonance frequency of 5 MHz were used for QCM measurements. SCLs and SAMs were prepared on QCM crystals as described above. All measurements were performed at a flow rate of 100 µl min$^{-1}$. Protein solutions (lysozyme, bovine serum albumin and fibrinogen (100 µg protein ml$^{-1}$ PBS)) or 10 vol% fetal bovine serum (Merck) in PBS were adsorbed on the layered samples for 1 h and subsequently subjected to a desorption regime for 30 min with PBS. Frequency and dissipation shifts induced by the adsorbed proteins were recorded in real time at the third, fifth, seventh, ninth, 11th and 13th overtones (15, 25, 35, 45, 55 and 65 MHz, respectively). The mass of adsorbed protein was calculated using the Sauerbrey equation[28] with Q-Sense DFind software (Biolin Scientific).

### Dynamic contact angle measurements

Contact angle measurements were performed using an OCA 30 optical contact angle measuring and contour analysis system equipped with a TPC 160 temperature-controlled chamber (DataPhysics Instruments). Droplets of degassed deionized water were dispensed and redispensed at varying velocities of 0.3–2.0 µl s$^{-1}$ to monitor advancing and receding contact angles and their time-dependent behaviour.

### Ellipsometry

Ellipsometry measurements were performed with an M-2000 ellipsometer (J. A. Woollam) equipped with a 50 W QTH lamp operating at wavelengths from 371 to 1,000 nm and an angle of incidence of 75°. The oxide layer thickness of the silicon wafer was determined by ellipsometry before layer assembly. The thickness of the assembled layers was calculated by an optical model that included three layers: Si, SiO$_2$ and a Cauchy layer.

### In situ ATR–FTIR

For in situ ATR–FTIR, 500 µl of a 2% cholesterol-chloroform solution was spin-coated on a germanium ATR crystal. Characterization of the deposited cholesterol SCLs by in situ ATR–FTIR was performed as described previously[29]. In situ ATR–FTIR spectroscopy was conducted using the single-beam sample reference technique to obtain

compensated ATR–FTIR spectra in dry and aqueous environments[30,31]. Dichroic measurements were performed according to a previously described method[32]. Infrared light was polarized by a wire grid polarizer (SPECAC). The ATR–FTIR attachment was operated on an IFS 55 Equinox spectrometer (BRUKER Optics) equipped with a Globar source and mercury-cadmium-telluride detector. P- and s-polarized spectra were recorded from dry cholesterol SCLs. The high dichroic ratios of the n(C-O) band of the C-O-H headgroup of cholesterol can be verified in the line of the ATR–FTIR dichroism measurements of lipid bilayers. The dichroic ratio $R = A_P/A_S$, with absorbance $A_p$ measured in p-polarization and $A_S$ measured in s-polarization, of infrared bands with a transition dipole moment ($M$) located perpendicular to the surface plane also showed high values, with $R > 4$. Briefly, under conditions of ATR at the interface of the dense medium (Si) and rare medium (air), an evanescent wave was established with an electrical field split into the three electrical field components, $E_x$, $E_y$ and $E_z$, which interact with, for example, adjacent organic layers. Parallel polarized infrared light ($E_P$) forms $E_x$ and $E_z$ whereas vertically polarized light ($E_S$) forms $E_y$. High values of either $R$ or $A_p$ are obtained when the $M$ of a functional group within the organic layer lies parallel to $E_z$ (out of plane), whereas low $R$ or high $A_S$ values are obtained when $M$ lies parallel to $E_y$ (in plane), which is due to the scalar product $A = E \times M = E \times M \times \cos(E, M)$ of the vectors **E** and **M**.

## Time-of-flight secondary ion mass spectrometry
Time-of-flight secondary ion mass spectrometry (ToF–SIMS) was conducted with a ToF–SIMS 5-100 instrument equipped with a 30 kV Bi liquid metal ion gun (IONTOF). Data were acquired in $Bi_3^{++}$ mode and calibrated against a list of reference peaks (SurfaceLab7, IONTOF). The area of analysis was $300 \times 300$ $\mu m^2$, which was scanned over $128 \times 128$ pixels. The sampling depth of this technique is as low as a few nanometres—that is, only the uppermost molecular layers of the sample contribute to the analysis. Characteristic signals of cholesterol (mass to charge ratio, $m/z = 369.3$) and stearyl palmitate ($m/z = 257.2$) were selected for semiquantitative characterization of SCLs.

## FRAP
We applied a previously described FRAP protocol to analyse diffusion coefficients and mobile fractions with a fluorescence confocal laser scanning microscope using the FRAP tool (SP5, Leica)[33,34]. For FRAP measurements, fluorescent cholesterol SCLs were prepared by the addition of 1/100 or 1/20 NBD cholesterol (ThermoFisher) to pure cholesterol solutions (2 wt%) and spin-coating to clean no. 1.5 glass coverslips (Corning). Cholesterol SCLs were subsequently submerged in deionized water or PBS, and FRAP was performed by photobleaching a defined spot with a diameter of $10 \pm 1$ $\mu m$ using the following protocol: ten images before bleaching followed by a high-power laser beam with subsequent bleaching for 4 s to achieve a completely bleached area. Recovery was recorded with a 40×/1.4 numerical aperture oil immersion objective at an image acquisition speed of 1 s per frame at $256 \times 256$ pixels, for a total of 300 s (SP5, Leica). The resulting time-lapse was analysed using the MATLAB (MathWorks) programme *frap_analysis*[35].

## AFM
All AFM measurements were performed with a NanoWizard IV AFM (JPK Instruments). The cantilevers used were calibrated before measurements. Measurements were conducted in PBS at room temperature (25 °C), except where stated otherwise.

**Topographic imaging.** Surface topography of the layered surfaces was recorded using the Quantitative Imaging mode of the AFM instrument using qp-BioAC cantilevers (Nanosensors). The acquisition parameters employed were as follows: 300 nm ramp, 10 ms pixel time and force trigger of 100 pN. Images of $30 \times 30$ $\mu m^2$ with a resolution of $256 \times 256$ pixels were recorded. The data-processing software provided by the

AFM manufacturer (JPK Instruments) was used to extract the surface roughness ($R_a$) from topography images.

**Colloidal probe force spectroscopy.** For colloidal probe measurements, individual silica beads (Kisker Biotech, Ø10 $\mu m$) were attached to a tipless cantilever (PNP-TR-TL-Au, Nanoworld, nominal force constant 0.08 N m$^{-1}$) as described previously[36]. Colloidal probe-modified AFM cantilevers were cleaned in isopropanol, and adsorbed water was removed by heating at 120 °C for 10 min. The colloidal probe was hydrophobized by incubation in hexamethyldisilazane vapour for 12 h and subsequent heating at 120 °C for 1 h. The force spectroscopy parameters employed were as follows: 3 nN setpoint force, 5 $\mu m$ s$^{-1}$ approach/retract velocity and 5 $\mu m$ pulling distance. Interaction forces were extracted from the retraction force–distance curves using data from the processing software provided by the AFM manufacturer.

**Single-cell force spectroscopy.** The colloidal probe-modified AFM cantilever (see section 'Colloidal probe force spectroscopy') was made cell adhesive by application of a polydopamine coating, and individual *E. coli* cells with cytoplasmic green fluorescent protein (strain MG1655 eGFP) were attached as described previously[37]. Measurements were conducted at 37 °C using a PetriDishHeater (JPK Instruments). The same force spectroscopy parameters and data-processing routines were used for colloidal probe spectroscopy measurements. Only datasets in which the position and orientation of the bacterial cell on the AFM cantilever was unchanged before and after measurements (that is, the contact conditions/geometry were constant during the measurement) were analysed.

**Force spectroscopy measurements to quantify the spatial heterogeneity of cholesterol SCLs.** Measurements were performed using the Quantitative Imaging mode of the AFM instrument using qp-BioAC cantilevers (CB-2, Nanosensors). The cantilevers were either hydrophobized by incubation in hexamethyldisilazane vapour as described above or hydrophilized by plasma cleaning (Harrick Plasma) for 10 min. The acquisition parameters used were as follows: 100 nm ramp, 20 ms pixel time and force trigger of 500 pN. Images of $5 \times 5$ $\mu m^2$ with a resolution of $50 \times 50$ pixels were recorded at different locations on the sample surface. Interaction forces were extracted from retraction force–distance curves using the data-processing software provided by the AFM manufacturer.

**Force spectroscopy measurements to quantify the temporal heterogeneity of cholesterol SCLs.** Time-dependent force spectroscopy measurements were performed with hydrophobized (see previous section) qp-BioAC cantilevers (CB-2, Nanosensors). The force spectroscopy parameters employed were as follows: 1 nN setpoint force, 1 $\mu m$ s$^{-1}$ approach/retract velocity and 200 nm pulling distance. A waiting period of 20 s was maintained between the 16 consecutive measurements made at a single location on the sample surface, to minimize the influence of measurements on the dynamics of cholesterol molecules. Measurements were repeated at different locations of the sample. Interaction forces were extracted from retraction force–distance curves using the data-processing software provided by the AFM manufacturer.

## MD simulations
Cholesterol or stigmasterol SCLs with four molecular layers—that is, two double layers (Extended Data Fig. 6a)—were modelled. At the interface of two double layers the hydrophilic hydroxyl groups of cholesterol or stigmasterol face each other; at the interface, hydroxyl groups face the water layer. The double layer–water interface normal was along the z axis in all simulations. To model a solid substrate at the bottom of the SCL, the motion of molecules was restrained in the xy plane for the lowermost lipid molecules. The simulation box dimensions were $7.1554 \times 7.1554 \times 50$ $nm^3$. Hydrophobic walls, modelled as direct 12-6 LJ

potential at $z = 0$ and $z = 50$ nm, were included at the bottom and top of the simulation box along the $z$ axis. Vacuum layers above and below the multilayer prevent water–wall and lipid–wall short-range non-bonded interactions.

Cholesterol and stigmasterol molecules were modelled with the CHARMM36 force field[38,39]. The TIP3P water model in CHARM[40–42] was used and included KCl salt in the simulations. First, double layers of cholesterol or stigmasterol were constructed on CHARMM-GUI, where the hydrophobic tails face each other[43,44]. The CHARMM36 force field parameters for cholesterol and stigmasterol, TIP3P water parameters and parameters for ions were obtained from CHARMM-GUI. Each layer in the double layer of cholesterol or stigmasterol contained 128 molecules. The double layer obtained from CHARMM-GUI was translated along the $z$ axis by 4 nm using the Visual Molecular Dynamics visualization programme[45], constructing multilayers of four layers containing 512 lipid molecules in total (Extended Data Fig. 6a). Water and ions were added on top of the multilayers. The cholesterol multilayer system contained 12,284 water molecules with 72 $K^+$ and $Cl^-$ ions and the stigmasterol multilayer system contained 12,363 water molecules with 74 $K^+$ and $Cl^-$ ions. The system was energy minimized and equilibration simulations for both cholesterol- and stigmasterol-containing systems were conducted using Gromacs 2019.4 (for details see Supplementary Note 4)[46,47].

To construct the systems with a reversed molecular orientation at the interface (top layer), 10% (13 molecules), 30% (39 molecules) and 50% (64 molecules) of the molecules were reversed in orientation compared with the equilibrium system. The Alchembed tool[48] was used to remove any overlaps between coordinates that might have appeared on reverting the orientation of molecules. Minimization and short equilibration runs were then conducted as described above (for details see Supplementary Note 4).

## Data availability

All data generated during this study are provided at https://is.gd/3cdZ6f. Source data are provided with this paper.

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

**Acknowledgements** We thank S. W. Grill, P. Fratzl, D. J. Müller and K. Simons for their most helpful and encouraging comments on our study. We thank D. Hahn, M. Nitschke and M. Müller for synthesis of thiocholenic acid, for performing ToF–SIMS measurements and for conducting ATR–FTIR measurements, respectively. L.D.R. acknowledges support from the Volkswagen Foundation. We thank C. Neinhuis for drawing our attention to the intriguing features of the *Collembola* cuticle.

**Author contributions** J.F., R.H., J.H., T.P. and L.D.R. designed and performed experiments and analysed data. P.R.P. and J.-U.S. performed molecular dynamics simulations. J.F., R.H., J.H., T.P., J.-U.S., L.D.R. and C.W. wrote the manuscript. All authors read and approved the final version of the manuscript.

**Funding** Open access funding provided by Leibniz-Institut für Polymerforschung Dresden e.V.

**Competing interests** J.F., R.H., L.D.R., T.P., J.-U.S. and C.W. are listed as co-inventors on patent application no. 102022104237.5. The remaining authors declare no competing interests.

**Additional information**
**Correspondence and requests for materials** should be addressed to Carsten Werner.

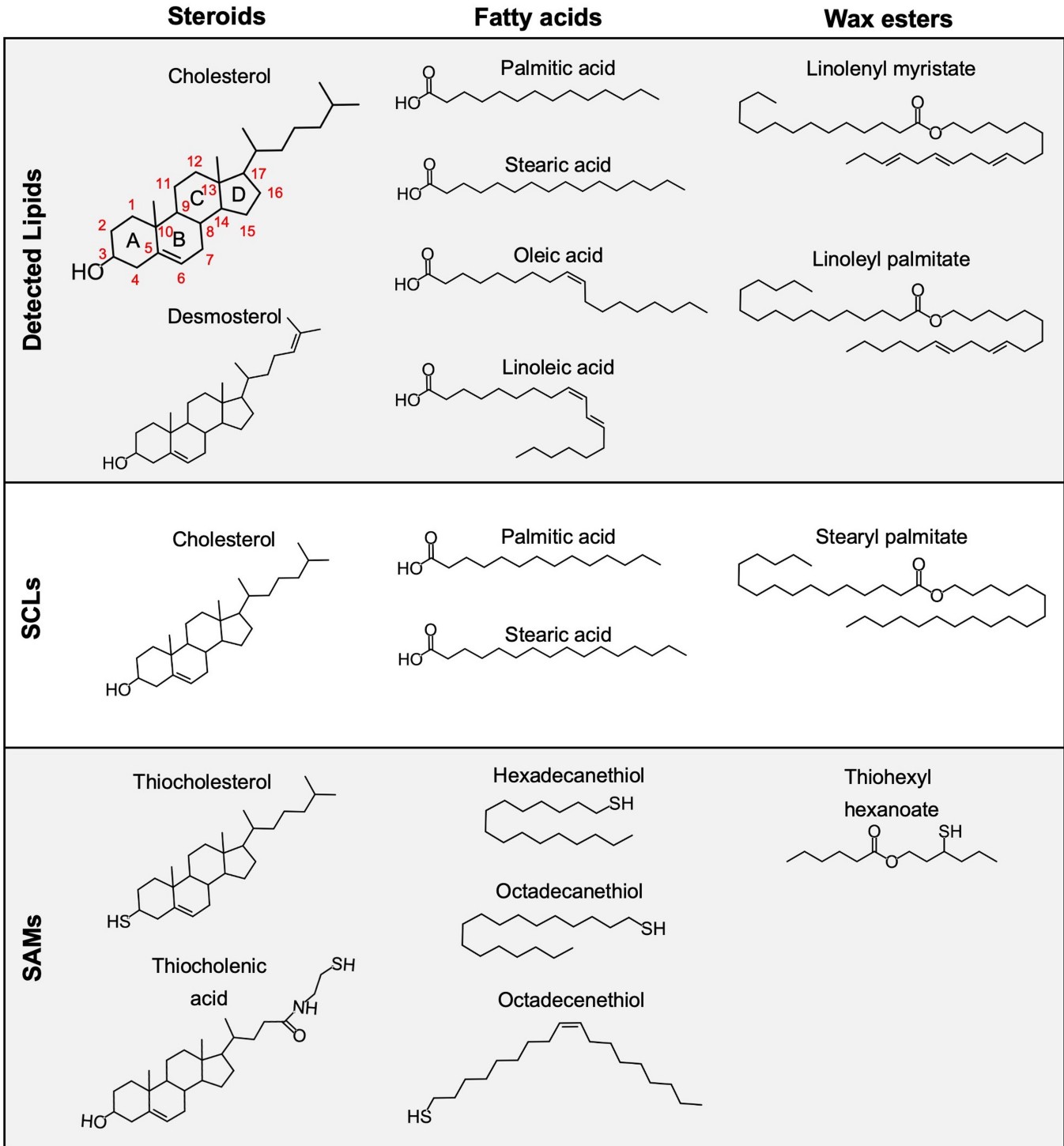

**Extended Data Fig. 1 | Overview of lipids used for layer preparation.**
Molecular structures of lipids detected in the cuticula of the Collembola
*T. bielanensis*[7] and lipids used in the preparation of SCLs and SAMs. SCLs were
prepared from cholesterol, palmitic and stearic acid, and stearyl palmitate.
Cholesterol and palmitic and stearic acid have been detected in the cuticula,
and stearyl palmitate is a substitute for the detected compound linoleyl
palmitate, which was not available in sufficient quantities as a synthetic
material. While both wax esters have the same hydrocarbon chain lengths,
linoleyl palmitate has two additional double bonds. Other lipids detected in the
cuticula were either not available as a synthetic material or unsuitable for the
preparation of SCLs (i.e. oleic and linoleic acid are liquids at room temperature

and immediately dewetted on the substrate after spin-coating). SAMs were
prepared using thiol-terminated lipids structurally similar to some of the lipids
detected in the Collembola cuticula. In thiocholesterol, the hydroxyl group at
the 3rd carbon atom of hydrocarbon ring A of cholesterol is replaced by a thiol
group and in cholenic acid, the cholesterol structure is chemically modified to
carry a thiol group at the end of the hydrocarbon chain attached to the 17th
carbon atom of hydrocarbon ring D (for details on the thiolation of cholenic
acid, see Supplementary Fig. 1). Hexadecanethiol, octadecanethiol, and
octadecenethiol are analogs of fatty acids detected in the cuticula, whereas
thiohexyl hexanoate represents a short-chain substitute for stearyl palmitate.

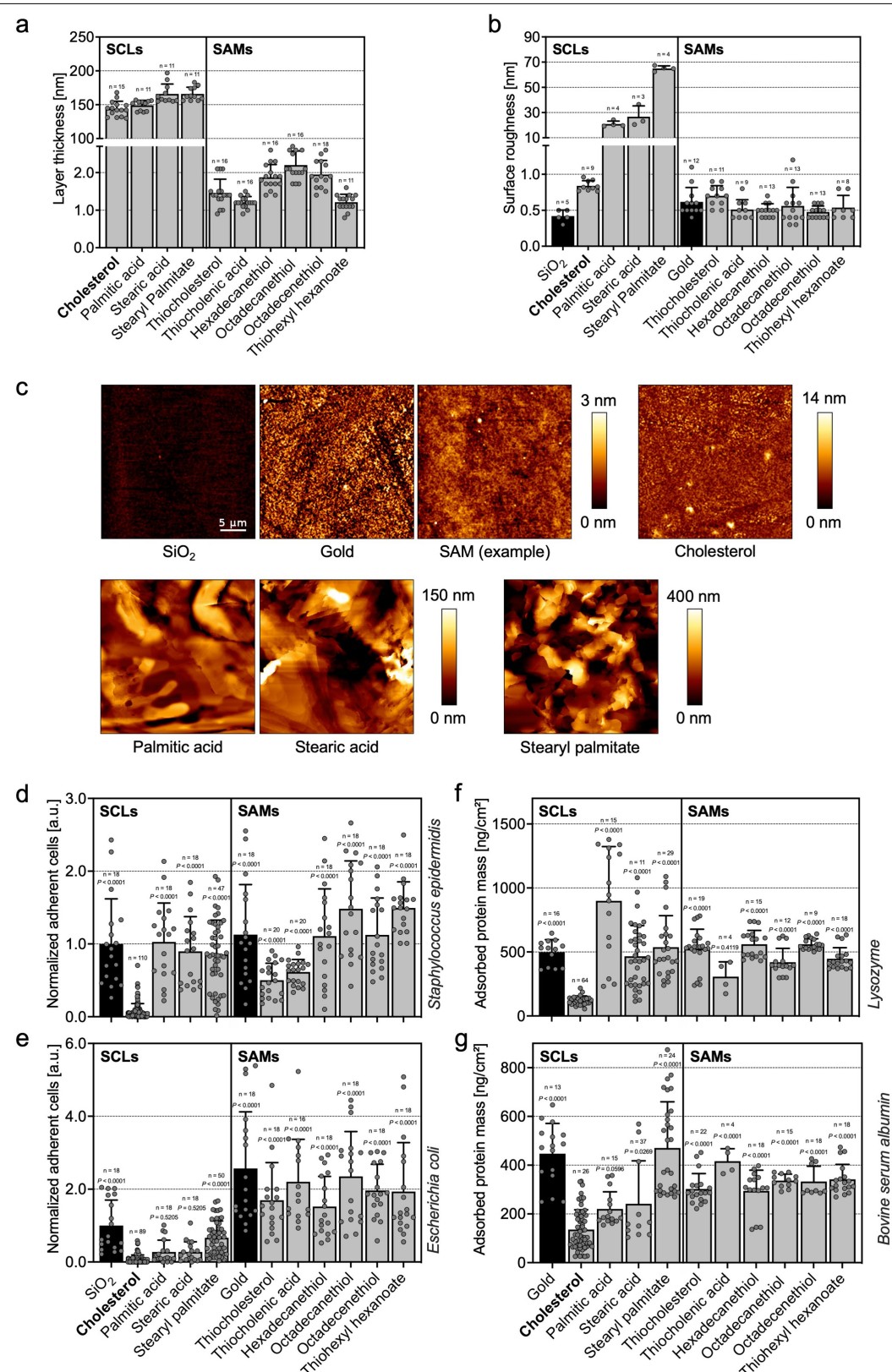

**Extended Data Fig. 2 | Intrinsic characteristics and bioadhesion of**
***Collembola* cuticular lipid layers.** (**a**) Layer thickness, (**b**) surface roughness
($R_a$), and (**c**) morphology of *Collembola* cuticular lipid layers as determined by
ellipsometry (layer thickness) and atomic force microscopy (surface roughness
and morphology). All images in panel (c) have the same scale. (**d,e**) Normalized
adherent cell density of (**d**) *S. epidermidis* and (**e**) *E. coli*. Data are normalized to

the average adherent cell density on the $SiO_2$ substrate. (**f,g**) Adsorbed amount
of (**f**) lysozyme and (**g**) bovine serum albumin as determined by quartz crystal
microbalance measurements. In all graphs, the mean + standard deviation are
shown. The number of observations (n) *is* indicated. *P* values (comparisons to
the cholesterol SCL condition) were determined using one-way ANOVA tests.

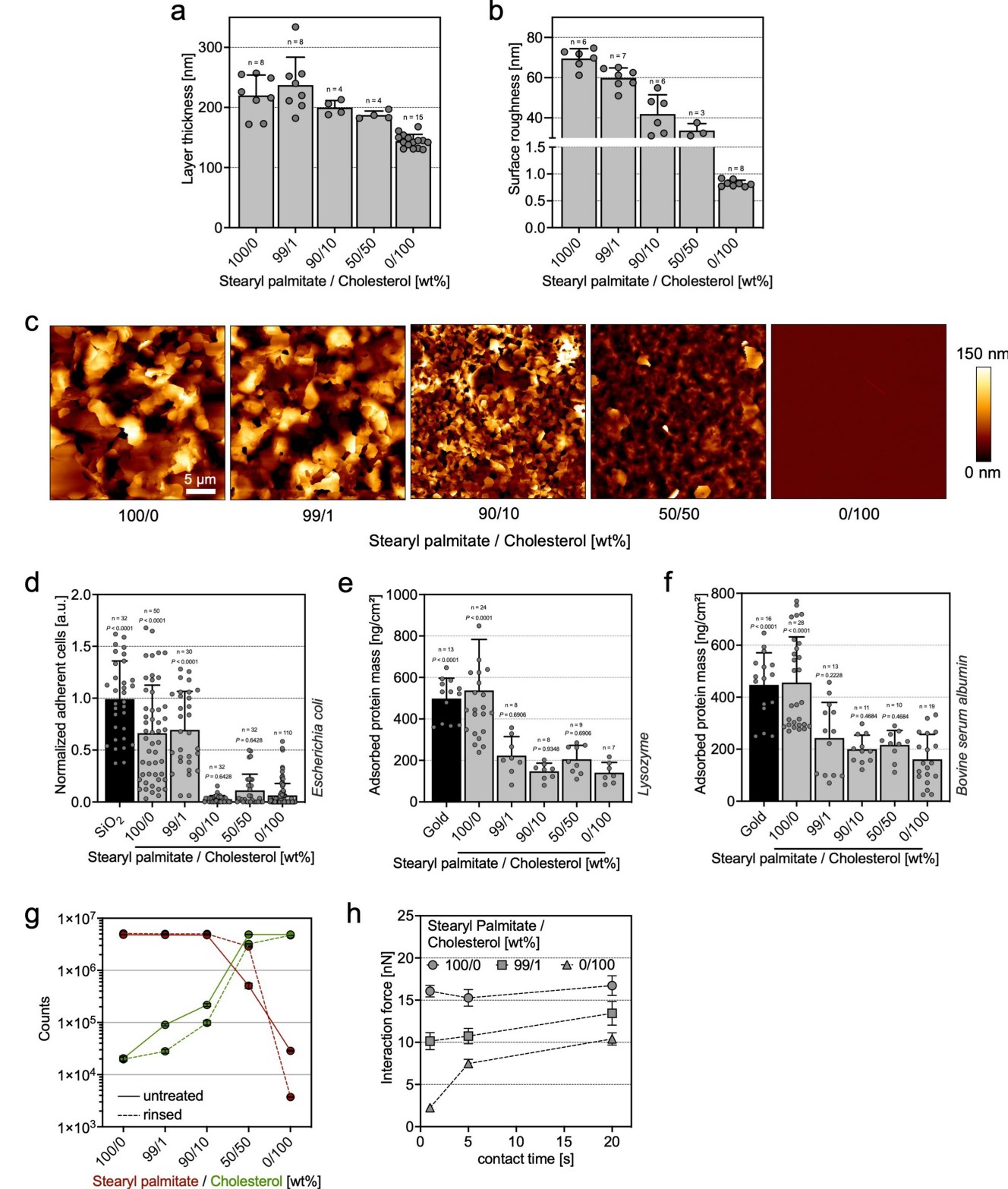

**Extended Data Fig. 3** | See next page for caption.

**Extended Data Fig. 3 | Bioadhesion characteristics of multicomponent SCLs.** (**a**) Layer thickness, (**b**) surface roughness ($R_a$), and (**c**) morphology of multicomponent SCLs containing stearyl palmitate and cholesterol in various weight ratios as determined by ellipsometry (layer thickness) and atomic force microscopy (surface roughness and morphology). All images in panel (c) have the same scale. (**d**) Normalized adherent cell density of *E. coli*. Data are normalized to the average adherent cell density on the $SiO_2$ substrate. (**e,f**) Adsorbed amount of (**e**) lysozyme and (**f**) bovine serum albumin as determined by quartz crystal microbalance measurements. The number of observations (n) is indicated. *P* values (comparisons to the condition "0/100") were determined using one-way ANOVA tests. (**g**) Characteristic ToF-SIMS signals of cholesterol (m/z 369.3) and stearyl palmitate (m/z 257.2), of multicomponent SCLs, as a function of layer composition. Rinsing the samples with Milli-Q water does not lead to significant changes in the detected signals, thus eliminating the possibility that the spatial arrangement of cholesterol and stearyl palmitate changes upon contact with polar fluids (i.e. interfacial enrichment of cholesterol on immersed multicomponent SCLs can be excluded as a reason for their anti-bioadhesive properties). In graphs (a-b, d-f), the mean + standard deviation are shown. (**h**) Contact time-dependent interaction forces (determined by AFM-based force spectroscopy) between a hydrophobic colloidal probe (Ø10 µm silica bead modified with hydrophobic silane) and the surface of multicomponent SCLs. The mean and ± standard error of the mean are shown in graph (h).

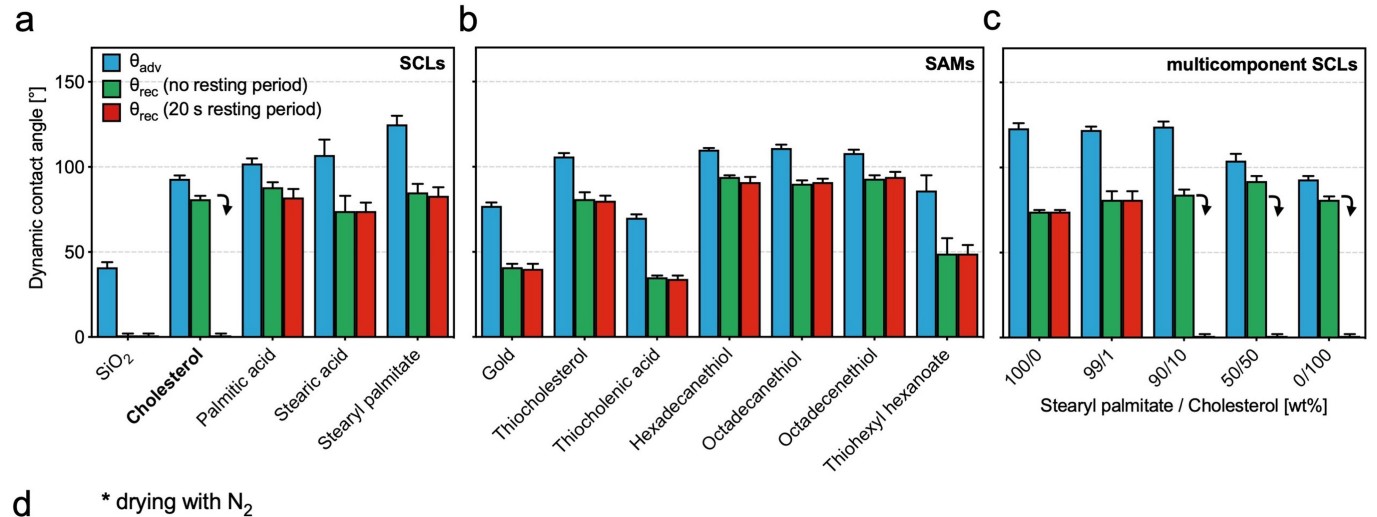

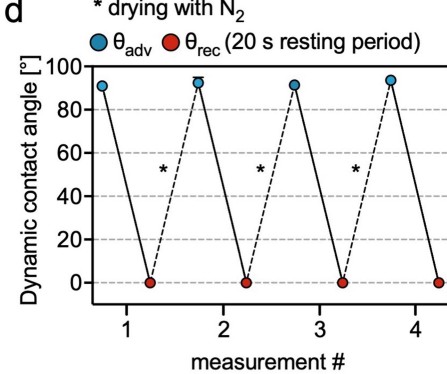

**Extended Data Fig. 4 | Dynamic water contact angle measurements.**
(**a**–**c**) Dynamic contact angle measurements on layers of Collembola cuticular lipids ((**a**) SCLs, (**b**) SAMs, and (**c**) multicomponent SCLs). For each sample type, the advancing ($\theta_{adv}$) and receding ($\theta_{rec}$) contact angle are displayed. For some samples the strong pinning of the three phase contact line after 20 s resting period (i.e. $\theta_{rec}$ is close to 0°) is additionally indicated by an arrow. Although palmitic and stearic acid are also amphiphilic, no resting-time-dependent decrease in the receding contact angle was observed for SCLs of either of these compounds. It can be assumed that the linear molecular structure of the fatty

acids (in contrast to the bulky steroid structure of cholesterol) causes tighter packing of the molecules in the SCL, limiting the mobility of the molecules. In all graphs, the mean + standard deviation are shown. (**d**) Successive contact angle measurements at the same position on a cholesterol SCL, where the measuring site is dried with nitrogen between two measurements. Consistent results are obtained (i.e. a hydrophobic advancing contact angle and no receding angle after a 20 s resting period). Data were obtained from at least three independent experiments.

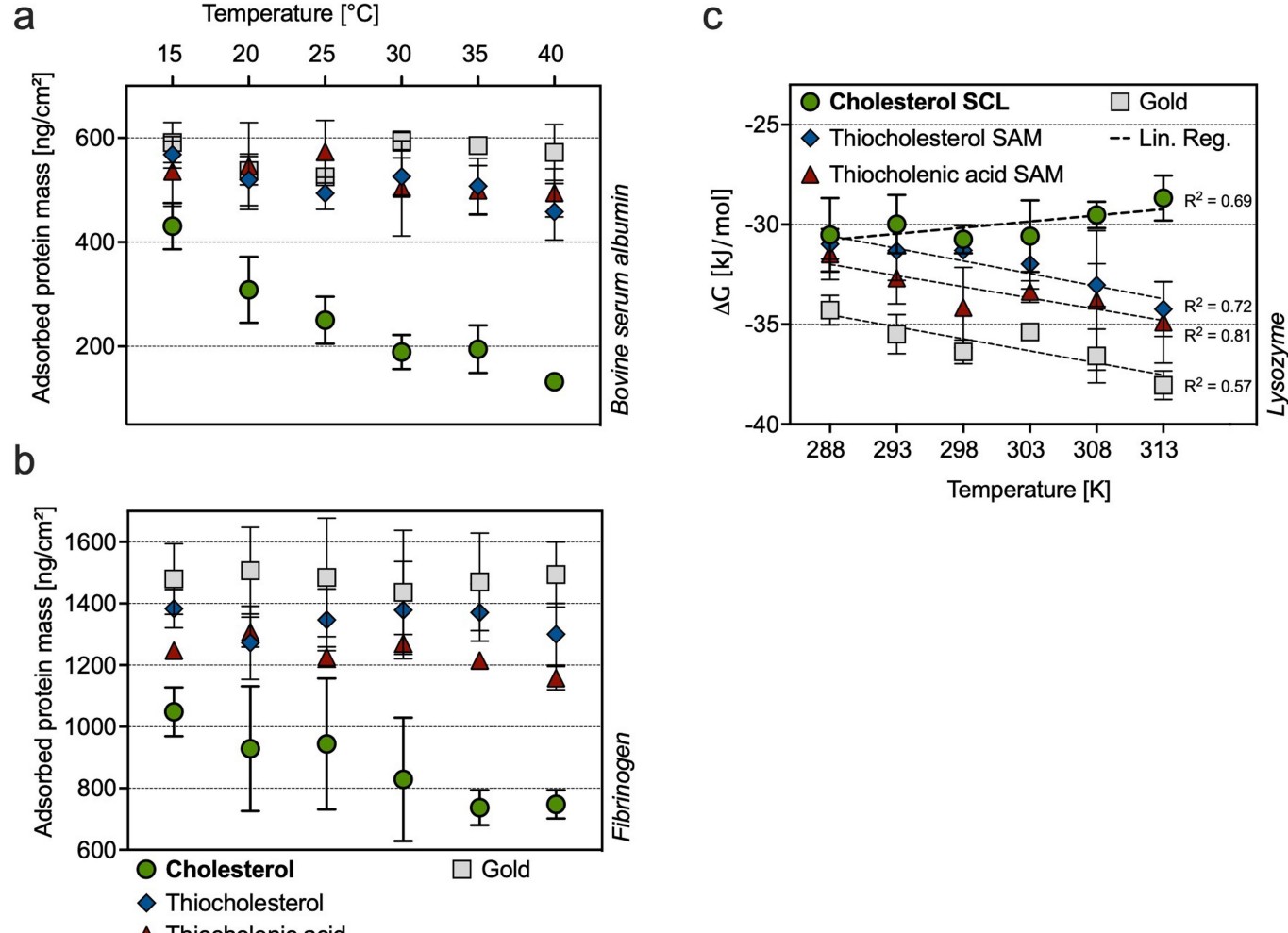

**Extended Data Fig. 5 | The temperature dependence of protein adsorption on cholesterol SCLs.** Temperature-dependent adsorbed amounts of (**a**) bovine serum albumin and (**b**) fibrinogen as determined by quartz crystal microbalance measurements. For cholesterol SCLs, a pronounced temperature dependence of protein adsorption is observed. Data were obtained from at least three independent experiments. The mean ± standard deviations are shown. (**c**) The temperature dependence of the Gibbs free energy of adsorption is derived from the amount of adsorbed lysozyme in quartz crystal microbalance measurements (Fig. 3a). Data points were calculated with $\Delta G = -RT \times \ln(q/(c^* - c^*q))$, where $R$ is the universal gas constant, $T$ the absolute temperature, $q$ the adsorbed surface fraction, and $c^*$ the normalized protein concentration ($c^* = 6.8 \cdot 10^{-6}$). Surface fractions were calculated as the ratio of the experimentally determined amount to the maximum amount of saturated monolayers of adsorbed lysozyme (for detailed explanation see Supplementary Note 2). The mean ± standard errors derived from the protein adsorption measurements (at least three independent experiments) are shown. The coefficients of determination ($R^2$) for linear fits of the first-principle determination of the entropic barrier $\Delta S$ (by $\Delta G = \Delta H - T\Delta S$) are also given in the figure.

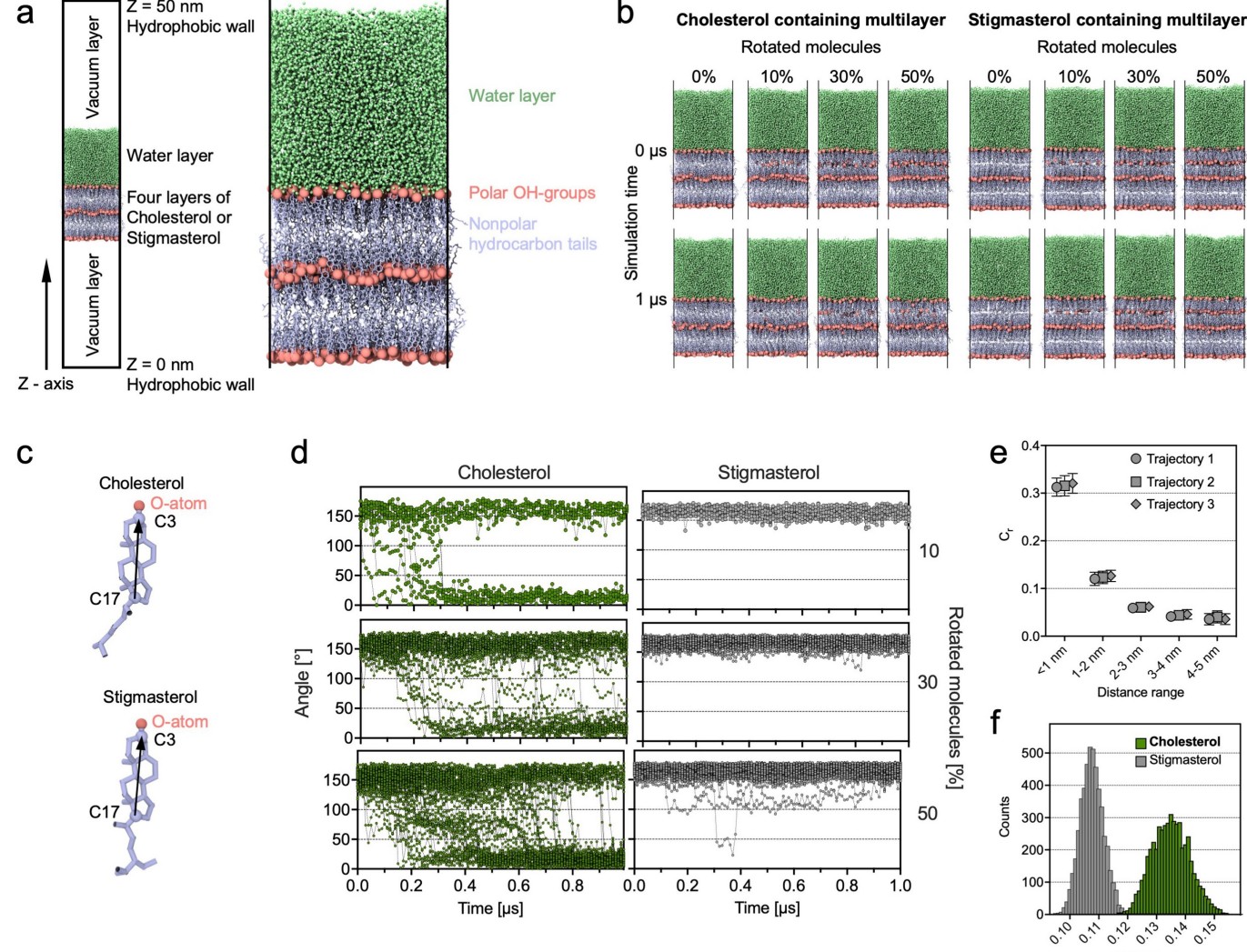

**Extended Data Fig. 6 | Molecular dynamics simulation of SCLs in contact with water.** (**a**) The left panel shows the simulation box for a system with four layers of cholesterol or stigmasterol in contact with a water layer and vacuum above and below. Hydrophobic walls were at z = 0 and z = 50 nm. The right panel shows the lipid multilayer–water system within the simulation box. For clarity, salt ions within the water layer are not shown. Oxygen atoms on the hydrophilic hydroxyl group of cholesterol or stigmasterol are shown as spheres and hydrophobic tails as blue sticks. (**b**) Simulation snapshots at 0 and 1 μs from a molecular dynamics (MD) trajectory for cholesterol and stigmasterol multilayer systems in contact with water with equilibrium states (0% rotated molecules) and intentionally reverted molecular orientation of 10, 30 and 50% of molecules in the interface layer. At 0 μs the production simulations were started after performing multiple equilibration steps (for details see Methods & Supplementary Note 4). (**c**) Visual representation of the vector connecting the C3 and C17 atoms on cholesterol or stigmasterol. This vector is used as the director for each molecule in the correlation analysis and the calculation of the angle of molecules at the interface layer (relative to the z-axis). (**d**) Angles between the vector connecting the C3 and C17 atoms and the z-axis from one representative simulation trajectory for lipid multilayer systems with 10, 30, and 50% reverted molecules. Cholesterol multilayer systems show spontaneous reorientation within 1 μs of simulation time, whereas no reorientation is observed for stigmasterol controls. (**e**) Correlation function ($C_r$) as a function of the distance to a reference molecule, plotted for three simulation trajectories of stigmasterol multilayer systems (without changed orientation of molecules at the interface), respectively. Error bars at each point represent the standard deviation at that point over the time intervals. (**f**) Histogram of mean RMSD over three simulation trajectories of z-component of the position of all oxygen atoms at the interface layer in the time range 0.5 to 1 μs shows higher fluctuation of the interface for cholesterol-containing multilayer system.

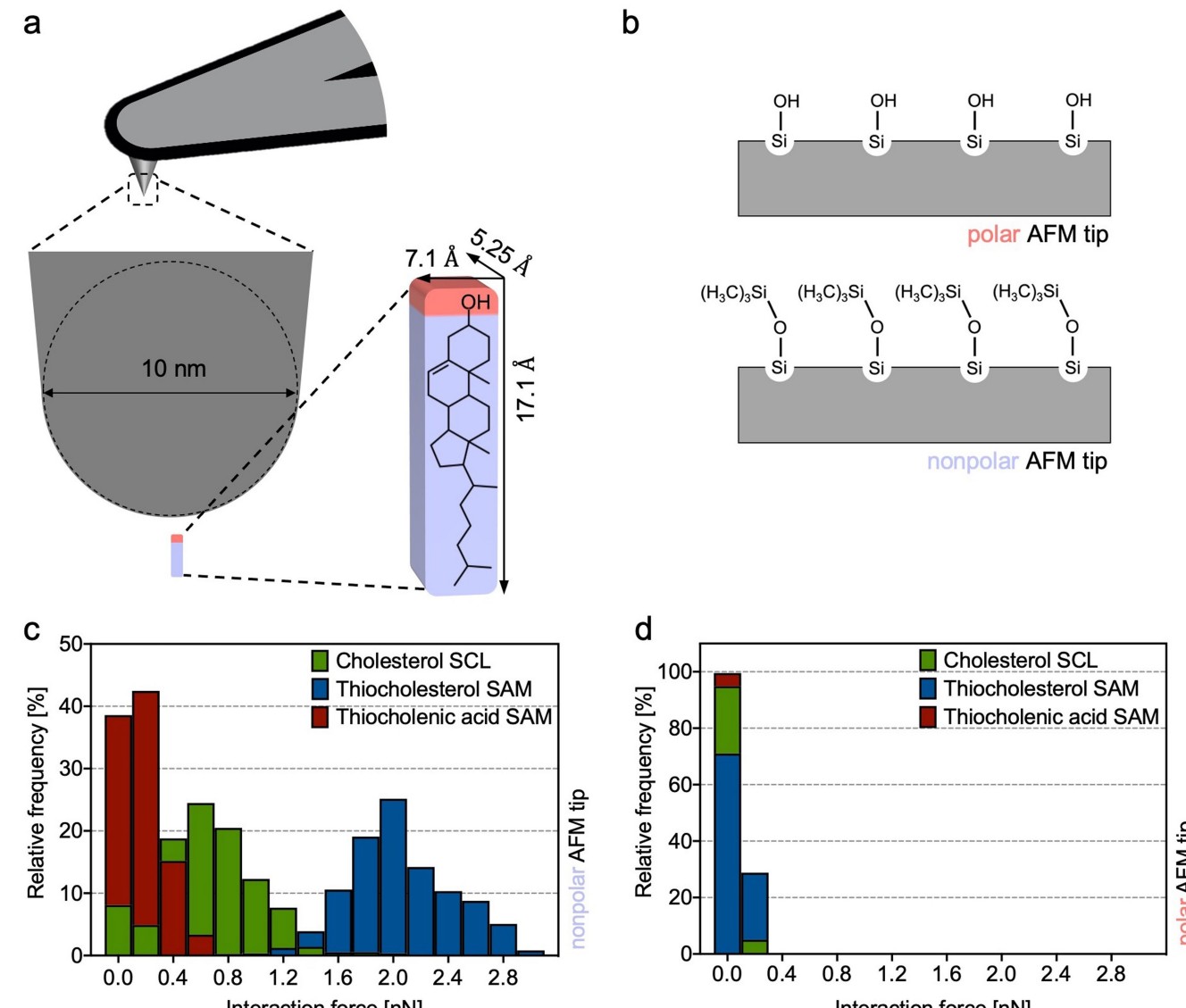

**Extended Data Fig. 7 | AFM-based force spectroscopy to probe the simultaneous exposure of polar and nonpolar molecular residues by cholesterol SCLs in a polar liquid.** (**a**) AFM-based, high-speed force mapping (*Quantitative Imaging*™, see Methods) was applied to detect the simultaneous exposure of polar (i.e., the cholesterol hydroxyl group) and nonpolar (i.e. the cholesterol hydrocarbon chain) molecular residues at the solid-liquid interface of cholesterol SCLs immersed in a polar liquid. Using polar (by applying an $O_2$ plasma in a plasma chamber) or nonpolar (hexamethyldisilazane coating) AFM tips (see (**b**)), interaction forces were quantified in a grid pattern on cholesterol SCLs, thiocholesterol SAMs (i.e., with the nonpolar cholesterol hydrocarbon chain permanently exposed), and thiocholenic acid SAMs (i.e. with the polar cholesterol hydroxyl group permanently exposed). The size ratios between an AFM tip and an individual cholesterol molecule are schematically shown. Although the AFM tip is very sharp (the typical radius of curvature of the tip is less than 10 nm), it is still significantly larger than a single cholesterol molecule. Thus, the individual force-distance curves record the integrated interactions of several cholesterol molecules. Force-distance curves were acquired at a high frequency (i.e., a few milliseconds per force-distance curve) that is sufficiently faster than the measured time for hydrophobic equilibration (i.e. a few seconds, Fig. 2c), thus capturing a 'snapshot' of the molecular surface orientation. (**c**) For nonpolar AFM tips, surface-dependent distributions of the interaction forces were found. For thiocholenic acid SAMs, predominantly low interaction forces were detected. The few higher forces may result from interactions of the AFM tip with the nonpolar part of the thiocholenic acid molecule, which can become accessible due to deformation of or defects in the SAM. Due to the hydrophobic interactions between the nonpolar AFM tip and the exposed nonpolar hydrocarbon tail of the thiocholesterol molecule, significantly higher interaction forces were detected for thiocholesterol SAMs. For cholesterol SCLs, the interaction forces were observed to span the range of forces measured for the two SAMs studied, supporting the hypothesis that both polar and nonpolar residues are exposed on immersed cholesterol SCLs. (**d**) For polar AFM tips, no difference between the tested surfaces were detected. Data were obtained from at least three independent experiments.

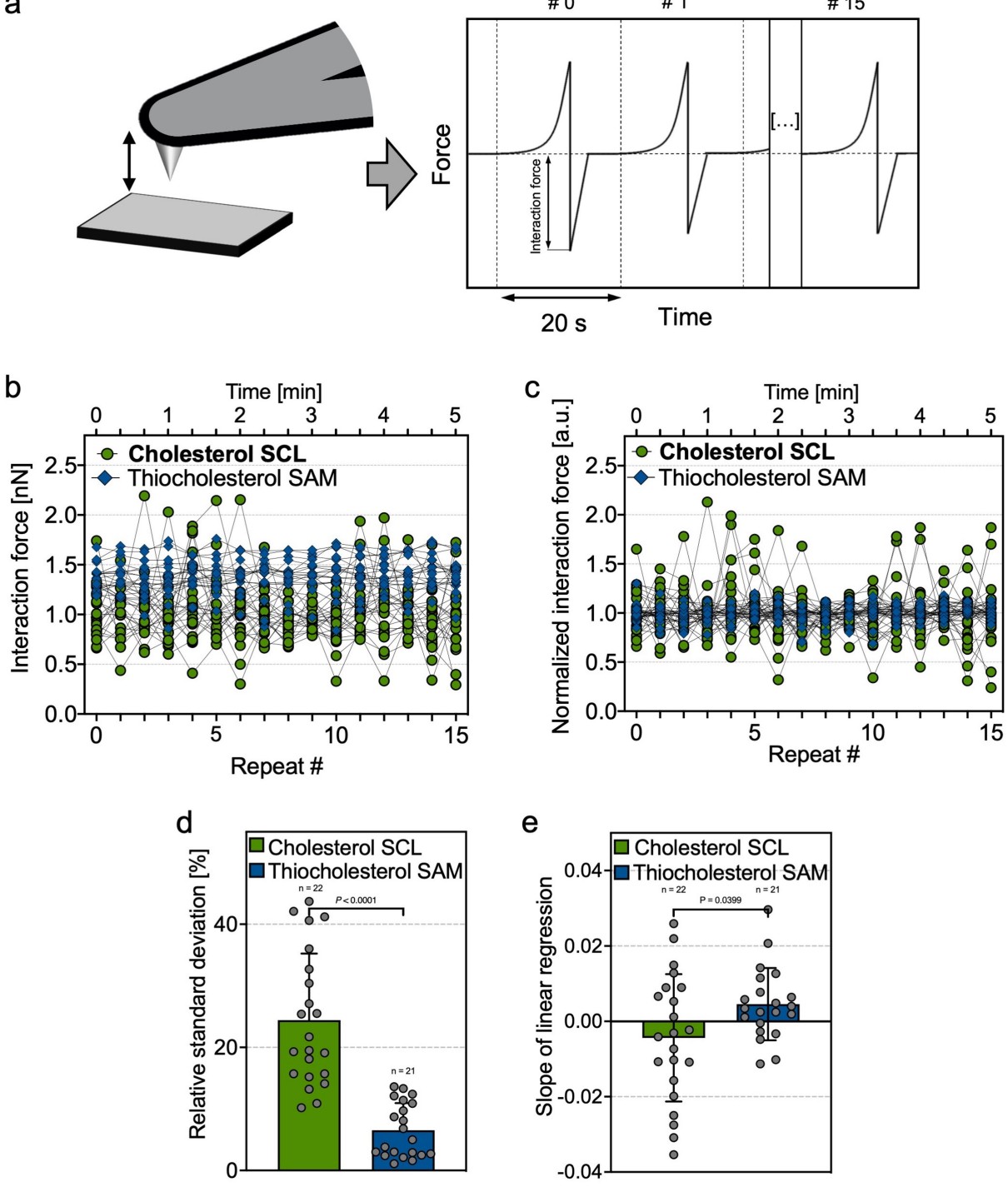

**Extended Data Fig. 8 | AFM-based force spectroscopy to study the interfacial dynamics of cholesterol SCLs in a polar liquid.** (**a**) The interaction forces of a nonpolar AFM tip (see Extended Data Fig. 7b) with cholesterol SCLs and thiocholesterol SAMs were repeatedly quantified at a singular location on the sample surface. To minimize the influence of the AFM tip on the dynamics of the interfacial cholesterol molecules, the measurement parameters were chosen in such a way that a pause of 20 s was maintained between two consecutive measurements. Sixteen consecutive force-distance measurements were obtained per position with high spatial repeatability to ensure that an identical set of surface molecules contributed to the interaction with the AFM tip. At least six different locations per condition were analyzed on three independent sample surfaces. (**b**) Plot of the absolute interaction forces for all measurement series. The consecutive number of force-displacement curves measured at a

single point and the elapsed time are plotted on the lower and upper x-axes, respectively. The individual measurement series were normalized to the mean value of each series. The resulting values are plotted in (**c**). (**d**) The distribution of the relative standard deviations determined from the individual curves in (c). The mean + standard deviation are shown. (**e**) Average slopes of linear regressions fitted to the individual measurement series presented in (c). There is only marginal evidence of a systematic change in interaction forces over the course of an individual measurement series (i.e., the recording of 16 force-distance curves at one position on the sample), whereby any influence of the measurement process due to wear or contamination of the modified AFM tip or the sample surface can be excluded. The mean ± standard deviation are shown. The number of observations (n) is indicated. P values were determined using unpaired t-tests.

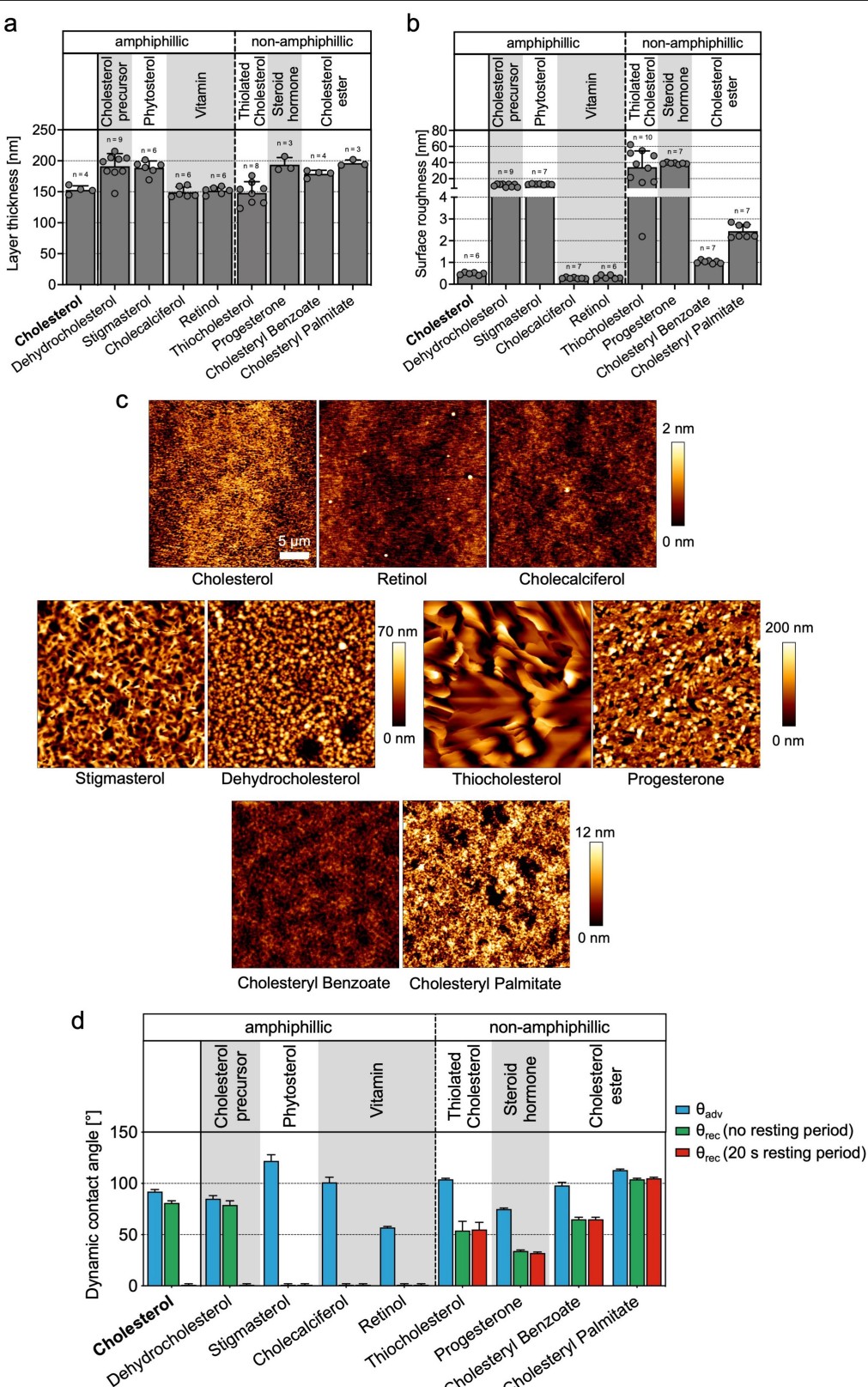

**Extended Data Fig. 9** | See next page for caption.

**Extended Data Fig. 9 | Characteristics of SCLs prepared from cholesterol analogs.** (**a**) Layer thickness, (**b**) surface roughness, and (**c**) morphology of cholesterol analog SCLs as determined by ellipsometry and atomic force microscopy. The contact mode atomic force microscopy images in (**c**) were used to determine the surface roughness. While all SCLs have a relatively uniform thickness of 150 – 200 nm, their morphology and roughness vary. However, a significant influence of the morphological surface properties on the bioadhesion properties is excluded, as e.g. cholesterol and dehydrocholesterol SCLs show a strongly diverging surface roughness but very similar bioadhesion properties (see Extended Data Fig. 10). (**d**) Dynamic contact angle measurements of the cholesterol analog SCLs. For SCLs of non-amphiphilic cholesterol analogs, only a surface roughness-dependent contact angle hysteresis (i.e. the rougher the surface, the higher the hysteresis due to pinning effects at elevations of the surface) but no resting time-dependent decrease of the receding contact angle (i.e. a dynamic interfacial polarity adaptation in response to changes in the polarity of the environment) is observed. Only for cholesterol and dehydrocholesterol SCLs a contact time-dependent decrease of the receding contact angle could be detected. In all graphs, the mean and + standard deviation are shown. The number of observations (n) is indicated.

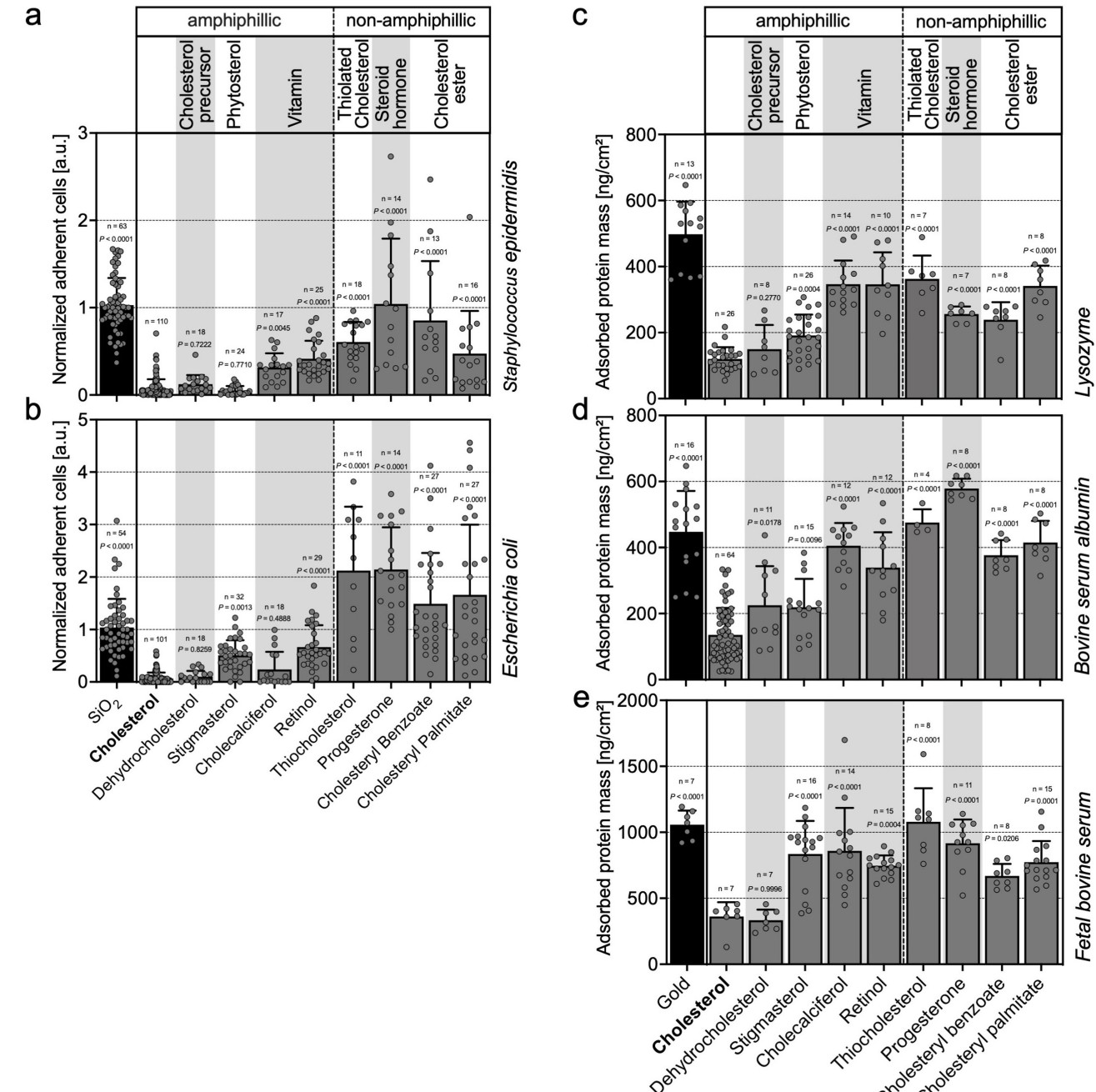

**Extended Data Fig. 10 | Bioadhesion characteristics of cholesterol analog SCLs.** Normalized adherent cell density of (**a**) *S. epidermidis* and (**b**) *E. coli* on cholesterol analog SCLs. Data are normalized to the average adherent cell density on the SiO₂ substrate. (**c,d,e**) Adsorbed amount of (**c**) lysozyme, (**d**) bovine serum albumin, and (**e**) 10% fetal bovine serum on cholesterol analog SCLs as determined by quartz crystal microbalance measurements. In all graphs, the mean + standard deviation are shown. The number of observations (n) is indicated. *P* values (comparison with the cholesterol condition) were determined using one-way ANOVA tests.