## [Peer Review File · Nature]

Manuscript Title: Entropic repulsion of cholesterol-containing layers counteracts bioadhesion

Reviewer Comments & Author Rebuttals

Reviewer Reports on the Initial Version:

Referees' comments:

Referee #1 (Remarks to the Author):

This paper describes an exciting new finding in bio-adhesion: and entropically driven repulsion mechanism, based on cholesterol island formation. The idea at the basis of this finding is novel and the proposed explanation is truly intriguing.

In the present form, no matter how much I would love this new theory to be true, I do not believe that the presented data come even close to supporting the data. Among many issues that I have with the paper, I will focus on the two key ones that I find most troubling. First, the key (negative) control for sample studies in the paper is self-assembled monolayers (SAMs) of cholesterol. These SAMs are supposed to show that, when cholesterol is not mobile, then it does not repel protein non-specific adsorption (and consequently bio-adhesion). There are two major problems with these samples used as control. First, protein adsorption often starts on defect (inevitable in monolayers) in the sample once a protein adsorbs on these defect, that (mainly if it misfolds) it acts as a nucleation point for other protein, basically short-circuiting the role of the molecular monolayers. Hence, all the adsorption observed by the authors could simply be due to defects. Second, if I understand correctly the authors used SAMs solely composed of cholesterol as controls for wax/cholesterol spin-coated sample where cholesterol was 10%, how can this be acceptable? (And were they to use mixed SAMs what would be the morphology of the SAMs and are we sure in that case there is no mobility?)

The second major problem I have is with Fig. 3a. It is a central point in their argument. By showing adsorption that decreases with temperature they can state that it is an entropy-driven repulsion. If one looks at the error bars in the figure, how can we conclude that there is a linear (note the linearity is needed for the theory to work) trend there? What is the R^2 of the fitting? Does it pass a statistical test? When correctly propagating the error to the fit can they defend the thermodynamic quantities that they extracted? Honestly, I am skeptical.

One last minor point I would like to mention is that the author presents measurements to resistance to protein non-specific adsorption, the accepted first step in bio-adhesion. They never clarify the difference between non-specific adsorption and bio-adhesion. It would be nice to clarify this point. In conclusion, I think this paper needs the authors to revisit the sample they chose as control, they need to think of a control sample that has the same thickness than their key sample and the same defect structure. They should also better substantiate the data presented in Figure 3a. The authors could for example think of creating a form of cholesterol that can be cross-linked to the wax. In the current form of the paper, I cannot support its publication anywhere.

Referee #2 (Remarks to the Author):

The authors characterized spin-coated cholesterol multilayers (SCLs) as a means to understand the antifouling role of molecular assemblies in nature, and to identify potential bioinspired synthetic antifouling strategies. The work is motivated by *Collembola* (springtail), which is known to employ a structured cuticle surface to inhibit biofouling. The authors are claiming in this report that the structured approach to antifouling is supplemented by a molecular antiadhesive involving cholesterol (Chol), and that the antifouling effect of Chol is based on entropic repulsion connected to the dynamics of Chol. The authors used a combination of physical, chemical and biological surface characterizations of SCLs to support their arguments. Strengths of this manuscript are that it is well written, well executed technically, and the entropic repulsion concept interesting to ponder. However, the entropic repulsion argument is supported by only limited data, and the level of antifouling performance demonstrated by SCLs does not appear to be remarkable compared to existing antifouling technologies.

Specific Comments To The Authors:

1. In their analysis of the thermodynamics of the system, the authors have considered primarily the dynamics and orientational changes in the SCL, but have not adequately considered other contributions, such as the dynamics and potential denaturation of protein upon adsorption, and interaction of protein with Chol. These can also make enthalpic and entropic contributions to overall free energy. For example, Pegg et al (*Protein & Peptide Letters*, 2008, Vol. 15, No. 4) showed that Chol-BSA complexation is enthalpy driven. In my view it is also likely that adsorption of protein to SCL will be accompanied by release of water bound to protein and surface. There should be an entropic component to this. On page 6, the authors suggest there are other entropic effects in protein adsorption but do not elaborate. What are these other effects and is a detailed accounting of them possible?
2. The authors argue in Figure 3 that protein adsorption will lead to an entropic penalty due to locally restricting orientational dynamics of Chol molecules. Is there experimental evidence for this? Was FRAP performed on protein adsorbed SCLs?
3. The practical significance of cholesterol-mimetic synthetic antiadhesive systems is unclear. For example, it is noted that the cholesterol multilayers support roughly 150-400 ng/cm² of protein adsorption depending on the protein (less for BSA, more for serum). This is actually a lot of protein, in fact very large in comparison to better known protein antifouling technologies such as OEG-terminated SAMs and zwitterionic polymers, which allow only a few ng/cm² protein adsorption under similar conditions. The bacterial adhesion performance of SCL is harder to assess, as bacterial adhesion was shown to be significantly reduced but log reduction values were not included (perhaps I missed them). Good bacterial antifouling technologies reduce bacterial adhesion by 3-4 log or more.
4. Although I am not a *Collembola* expert, what information I could find suggests that it is a terrestrial organism that likes moist environments but apparently does not live under water. This left me wondering about how predictive antifouling experiments conducted fully under water can be to understanding *Collembola* antifouling.

5. Bacterial adhesion was assayed by incubating substrates in bacteria suspended in lysogeny broth (LB). LB is a soup of many things including peptides. Like proteins, peptides presumably adsorb to SCL. Therefore, bacterial adhesion to SCL is likely mediated by preadsorbed peptides from the broth. What does this mean for the interpretations of the authors?

6. In the bacterial adhesion assay, the methods describe fixing and then imaging by SEM. This seems very unusual for counting of adhered bacteria. Not enough information is provided on certain aspects (e.g. # cells counted). Raw numbers of adhered bacteria should be reported.

7. In the single cell force spectroscopy experiment, it is unclear if the authors confirmed that the presence and orientation of the cell is unchanged after the contact experiment. Was the cell intact? It seems that the orientation and location of the cell on the AFM cantilever should affect contact area and therefore interaction force with the SCL surface. Did the authors measure this or account for differences in orientation/contact area of cells? The reference cited for the method (#33) has little to say about this.

Referee #3 (Remarks to the Author):

1. Description of key achievements of the paper. Please let us know in a paragraph or two the advance presented in the paper, and your opinion of the potential excitement of the findings for your community more generally. Will the work open up completely new research directions or technological opportunities? Are there any outstanding features that deserve a special mention?

In this manuscript, Jens Friedrichs et al reported that entropic repulsion caused by dynamic interfacial fluctuations of cholesterol layers strongly restricts protein adsorption and bacterial adhesion. The adsorption of proteins is of high physiological relevance, and as such they adsorb with different mechanisms than their molecular or atomic analogs. Compare to current studies in the field of bio-adhesions, I am delighted to see a small biomolecule could show special functions in this way. How cholesterol influence the bio-adhesion process revealed from this work may inspire researchers proposing new strategies for designing antifouling molecules in the future.

2. Novelty. If the conclusions are not original, it would be very helpful if you could provide relevant references, authors invariably question criticisms of novelty if there are no supporting references. The authors claimed that the entropic repulsion mechanism was previously unknown. However, there have been many studies of this mechanism, which have been reported to be useful in anti-adhesion (eg. P. G. D. Gennes et al. *J. Colloid Interface Sci.* 1991, 142, 149-158; Julie M. Goddard et al. *J. Agric. Food Chem.* 2012, 60, 2943–2957; Dong Keun Han et al. *Progress in Polymer Science* 2015, 44, 28–61). In particular, as early as 1991, the P. G. D. Gennes proposed that the approach of the protein initiates the compression of PEO chains, which induces a decrease in entropy and a steric repulsion effect. Therefore, the authors claimed the entropic repulsion mechanism is fully novel, which is not justified. Besides, the authors proposed that spatially constrained cholesterol SAMs showed less anti-adhesive effects, which is similar to the previous report (see S. Jiang et al. *J. Phys. Chem. B* 2005, 109, 2934-2941). Jiang have proposed that proteins adsorb more on the densely

packed SAMs of EG2OH and EG4OH, while EG6OH-SAMs generally resist protein adsorption due to lower packing density. In addition, self-assembled molecular surfaces that are resistant to protein adsorption have also been extensively reported. For example, in 1991, G. M. Whitesides et al. have reported the model systems of self-assembled organic monolayers for studying adsorption of proteins at surfaces (Science 1991, 252: 1164-1167; J. Am. Chem. Soc. 1993, 115: 10714-10721).

In my opinion, I would suggest the authors to prove if there are any advantages of the entropic repulsion mechanism for small molecular as cholesterol than traditional ones.

3. Technical robustness. Does the manuscript have technical flaws which should prohibit its publication? If so, please provide details. Are these flaws resolvable with minor / major revisions?

1) Fig. 3a is a key experimental evidence used to discuss and quantification of entropic repulsion by cholesterol SCLs. However, I see a few questions in this figure. First of all, the slope of cholesterol SCLs between the temperature of 20~30 Celsius seemed unchanged. So, the reliability of this slope calculated with other two temperature points is deeply doubtful. Second, this experiment only tested 4 temperature points. More temperature points are needed (similar issues can also be found in Fig 2 a&b). Third, only one protein (bovine serum albumin) was tested. More proteins are need to be tested to confirm whether this result is universal.

2) Detailed structures inside cholesterol SCLs have not been shown, SAXS experiments may reveal more underlying facts.

3) The CA experiments showed that after contacting with water, a switch in the polar orientation of cholesterol have taken place in no more than 20 seconds. Though cholesterol showed slower switching speed than phytosterol and other cholesterol analogs, but still showed strong interactions with water molecular. It is known that the displacement of water from the surface of a hydrophilic material represents a large energy barrier to adsorption by proteins, would this be the case?

4) Phytosterol showed no resting time-dependent decrease of the receding contact angle, but also possessed impressive anti-bioadhesion behaviors as shown in Extended Data Fig. 14 for S. epidermidis, bovine serum albumin and lysozyme. How to understand this result, and it is also related to with entropic repulsion effect?

5) In practical environments, the stability of membrane is essential. The authors tried to prove the stability of cholesterol SCLs when measured with TOC after incubation in pure water for 24h. Since water molecules have strong influence on the conformation of cholesterol molecules, it would not be surprising to see the cholesterol SCLs fallen apart in a longer period. A stability test with 7 days and different temperatures is recommended.

Though the authors gave a large amount of description and data to illustrate this hypothesis, I cannot agree on many aspects. Thus, I cannot recommend its publication on Nature currently.

Referee #4 (Remarks to the Author):

The manuscript is interesting and touches theoretical and practical relevant topics.

I believe that if the authors revise the manuscript properly (major revisions), the article could be

published. Nature Physics could also be an option.

Author Rebuttals to Initial Comments:

Referees' comments:

Referee #1 (Remarks to the Author):

This paper describes an exciting new finding in bio-adhesion: and entropically driven repulsion mechanism, based on cholesterol island formation. The idea at the basis of this finding is novel and the proposed explanation is truly intriguing.

In the present form, no matter how much I would love this new theory to be true, I do not believe that the presented data come even close to supporting the data. Among many issues that I have with the paper, I will focus on the two key ones that I find most troubling. First, the key (negative) control for sample studies in the paper is self-assembled monolayers (SAMs) of cholesterol. These SAMs are supposed to show that, when cholesterol is not mobile, then it does not repel protein non-specific adsorption (and consequently bio-adhesion. There are two major problems with these samples used as control. First, protein adsorption often starts on defect (inevitable in monolayers) in the sample once a protein adsorbs on these defect, that (mainly if it misfolds) it acts as a nucleation point for other protein, basically short-circuiting the role of the molecular monolayers. Hence, all the adsorption observed by the authors could simply be due to defects.

We are very pleased about the reviewer's comment on the novelty and relevance of our findings and address below the concern about the suitability of the used model surfaces:

Firstly, we agree that SAMs are extensively characterised molecular model layers¹ but may contain defects even when prepared with the highest care.² However, numerous reports nonetheless demonstrate that SAMs (e.g. OEG-based SAMs³) can nonetheless display excellent anti-adhesive properties.

To minimise the occurrence of defect structures in SAMs, we applied dedicated pre-cleaning protocols (RCA cleaning) and prepared the SAMs in a classified clean room laboratory using the highest quality grade chemicals. Moreover, all samples were used immediately after preparation. We regret this information was missing in the original submission and have extended the Material and Methods section of the revised manuscript accordingly:

“Gold substrates were cleaned by immersion in a solution of deionized water, ammonia and hydrogen peroxide (volume fraction 5/1/1) for 15 min at 70°C, rinsed repeatedly in Milli-Q water and then dried in a nitrogen stream. To minimise defects, the gold substrates were used for SAM formation immediately after cleaning.”

To determine the efficacy of these measures, we conducted additional experiments on the adsorption of fluorescently labelled bovine serum albumin (BSA) on thiocholesterol SAMs (**Response Letter Figure 1**). We analysed the homogeneity using the standard deviation as a parameter of BSA fluorescence after one hour of adsorption on glass, gold and a thiocholesterol SAM. Compared to glass, very low standard deviations were found on gold and SAMs, indicating that the protein layers adsorb homogeneously.

Response Letter Figure 1: Fluorescently labelled bovine serum albumin homogeneously adsorbs to thiocholesterol SAMs. $\sigma_{\text{Fluorescence}}$ is the standard deviation of the fluorescence signal of an image, indicating the homogeneity of surface coverage.

Furthermore, while thiocholesterol and thiocholenic acid SAMs were used here to explore how restricted molecular mobility deteriorates the anti-bioadhesive properties of cholesterol (independent of its orientation), other investigated systems of restricted molecular mobility similarly show significant levels of bioadhesion: For example, Spin-Coated Lipid Multilayers (SCLs) of non-amphiphilic cholesterol analogs lacking the terminal hydroxyl group neither responded to changes of the environmental polarity (see wetting experiments in **Extended Data Fig. 15d** of the revised manuscript) nor exhibited anti-bioadhesive behaviour (**Fig. 4** and **Extended Data Fig. 16** of the revised manuscript).

Second, if I understand correctly the authors used SAMs solely composed of cholesterols as controls for wax/cholesterol spin-coated sample where cholesterol was 10%, how can

this be acceptable? (And were they to use mixed SAMs what would be the morphology of the SAMs and are we sure in that case there is no mobility?)

We regret that our original manuscript was apparently not sufficiently clear and caused a misunderstanding: Thiocholesterol and thiocholenic acid SAMs were used in our study solely for comparison with cholesterol SCLs.

*Beyond that, we present data on multicomponent SCLs in **Fig. 1h, i** and **Extended Data Fig. 7** of the revised manuscript to show that even a small amount of cholesterol in an otherwise adhesion-promoting matrix reduces bioadhesion. Multicomponent SCLs were prepared from solutions containing stearyl palmitate and cholesterol in varying weight percentages (i.e., 1%, 10% and 50% cholesterol with 99%, 90% and 50% stearyl palmitate, respectively). Stearyl palmitate was chosen as the adhesion-promoting component of multicomponent SCLs (**Fig. 1f, g** of the revised manuscript). However, SAMs were in fact not considered in the context of multicomponent SCLs.*

To clarify this, the use of SCLs is marked in the caption of Fig. 1 h, i of the revised manuscript:

*“Normalized adherent cells of *Staphylococcus epidermidis* on (f) layers of *Collembola* cuticular lipids and (h) **multicomponent SCLs of stearyl palmitate and cholesterol**... Adsorbed amount of protein (lysozyme or fetal bovine serum) on (g) layers of *Collembola* cuticular lipids and (i) **multicomponent SCLs of stearyl palmitate and cholesterol**, determined by quartz crystal microbalance measurements. (i) **multicomponent lipid SCLs of stearyl palmitate and cholesterol**, determined by quartz crystal microbalance measurements.”*

The use of SCLs is also pointed out in the main text of the revised manuscript:

*“**Multicomponent SCLs containing stearyl palmitate** – a lipid that causes strong bioadhesion to its single-component SCLs (**Fig. 1f, g** and **Extended Data Fig. 7d, g**) – **and cholesterol**, with cholesterol contents ranging from 0 to 100 wt%, were investigated using similar bioadhesion assays...”*

In addition, the caption of Fig. 1e was adjusted accordingly:

*“**Self-Assembled Monolayers (SAMs)** chemisorbed to gold via thiol groups, with either the polar or nonpolar sides of cholesterol oriented to the interface, served as references **in selected experiments**.”*

The second major problem I have is with Fig. 3a. It is a central point in their argument. By showing adsorption that decreases with temperature they can state that it is an entropy-driven repulsion. If one looks at the error bars in the figure, how can we conclude that there is a linear (note the linearity is needed for the theory to work) trend there? What is the R² of the fitting? Does it pass a statistical test? When correctly propagating the error to the fit can they defend the thermodynamic quantities that they extracted? Honestly, I am skeptical.

We very much appreciate the critical comment of the reviewer on this particularly important part of our report and made the following changes and additions to address them:

The line-like emphases in Fig. 3a of the revised manuscript of the temperature-dependent adsorbed protein data were removed as they might have suggested linearity which is neither given nor required for the analysis. In fact, it is the linear temperature-dependence of the free energy of adsorption (ΔG) data that is used to derive ΔS from the slope. To improve the precision of this evaluation, a higher number of measurement points (temperatures) within the accessible range was recorded in all protein adsorption experiments (see Fig. 3a & Extended Data Fig. 10 of the revised manuscript).

Fig. 3a - Temperature-dependent adsorbed amounts of lysozyme as determined by quartz crystal microbalance measurements. For cholesterol SCLs, a pronounced temperature dependence of lysozyme adsorption is observed. Data were obtained from at least three independent experiments. The mean \pm standard deviations are shown.

The more comprehensive data sets were subjected to a thorough statistical analysis (see **Extended Data Fig. 11** & **Extended Data Calculations Part 2** of the revised manuscript), confirming validity and significance of the evaluation.

Extended Data Fig. 11 - The temperature dependence of the Gibbs free energy of adsorption is derived from the amount of adsorbed lysozyme in quartz crystal microbalance measurements (Fig. 3a). Data points were calculated with $\Delta G = -RT \times \ln(q / (c^* - c^*q))$, where R is the universal gas constant, T the absolute temperature, q the adsorbed surface fraction, and c^* the normalized protein concentration ($c^* = 6.8 \cdot 10^{-6}$). Surface fractions were calculated as the ratio of the experimentally determined amount to the maximum amount of adsorbed lysozyme (see Extended Data Calculations Part 2). The coefficients of determination (R^2) for the individual fits are given in the figure.

Furthermore, to demonstrate the generality of the uncovered anti-adhesive effect, temperature-dependent adsorption data of two additional globular proteins – lysozyme and fibrinogen – were collected and evaluated (see Fig. 3a & the newly inserted **Extended Data Fig. 10** of the revised manuscript). Lysozyme and fibrinogen displayed entropic repulsion effects similar to serum albumin when adsorbed at cholesterol SCLs, indicating the significant entropic barrier counteracting bioadhesion at cholesterol SCLs to be a ubiquitous phenomenon (**Extended Data Calculations Part 2**).

Extended Data Fig. 10 - The temperature dependence of protein adsorption on cholesterol SCLs. Temperature-dependent adsorbed amounts of (a) bovine serum albumin and (b) fibrinogen as determined by quartz crystal microbalance measurements. For cholesterol SCLs, a pronounced temperature dependence of protein adsorption is observed. Data were obtained from at least three independent experiments. The mean \pm standard deviations are shown.

One last minor point I would like to mention is that the author presents measurements to resistance to protein non-specific adsorption, the accepted first step in bio-adhesion. They never clarify the difference between non-specific adsorption and bio-adhesion. It would be nice to clarify this point.

We fully agree with the reviewer that protein adsorption and bacterial/cellular adhesion to surfaces are distinct but connected and occur subsequently. Since protein adsorption and bacterial adhesion were both assessed in our reported study to holistically determine and score the efficacy of entropic repulsion, we summarized both phenomena with the term 'bio-adhesion' in the revised manuscript and rephrased the text as follows:

“While molecular and structural characteristics governing interfacial phenomena in nature are widely studied, the physical mechanisms underlying the control of bioadhesion – the interfacial accumulation of biopolymers and cells (including bacteria) – are not yet thoroughly understood.”

In conclusion, I think this paper needs the authors to revisit the sample they chose as control, they need to think of a control sample that has the same thickness than their key sample and the same defect structure. They should also better substantiate the data presented in Figure 3a. The authors could for example think of creating a form of cholesterol that can be cross-linked to the wax. In the current form of the paper, I cannot support its publication anywhere.

We very much appreciate all the most helpful critical comments and suggestions of the reviewer. Referring to the answers given above, we would like to point out that the data analyzed to determine the repulsive entropic barrier against bioadhesion to cholesterol SCLs were extended (three different globular proteins were now used, a higher number of temperatures within the accessible range was recorded). Also, the revised manuscript provides a more detailed explanation of the choice and characteristics of the set of samples and controls used in our study. By that, we hope to meet the expectations of the reviewer for the validation of our newly proposed mechanism.

Referee #2 (Remarks to the Author):

The authors characterized spin-coated cholesterol multilayers (SCLs) as a means to understand the antifouling role of molecular assemblies in nature, and to identify potential bioinspired synthetic antifouling strategies. The work is motivated by Collembola (springtail), which is known to employ a structured cuticle surface to inhibit biofouling. The authors are claiming in this report that the structured approach to antifouling is supplemented by a molecular antiadhesive involving cholesterol (Chol), and that the antifouling effect of Chol is based on entropic repulsion connected to the dynamics of Chol. The authors used a combination of physical, chemical and biological surface characterizations of SCLs to support their arguments. Strengths of this manuscript are that it is well written, well executed technically, and the entropic repulsion concept interesting to ponder.

However, the entropic repulsion argument is supported by only limited data, and the level of antifouling performance demonstrated by SCLs does not appear to be remarkable compared to existing antifouling technologies.

We are grateful for the insightful analysis of our original submission and for highlighting the necessity for improvements. As described in detail below, the revised manuscript reports several additional experiments that further substantiate the uncovered entropic repulsion mechanism and its relevance for the restriction of bioadhesion.

Specifically, we have strengthened the 'entropic repulsion argument' by:

- Expanding the temperature-dependent protein adsorption data with additional measuring points within the investigated temperature interval (see **Fig. 3a**)

Fig. 3a - Temperature-dependent adsorbed amounts of lysozyme as determined by quartz crystal microbalance measurements. For cholesterol SCLs, a pronounced temperature dependence of lysozyme adsorption is observed. Data were obtained from at least three independent experiments. The mean \pm standard deviations are shown.

- A thorough statistical analysis of the more comprehensive datasets (see **Extended Data Fig. 11 & Extended Data Calculations Part 2** of the revised manuscript).

Extended Data Fig. 11 - The temperature dependence of the Gibbs free energy of adsorption is derived from the amount of adsorbed lysozyme in quartz crystal microbalance measurements (*Fig. 3a*). Data points were calculated with $\Delta G = -RT \times \ln(q/(c^* - c^*q))$, where R is the universal gas constant, T the absolute temperature, q the adsorbed surface fraction, and c^* the normalized protein concentration ($c^* = 6.8 \cdot 10^{-6}$). Surface fractions were calculated as the ratio of the experimentally determined amount to the maximum amount of adsorbed lysozyme (see *Extended Data Calculations Part 2*). The coefficients of determination with (R^2) for the individual fits are given in the figure.

- Demonstration of the general validity of the uncovered anti-adhesive effect by temperature-dependent adsorption data of two additional globular proteins – lysozyme

and fibrinogen (see **Fig. 3a** & the newly inserted **Extended Data Fig. 10** of the revised manuscript)

Extended Data Fig. 10 - The temperature dependence of protein adsorption on cholesterol SCLs. Temperature-dependent adsorbed amounts of (a) bovine serum albumin and (b) fibrinogen as determined by quartz crystal microbalance measurements. For cholesterol SCLs, a pronounced temperature dependence of protein adsorption is observed. Data were obtained from at least three independent experiments. The mean \pm standard deviations are shown.

- Newly performed atomistic simulations that clearly show molecular orientational fluctuations at the interface of cholesterol multilayers, strongly supporting orientational fluctuations to be the key process causing interfacial entropic repulsion (**Fig. 3b-e and Extended Data Fig. 12**)

Fig. 3 b) MD simulations of cholesterol multilayers in contact with water with equilibrium state (left) and intentionally reverted molecular orientation of 10% of molecules in interface layer (right) (for details, see **Extended Data Calculations Part 4**). Oxygen atoms of the hydroxyl groups are shown as orange spheres. **(c)** Angles between the vector connecting the C3 and C17 atoms on cholesterol and stigmastrol (**Extended Data Fig. 1c** and **Extended Data Calculations Part 4**) and the z-axis for interfacial molecules as a function of simulation time for one representative simulation trajectory. In cholesterol multilayer systems with 10% of reverted molecules spontaneous back-reorientation within 1 μs of simulation time is observed. No back-orientation is observed for stigmastrol controls. **(d)** Correlation function (C_r) in dependence on the distance to a reference molecule, plotted for three simulation trajectories of cholesterol multilayer systems (without changed orientation of molecules at the interface). Error bars represent the standard deviation at that point over the time intervals. **(e)** RMSD (for cholesterol multilayers and stigmastrol controls) over three simulation trajectories of z-component of the position of all O-atoms at the interface layer as a function of time for frames every 0.01 μs with error bars stating the standard error of the mean.

Extended Data Fig. 12 Molecular dynamics simulation of SCLs in contact with water. **(a)** The left panel shows the simulation box for a system with four layers of cholesterol or stigmastrol in contact with a water layer and vacuum above and below. Hydrophobic walls were at $z = 0$ and $z = 50$ nm. The right panel shows the lipid multilayer – water system within the simulation box. For clarity, salt ions

within the water layer are not shown. Oxygen atoms on the hydrophilic hydroxyl group of cholesterol or stigmaterol are shown as spheres and hydrophobic tails as blue sticks. **(b)** Simulation snapshots at 0 and 1 μ s from a molecular dynamics (MD) trajectory for cholesterol and stigmaterol multilayer systems in contact with water with equilibrium states (0% rotated molecules) and intentionally reverted molecular orientation of 10, 30 and 50 % of molecules in the interface layer. At 0 μ s the production simulations were started after performing multiple equilibration steps (for details see *Methods & Extended Data Calculation Part 4*). **(c)** Visual representation of the vector connecting the C3 and C17 atoms on cholesterol or stigmaterol. This vector is used as the director for each molecule in the correlation analysis and the calculation of the angle of molecules at the interface layer (relative to the z-axis). **(d)** Angles between the vector connecting the C3 and C17 atoms and the z-axis from one representative simulation trajectory for lipid multilayer systems with 10, 30, and 50% reverted molecules. Cholesterol multilayer systems show spontaneous back-reorientation within 1 μ s of simulation time, whereas no back-orientation is observed for stigmaterol controls. **(e)** Correlation function (C_r) as a function of the distance to a reference molecule, plotted for three simulation trajectories of stigmaterol multilayer systems (without changed orientation of molecules at the interface), respectively. Error bars at each point represent the standard deviation at that point over the time intervals. **(f)** Histogram of mean RMSD over three simulation trajectories of z-component of position of all oxygen atoms at the interface layer in the time range 0.5 to 1 μ s shows higher fluctuation of the interface for cholesterol containing multilayer system.

While the described antiadhesive effect of cholesterol SCLs proves to be strong, we are by no means claiming that our cholesterol-based model systems developed to investigate the hitherto unknown phenomenon would immediately outcompete existing antifouling technologies. As stated in the conclusions of the revised manuscript, further work will be needed to enable the technological translation of the obtained findings. For that aim, we envision dynamic assemblies of synthetic amphiphilic structures designed to maximize entropic repulsion as a more effective and robust alternative to cholesterol-containing coatings.

Specific Comments To The Authors:

1. In their analysis of the thermodynamics of the system, the authors have considered primarily the dynamics and orientational changes in the SCL, but have not adequately considered other contributions, such as the dynamics and potential denaturation of protein upon adsorption, and interaction of protein with Chol. These can also make enthalpic and entropic contributions to overall free energy. For example, Pegg et al (Protein & Peptide Letters, 2008, Vol. 15, No. 4) showed that Chol-BSA complexation is enthalpy driven. In my view it is also likely that adsorption of protein to SCL will be accompanied by release of water bound to protein and surface. There should be an entropic component to this. On page 6, the authors suggest there are other entropic effects in protein adsorption but do not elaborate. What are these other effects and is a detailed accounting of them possible?

We appreciate this comment as it points at the necessity to better delineate our analytical approach. The reviewer refers to a publication on the interactions of dissolved cholesterol, whereas our manuscript concerns layered systems that exhibit repulsive characteristics due the fact that cholesterol is assembled. We are fully aware that the effects described by Peng et al. could nonetheless contribute to the interactions of cholesterol SCLs with proteins. Therefore, the entropic repulsion of cholesterol SCLs was derived in our study relative to several controls, including gold surfaces, thiocholesterol SAMs and thiocholenic acid SAMs, and from adsorption experiments with three proteins of very different characteristics. The revised manuscript was extended to clarify this approach.

2. The authors argue in Figure 3 that protein adsorption will lead to an entropic penalty due to locally restricting orientational dynamics of Chol molecules. Is there experimental evidence for this? Was FRAP performed on protein adsorbed SCLs?

We agree that the question for direct experimental evidence of the proposed mechanism is of obvious interest. AFM-based experiments revealed polarity fluctuations of cholesterol SCLs (Fig. 3g, Extended Data Fig. 13 and Extended Data Fig. 14 of the revised manuscript), however, without proving that these fluctuations are constrained by adsorbed proteins. Additionally, our new molecular dynamics simulations independently substantiate molecular orientational fluctuations at the interface of cholesterol SCLs (Fig. 3b-e & Extended Data Fig. 12).

FRAP experiments revealed lateral mobility of cholesterol in SCLs comparable to mobility in smectic liquid crystals (Extended Data Fig. 17 of the revised manuscript) but cannot provide information about orientational fluctuations. Modifying cholesterol regiospecifically with fluorogenic or solvatochromic dyes (i.e. dyes that change fluorescence intensity or colour, respectively, in response to changes in their microenvironment polarity) was considered but not implemented since the fluorophores contain combined aromatic groups, planar or cyclic molecules with π -bonds that would significantly alter the physicochemical properties of cholesterol. Since even minor changes in the molecular structure of cholesterol were shown to have strong effects on the bioadhesion properties of SCLs (Fig. 4 of the revised manuscript), this analytical approach was ultimately not pursued.

Thus, the experimental evidence we can provide concerns the detection of orientational molecular fluctuations of the plain interface of cholesterol-containing SCLs. It is, however, very obvious that any adsorption or adhesion process would interfere with these fluctuations, thereby amounting to an entropic penalty.

3. The practical significance of cholesterol-mimetic synthetic antiadhesive systems is unclear. For example, it is noted that the cholesterol multilayers support roughly 150-400 ng/cm² of protein adsorption depending on the protein (less for BSA, more for serum). This is actually a lot of protein, in fact very large in comparison to better known protein antifouling technologies such as OEG-terminated SAMs and zwitterionic polymers, which allow only a few ng/cm² protein adsorption under similar conditions. The bacterial adhesion performance of SCL is harder to assess, as bacterial adhesion was shown to be significantly reduced but log reduction values were not included (perhaps I missed them). Good bacterial antifouling technologies reduce bacterial adhesion by 3-4 log or more.

As stated above, our reported study aimed at investigating the previously unknown phenomenon of entropic repulsion at cholesterol-containing layers per se. While the effect is shown to be rather strong (in comparison to systems lacking cholesterol or restricting its orientational mobility), we do not claim that the investigated cholesterol-based model systems outcompete existing antifouling technologies.

Indeed, we would like to point out that further work will be needed to enable the technological translation of the obtained findings. For that aim, as stated in the conclusions of the revised manuscript, we envision dynamic assemblies of synthetic amphiphilic/reorientable interfacial structures designed to maximize entropic repulsion as a more effective and robust alternative to cholesterol-containing coatings. We are confident that this will create unprecedented and highly useful new technological options.

4. Although I am not a Collembola expert, what information I could find suggests that it is a terrestrial organism that likes moist environments but apparently does not live under water. This left me wondering about how predictive antifouling experiments conducted fully under water can be to understanding Collembola antifouling.

The reviewer is fully correct that Collembola do not live under water, but they may be exposed to aqueous environments for shorter or longer periods (up to several hours), depending on their habitat. This highly relevant challenged condition for Collembola poses the risks of wetting and microbial colonization. As explained in the introduction of the manuscript, the evolved principles to minimize these processes (which would otherwise result in suffocation of the skin-breathing organisms) include a unique omniphobic nanomorphology of the cuticle – and the characteristics of its topmost molecular layers which provided the starting point of our reported study.

5. Bacterial adhesion was assayed by incubating substrates in bacteria suspended in lysogeny broth (LB). LB is a soup of many things including peptides. Like proteins, peptides presumably adsorb to SCL. Therefore, bacterial adhesion to SCL is likely mediated by preadsorbed peptides from the broth. What does this mean for the interpretations of the authors?

*LB medium was used in bacterial culture experiments to avoid artefacts due to nutrient deficiency. We agree with the reviewer that performing bacterial tests in LB can cause the formation of interfacial “conditioning films” at the substrates before bacteria arrive, thereby mediating the SCL-bacteria adhesion. However, bacterial adhesion to cholesterol-containing SCLs was found to be effectively prevented despite the presence of pre-adsorbed peptides from the medium. Additional bacterial adhesion experiments in PBS gave results comparable to the experiments in LB medium (**Response Letter Figure 2**). Thus, the used experimental conditions demonstrate the capacity of the uncovered repulsive mechanism to counteract bacterial adhesion even at more demanding and realistic settings.*

Response Letter Figure 2: Bacterial adhesion on Collembola cuticular SCLs. (a,b) Normalized adherent cell density of (a) *S. epidermidis* and (b) *E. coli*. Data are normalized to the average adherent cell density on the SiO₂ substrate. In all graphs, the mean + standard deviation values are shown. Data were obtained from at least three independent experiments.

6. In the bacterial adhesion assay, the methods describe fixing and then imaging by SEM. This seems very unusual for counting of adhered bacteria. Not enough information is provided on certain aspects (e.g. # cells counted). Raw numbers of adhered bacteria should be reported.

We agree that determining the number of adherent bacteria using SEM images is indeed not as common as using fluorescence microscopy. This more laborious method was primarily used to detect even weakly attached bacteria (at non-adhesive/repulsive surfaces) which are likely to be removed upon more harsh washing steps. The SEM analysis furthermore provides information about the quality of the substrate and the shape of bacterial cells (displaying their state).

*Due to the variation in the absolute number of adherent cells between experiments, we used a normalization approach. In each experiment, the number of adherent bacteria was normalized to the control (SiO₂ or gold substrate, depending on the type of surface analyzed). At least 6 images were taken per experiment per sample and the bacteria were counted with ImageJ. For control samples, this results in an average number of about 7,000 and 4,000 cells per mm² for *E. coli* on SiO₂ and Au, and 60,000 and 30,000 cells per mm² for *S. epidermidis* on SiO₂ and Au, respectively. In the revised manuscript, the determination of the data was described in more detail as follows:*

“The scale bars of the SEM images were used to calibrate the pixel width. The Fiji 'cell counter' plugin was used to count the number of cells in the calculated area ($\sim 10^3 \mu\text{m}^2$). The counted numbers were either normalised against the median value of the silicon reference for relative comparisons or scaled up to cells per mm^2 for absolute values.”

Furthermore, we are providing with this response letter a tabular list of the raw data of all bacterial experiments conducted.

7. In the single cell force spectroscopy experiment, it is unclear if the authors confirmed that the presence and orientation of the cell is unchanged after the contact experiment. Was the cell intact? It seems that the orientation and location of the cell on the AFM cantilever should affect contact area and therefore interaction force with the SCL surface. Did the authors measure this or account for differences in orientation/contact area of cells? The reference cited for the method (#33) has little to say about this.

We are grateful for the advice to better explain the applied methodology of the SCFS experiments. We used fluorescent, GFP-expressing E.coli cells which not only facilitates the initial attachment of the bacterial cell to the AFM cantilever, but also allows for monitoring the position/orientation of the cell during the adhesion measurement. Moreover, the fluorescence signal can be used to trace viability as it indicates whether the cell is intact - lysed cells lose the cytoplasmic GFP. Only data sets of viable cells of appropriate position/alignment on the cantilever during the measurement were used. We have extended this information in the Materials and Methods section of the revised manuscript to better explain the applied methodology:

“Only data sets where the position and orientation of the bacterial cell on the AFM cantilever was unchanged before and after the measurements (i.e., the contact conditions/geometry was constant during the measurement) were analysed.”

Referee #3 (Remarks to the Author):

In this manuscript, Jens Friedrichs et al reported that entropic repulsion caused by dynamic interfacial fluctuations of cholesterol layers strongly restricts protein adsorption and bacterial adhesion. The adsorption of proteins is of high physiological relevance, and as such they adsorb with different mechanisms than their molecular or atomic analogs. Compare to current studies in the field of bio-adhesions, I am delighted to see a small biomolecule could show special functions in this way. How cholesterol influence the bio-adhesion process revealed from this work may

inspire researchers proposing new strategies for designing antifouling molecules in the future. The authors claimed that the entropic repulsion mechanism was previously unknown. However, there have been many studies of this mechanism, which have been reported to be useful in anti-adhesion (eg. P. G. D. Gennes et al. *J. Colloid Interface Sci.* 1991, 142, 149-158; Julie M. Goddard et al. *J. Agric. Food Chem.* 2012, 60, 2943–2957; Dong Keun Han et al. *Progress in Polymer Science* 2015, 44, 28–61). In particular, as early as 1991, the P. G. D. Gennes proposed that the approach of the protein initiates the compression of PEO chains, which induces a decrease in entropy and a steric repulsion effect. Therefore, the authors claimed the entropic repulsion mechanism is fully novel, which is not justified.

We are pleased about the reviewer's confirmation of the relevance of our report. Also, we fully agree that 'entropic repulsion' phenomena have been discussed for other adhesion-reducing surfaces before. We regret that our original submission was not clear enough and made the reviewer think we would claim entropic repulsion per se to be an entirely novel idea. This is not our intention and we have revised the manuscript to clarify this point.

However, the classical work mentioned by the reviewer concerns distinctly different phenomena and structures: Entropic repulsion effects caused by the compression of surface-bound oligomer chains are in fact well known, while our current report of entropic repulsion due to orientational fluctuations of amphiphilic interfacial components is – to the best of our knowledge – entirely new and has never been considered before. We understand that a better demarcation is required and have clarified this point in the revised manuscript, including additional references to the classical work by de Gennes⁴.

Specifically, we have modified the introduction as follows:

*“Cholesterol-containing lipid layers were discovered to counteract bioadhesion by a previously unknown entropic repulsion mechanism, **which is - unlike earlier reported interfacial effects**³⁻⁵- **is caused by orientational fluctuations.**”*

The earlier reports are further specified in the results section:

“Entropic repulsion effects were previously only reported to occur due to pinned conformational flexibility of grafted oligo(ethylene glycol)³ and poly(ethylene glycol) brushes⁴ or enforced smoothening of height fluctuations (capillary waves) of lipid bilayers”

Besides, the authors proposed that spatially constrained cholesterol SAMs showed less anti-adhesive effects, which is similar to the previous report (see S. Jiang et al. *J. Phys. Chem. B* 2005, 109, 2934-2941). Jiang have proposed that proteins adsorb more on the densely packed SAMs of EG2OH and EG4OH, while EG6OH-SAMs generally resist protein adsorption due to lower packing

density. In addition, self-assembled molecular surfaces that are resistant to protein adsorption have also been extensively reported. For example, in 1991, G. M. Whitesides et al. have reported the model systems of self-assembled organic monolayers for studying adsorption of proteins at surfaces (Science 1991, 252: 1164-1167; J. Am. Chem. Soc. 1993, 115: 10714-10721).

We are grateful for the reviewer's advice to consider potential similarities to other anti-adhesive systems. We agree and are aware that the mentioned studies impressively showed how the packing density of oligo(ethylene glycol) SAMs correlates with their adhesive properties, suggesting that molecular flexibility is relevant for repulsive characteristics. However, the related structures – alkyl oligo(ethylene glycol) – are very different from cholesterol SCLs and so is the origin of interfacial dynamics that is restricted upon adsorption/adhesion of proteins and cells. In particular, the thiocholesterol and thiocholenic acid SAMs used in our study prevent orientational fluctuations of the layered cholesterol derivatives, which are, however present in cholesterol-containing SCLs – a feature that is absent in all types of SAMs carrying grafted oligo(ethylene glycol) units. In the revised manuscript, we have explained this important distinction and included a reference to the mentioned work:

“Entropic repulsion effects were previously reported to occur on SAMs (e.g. OEGs³), polymer brushes (e.g. PEO⁴) and lipid bilayers due to pinned molecular/conformational flexibility of OEG SAMs and PEO brushes and enforced smoothening of height fluctuations (capillary waves) at lipid bilayers, respectively.”

In my opinion, I would suggest the authors to prove if there are any advantages of the entropic repulsion mechanism for small molecular as cholesterol than traditional ones.

We appreciate this suggestion and agree that repulsive effects caused by orientational fluctuations of non-covalently assembled small molecules can be of significant interest. However, our current report concerns the identification of this entropic repulsion mechanism of cholesterol-containing layers per se. As stated above, a realistic classification/‘benchmarking’ of the repulsive performance of the uncovered principle will require further technological translation which clearly goes beyond the current study. We have nonetheless extended the manuscript to further comment on the obvious differences of our uncovered repulsive mechanisms and its structural prerequisites/features in comparison to previously known repulsive surface structures.

1) Fig. 3a is a key experimental evidence used to discuss and quantification of entropic repulsion by cholesterol SCLs. However, I see a few questions in this figure. First of all, the slope of cholesterol SCLs between the temperature of 20~30 Celsius seemed unchanged. So, the reliability of this slope calculated with other two temperature points is deeply doubtful. Second, this experiment only tested 4 temperature points. More temperature points are needed (similar issues can also be found in Fig 2 a&b). Third, only one protein (bovine serum albumin) was tested. More proteins are need to be tested to confirm whether this result is universal.

We agree that the data and analysis presented in Figure 3a of our original submission had to be improved and extended, and the manuscript was revised accordingly (see our answer to the comments of reviewer #1 above, the answer is identical):

To improve the precision of this evaluation, a higher number of measurement points (temperatures) within the accessible range was recorded in all protein adsorption experiments (see **Fig. 3a & Extended Data Fig. 10** of the revised manuscript).

Fig. 3a - Temperature-dependent adsorbed amounts of lysozyme as determined by quartz crystal microbalance measurements. For cholesterol SCLs, a pronounced temperature dependence of lysozyme adsorption is observed. Data were obtained from at least three independent experiments. The mean \pm standard deviations are shown.

The more comprehensive data sets were subjected to a thorough statistical analysis (see **Extended Data Fig. 11 & Extended Data Calculations Part 2** of the revised manuscript), confirming validity and significance of the evaluation.

Extended Data Fig. 11 - The temperature dependence of the Gibbs free energy of adsorption is derived from the amount of adsorbed lysozyme in quartz crystal microbalance measurements (Fig. 3a). Data points were calculated with $\Delta G = -RT \times \ln(q/(c^* - c^*q))$, where R is the universal gas constant, T the absolute temperature, q the adsorbed surface fraction, and c^* the normalized protein concentration ($c^* = 6.8 \cdot 10^{-6}$). Surface fractions were calculated as the ratio of the experimentally determined amount to the maximum amount of adsorbed lysozyme (see Extended Data Calculations Part 2). The coefficients of determination (R^2) for the individual fits are given in the figure.

Furthermore, to demonstrate the generality of the uncovered anti-adhesive effect, temperature-dependent adsorption data of two additional globular proteins – lysozyme and fibrinogen – were collected and evaluated (see **Figure 3a** & **Extended Data Fig. 10** of the revised manuscript). Lysozyme and fibrinogen displayed entropic repulsion effects similar to serum albumin when adsorbed at cholesterol SCLs, indicating the significant entropic barrier counteracting bioadhesion at cholesterol SCLs to be a ubiquitous phenomenon.

Extended Data Fig. 10 - The temperature dependence of protein adsorption on cholesterol SCLs. Temperature-dependent adsorbed amounts of (a) bovine serum albumin and (b) fibrinogen as determined by quartz crystal microbalance measurements. For cholesterol SCLs, a pronounced temperature dependence of protein adsorption is observed. Data were obtained from at least three independent experiments. The mean \pm standard deviations are shown.

2) Detailed structures inside cholesterol SCLs have not been shown, SAXS experiments may reveal more underlying facts.

We would like to point out that we report ATR-FTIR data of cholesterol SCLs (**Extended Data Fig. 4** of the revised manuscript) that indicate highly ordered cholesterol molecules, a well-known feature of cholesterol layers.^{6,7} From the layer thickness data (**Extended Data Fig. 5a** of the revised manuscript) it is evident that cholesterol SCLs are composed of multiple molecular layers. Considering the high advancing contact angle of cholesterol SCLs (**Extended Data Fig. 9a** of the revised manuscript), it can be deduced that the hydrocarbon tail of the outer cholesterol layer is initially oriented towards the interface, while the OH groups are oriented towards the interior. Based on that, a multilamellar layer of stacked bilayers in which hydrophilic OH groups point to each other can be reasonably assumed. The formation of these highly ordered and oriented cholesterol layers occurs through self-assembly during spin coating. We have clarified the related text in the revised manuscript to better highlight these features of the investigated layers:

“Dynamic wetting experiments on cholesterol SCLs in air displayed similarly high advancing and receding water contact angles when the droplet was immediately withdrawn after deposition (no resting period) (Fig. 2c, Extended Data Fig. 9a, and Supplementary Movie 1), indicating that the hydrocarbon tail of the outer cholesterol layer is initially oriented toward the interface, while the OH groups are oriented inward.”

In addition, the caption of Fig.1e was adapted:

ATR-FTIR (Extended Data Fig. 4) and dynamic contact angle measurements (Fig. 2c and Extended Data Fig. 9a) indicate highly ordered cholesterol molecules, with the hydrocarbon tail of the outer cholesterol layer initially oriented towards the interface, while the OH groups are oriented inward.

3) The CA experiments showed that after contacting with water, a switch in the polar orientation of cholesterol have taken place in no more than 20 seconds. Though cholesterol showed slower switching speed than phytosterol and other cholesterol analogs, but still showed strong interactions with water molecular. It is known that the displacement of water from the surface of a hydrophilic material represents a large energy barrier to adsorption by proteins, would this be the case?

We fully agree that the displacement of the hydration shell from hydrophilic surfaces is a well-known barrier to protein adsorption and deserves consideration. In line with this, our data show for all amphiphilic steroids lower levels of bioadhesion. However, our reported analysis can clearly distinguish the newly uncovered entropic repulsion from such anti-adhesive effects since we were using appropriate hydrophilic but non-dynamic interfaces as controls (Au, thiocholenic acid SAMs, thiocholesterol SAMs) in the respective protein adsorption experiments: The anti-adhesive characteristics of cholesterol SCLs were analyzed relative to these controls.

4) Phytosterol showed no resting time-dependent decrease of the receding contact angle, but also possessed impressive anti-bioadhesion behaviours as shown in Extended Data Fig. 14 for S. epidermidis, bovine serum albumin and lysozyme. How to understand this result, and it is also related to with entropic repulsion effect?

Since molecular amphiphilicity and interfacial orientational mobility were identified to be prerequisites of the uncovered entropic repulsion effect of SCLs, which are given for stigmasterol (phytosterol) layers (Fig. 4 and Extended Data Fig. 15d of the revised manuscript), we can reasonably assume that the repulsive effect contributes to the characteristics of these layers as well. Nonetheless, the correlation

of the reorientation dynamics (absence of resting time-dependent decrease of the receding contact angle) and the bioadhesion data is obvious from the holistic comparison: While stigmasterol SCLs in fact perform similar to cholesterol SCLs in adhesion experiments with *S. epidermidis* and nearly similar in lysozyme and serum albumin adsorption experiments, these layers perform substantially less in all other bioadhesion tests.

We would like to point out that our new molecular dynamics simulations (**Fig. 3b-c and Extended Data Fig. 12** of the revised manuscript) clearly indicate a difference between cholesterol and stigmasterol layers with weaker interfacial surface fluctuations and no spontaneous molecular reorientations of the latter. These features of the stigmasterol layers can be expected to increase bioadhesion.

Hence, while entropic repulsion may influence the bioadhesion characteristics of stigmasterol layers as well, the effect is much weaker than for cholesterol layers as impressively shown experimentally for fetal serum adsorption and *E. coli* adhesion.

5) In practical environments, the stability of membrane is essential. The authors tried to prove the stability of cholesterol SCLs when measured with TOC after incubation in pure water for 24h. Since water molecules have strong influence on the conformation of cholesterol molecules, it would not be surprising to see the cholesterol SCLs fallen apart in a longer period. A stability test with 7 days and different temperatures is recommended.

We agree that layer stability is of particular importance and performed the proposed experiments: Cholesterol SCLs were shown to be stable even after an incubation period of 7 days at elevated temperatures (40°C). The corresponding data can be found in the **newly inserted Extended Data Fig. 6** of the revised manuscript.

Extended Data Fig. 2 Stability of cholesterol SCLs in Milli-Q water. The total organic carbon (TOC) content of Milli-Q water was measured after incubation of cholesterol SCLs for one (1 d) and seven days (7 d). Pure Milli-Q water and Milli-Q water incubated with cleaned SiO₂ substrates served as negative controls. An saturated aqueous cholesterol solution (positive control), was prepared by dissolving 2 mg/ml cholesterol in Methanol. The dissolved cholesterol was precipitated by adding Milli-Q water, the methanol was boiled out, the solution was passed through a sterile filter and the filtered solution was used for TOC measurements. For both time points, the TOC content of the Milli-Q water incubated with cholesterol SCLs was not significantly different from the corresponding control (i.e. cleaned SiO₂ substrate incubated with Milli-Q water). The time-dependent systematic increase of the measured values can be attributed to unavoidable contamination during storage of the samples. The mean and + standard deviation are shown. Data were obtained from at least three independent experiments. ns ($p > 0.05$), unpaired t-test.

Though the authors gave a large amount of description and data to illustrate this hypothesis, I cannot agree on many aspects. Thus, I cannot recommend its publication on Nature currently.

We very much appreciate the detailed comments, criticisms and suggestions of the reviewer and have conducted several additional experiments, analyses and simulations to address them in full.

Referee #4 (Remarks to the Author):

The manuscript is interesting and touches theoretical and practical relevant topics. I believe that if the authors revise the manuscript properly (major revisions), the article could be published. Nature Physics could also be an option.

We are grateful for the comment of the reviewer and hope our revision improved the manuscript such that he finds publication warranted.

References:

1. Ulman, A. Formation and Structure of Self-Assembled Monolayers. *Chem. Rev.* **96**, 1533–1554 (1996).
2. Vericat, C., Vela, M. E., Benitez, G., Carro, P. & Salvarezza, R. C. Self-assembled monolayers of thiols and dithiols on gold: new challenges for a well-known system. *Chem. Soc. Rev.* **39**, 1805–1834 (2010).
3. Li, L., Chen, S., Zheng, J., Ratner, B. D. & Jiang, S. Protein adsorption on oligo(ethylene glycol)-terminated alkanethiolate self-assembled monolayers: The molecular basis for nonfouling behavior. *J. Phys. Chem. B* **109**, 2934–2941 (2005).
4. Jeon, S. I., Lee, J. H., Andrade, J. D. & De Gennes, P. G. Protein—surface interactions in the presence of polyethylene oxide: I. Simplified theory. *J. Colloid Interface Sci.* **142**, 149–158 (1991).
5. Lipowsky, R. Generic interactions of flexible membranes. in *Handbook of Biological Physics* (eds. Lipowsky, R. & Sackmann, E. B. T.-H. of B. P.) vol. 1 521–602 (North-Holland, 1995).
6. Loomis, C. R., Shipley, G. G. & Small, D. M. The phase behavior of hydrated cholesterol. *J. Lipid Res.* **20**, 525–535 (1979).
7. Shieh, H. S., Hoard, L. G. & Nordman, C. E. Crystal structure of anhydrous cholesterol [39]. *Nature* vol. 267 287–289 (1977).

Reviewer Reports on the First Revision:

Reviewer #4 full comments:

Review:

The manuscript is overall interesting and touches on a theoretically and practically relevant topic. I believe that if the authors revise the manuscript properly (major revisions), the article could be published. Nature Physics could also be an option.

General remarks:

1. Description of key achievements of the paper. Please let us know in a paragraph or two the advance presented in the paper, and your opinion of the potential excitement of the findings for your community more generally. Will the work open up completely new research directions or technological opportunities? Are there any outstanding features that deserve a special mention?

Answer: The work is very interesting as it demonstrates the importance of interface control at the molecular and nanometric scale to control bio-adhesion processes. The composition of the interfaces controls the subsequent behaviour of the system. The most natural interfaces (as part of membranes) are made of phospholipids, cholesterol and membrane proteins (and carbohydrates). Here the authors focus on (different types of) lipid multilayers (prepared with spin coating) and self-assembled monolayers, as a system to compare. The authors study many systems (with appropriate experimental techniques) to conclude that entropic repulsion is the main reason for the non-adhesion of proteins and bacteria. This has to do with the type of cholesterol (structure) and its degree of freedom in the interface.

The work is important for the community working on biomimetic surfaces, lipid layers, adhesion and surface interaction forces. This research work, if published in Nature, could open the debate of the importance of cholesterol on model membrane systems. Bio-adhesion is a topic that is important on its own (i.e. Lotus effect). Many groups are investigating in this direction (e.g. superhydrophobic interfaces) but this study tries to bring some molecular light in the understanding of the process. The authors have made an effort in providing a molecular explanation. Of course, one can say that they focused on the lipid components of the *Collembola* cuticula, but the results, if correct measured, should represent an "universal" approach to explain adhesion phenomena for these systems.

2. Novelty. If the conclusions are not original, it would be very helpful if you could provide relevant references, authors invariably question criticisms of novelty if there are no supporting references.

Answer: The findings are new and interesting. The authors refer to the entropic repulsion that already appears in textbooks on colloidal stability. However, in my view, the notion of entropic repulsion here refers to the mobility and structure of lipid structures (i.e. cholesterol).

A weak point could be the use of references. Many of them are "classical" providing a theoretical framework. However the youngest reference is from 2016 (if I am not mistaken). I think that references concerning the role of cholesterol in, for example, protein adsorption are missing. References related to residence time and adhesion are also missing, especially those discussing specific vs. non-specific forces. In general, the authors also should "explain" the

importance of their work in other fields, and provide references accordingly.

3. Technical robustness. Does the manuscript have technical flaws which should prohibit its publication? If so, please provide details. Are these flaws resolvable with minor / major revisions?

Answer: The manuscript provides a considerable amount of data. The authors utilize complementary experimental techniques, which are well-established in the scientific community (e.g. quartz crystal microbalance, ellipsometry, FRAP, FTIR, wetting measurements, atomic force microscopy...). My perception is that it is technically well done, although the authors should clarify some issues regarding the experimental data. The statistics look correct; however, I would like to see the fitting of the third order kinetics in the extended part. All the fitting procedures should be presented in detail.

Other comments:

(i) The authors should write more generally why adhesion is important. They should connect other scientific and technological aspects of their work.

(ii) Page 4, "The resulting interaction forces...are found to increase continuously with contact time." This sentence might imply that there is not a maximum value for the force, since you can increase the contact time as desire. Please, comment.

(iii) I guess, if the authors could analyse the AFM images to try to distinguish topographic patterns (e.g. extended Fig 7).

(iv) The authors should show a phase diagram of the behaviour of lipid systems. How does their structure change when heated?

(v) In QCM-D experiments the crystal baseline can change with temperature. That is, in one cycle it could have hysteresis. The authors should show such control to see if this could influence the final outcome.

(vi) It would be interesting if the authors using ellipsometry and QCM-D measurements could distinguish between wet and dry mass. This would represent an improvement on the experimental data.

(vii) With regard to figure 12 (extended information). I wonder what topographical error the AFM is making. For a 20 nm apex, what is the error that can be made? Does the tip always scan the same position? Why do you choose a time of 20 seconds and not 10 to 30? Please explain.

(viii) The authors should consider the contact area when measuring the interaction from particles and bacteria. The axis could be renamed as (force/radius). Furthermore, such contact area could be used to discussed FRAP results.

(ix) Would it make sense to do contact angle measurements as a function of the temperature? Please, comment.

Author Rebuttals to First Revision:

Referee #4 (Remarks to the Author):

The manuscript is overall interesting and touches on a theoretically and practically relevant topic. I believe that if the authors revise the manuscript properly (major revisions), the article could be published. Nature Physics could also be an option.

We very much appreciate the encouraging evaluation by the reviewer and hope to have adequately addressed his/her specific comments and suggestions.

We would like to point out that the revised manuscript (as explained in detail above) provides comprehensive temperature-dependent protein adsorption data for three globular plasma proteins of different size, characteristics and high biomedical relevance, thoroughly confirming the proposed entropic repulsion effect and demonstrating its general validity. Moreover, we

performed molecular dynamics simulations that display molecular orientational fluctuations at the interface of cholesterol multilayers and unravel how they are caused by the specific molecular characteristics and interactions of cholesterol. These new findings independently support the view that orientational fluctuations are key to entropic repulsion at the interface of the investigated systems, which strongly confirms the results of our experimental analyses.

General remarks:

1. Description of key achievements of the paper. Please let us know in a paragraph or two the advance presented in the paper, and your opinion of the potential excitement of the findings for your community more generally. Will the work open up completely new research directions or technological opportunities? Are there any outstanding features that deserve a special mention?

Answer: The work is very interesting as it demonstrates the importance of interface control at the molecular and nanometric scale to control bio-adhesion processes. The composition of the interfaces controls the subsequent behaviour of the system. The most natural interfaces (as part of membranes) are made of phospholipids, cholesterol and membrane proteins (and carbohydrates). Here the authors focus on (different types of) lipid multilayers (prepared with spin coating) and self-assembled monolayers, as a system to compare. The authors study many systems (with appropriate experimental techniques) to conclude that entropic repulsion is the main reason for the non-adhesion of proteins and bacteria. This has to do with the type of cholesterol (structure) and its degree of freedom in the interface.

The work is important for the community working on biomimetic surfaces, lipid layers, adhesion and surface interaction forces. This research work, if published in Nature, could open the debate of the importance of cholesterol on model membrane systems. Bio-adhesion is a topic that is important on its own (i.e. Lotus effect). Many groups are investigating in this direction (e.g. superhydrophobic interfaces) but this study tries to bring some molecular light in the understanding of the process. The authors have made an effort in providing a molecular explanation. Of course, one can say that they focused on the lipid components of the *Collembola* cuticula, but the results, if correct measured, should represent an “universal” approach to explain adhesion phenomena for these systems.

We are particularly grateful for these insightful comments on the relevance and the proposed framing of our study. In fact, we primarily expect our report on the interplay of molecular features, assembly and interfacial bioadhesion to initiate further discussion on the conditions of effective entropic repulsion in living matter and in engineered systems. In the latter context, the newly proposed mechanism of entropic repulsion induced by molecular orientational fluctuations may guide the design of powerful and robust anti-bioadhesive structures. At the same time, we fully agree with the reviewer about the potential importance of our findings for future research on cholesterol-containing lipid membranes. Our data unravel the anti-bioadhesive characteristics of unconstrained cholesterol assemblies at the interface of multilayers and highlight the relevance of orientational freedom as a key prerequisite. Investigating whether and how cholesterol assemblies in lipid membranes can meet this criterion and display similar effects will be a highly promising future research avenue. To point out this perspective, we have extended the concluding paragraph of the revised manuscript (ln. 341-350):

“Exploring the compositional ‘line of defense’ of the anti-adhesive Collembola cuticula, we identified previously unknown physicochemical features of cholesterol-containing layers that result in effective entropic repulsion. Given the widespread occurrence of cholesterol at various different types of tissue boundaries, our findings may shed new light on fundamentally important and ubiquitous interfacial processes of living matter. In particular, investigating whether cholesterol assemblies in lipid membranes can display fluctuating reorientations that entropically control bioadhesion will be a promising future research avenue. Beyond that, the development of cholesterol-inspired synthetic and scalable materials and systems to control bioadhesion is highly attractive but demands further analyses of the molecular requirements that govern the newly identified mechanism.”

2. Novelty. If the conclusions are not original, it would be very helpful if you could provide relevant references, authors invariably question criticisms of novelty if there are no supporting references.

Answer: The findings are new and interesting. The authors refer to the entropic repulsion that already appears in textbooks on colloidal stability. However, in my view, the notion of entropic repulsion here refers to the mobility and structure of lipid structures (i.e. cholesterol).

We are grateful for the confirmation of novelty and interest, and we fully agree that the phenomenon of entropic repulsion has previously been discussed in a different context. As noted by the reviewer, entropic repulsion due to molecular orientational fluctuations of amphiphilic interfacial components has never been considered before to the best of our knowledge. To clarify this point, we have modified the text of the revised manuscript and added references to earlier studies on other types of entropic repulsion (ln. 253-256):

“Entropic repulsion effects were previously reported to occur on SAMs (e.g. OEGs²⁹), polymer brushes (e.g. PEO¹⁶) and lipid bilayers due to pinned molecular/conformational flexibility of OEG SAMs and PEO brushes and enforced smoothing of height fluctuations (capillary waves) at lipid bilayers, respectively.”

A weak point could be the use of references. Many of them are “classical” providing a theoretical framework. However the youngest reference is from 2016 (if I am not mistaken). I think that references concerning the role of cholesterol in, for example, protein adsorption are missing. References related to residence time and adhesion are also might missing, especially those discussing specific vs. non-specific forces. In general, the authors also should “explain” the importance of their work in other fields, and provide references accordingly.

We are grateful for the request of scrutinizing the cited references.

In fact, the "classical" principles of entropic repulsion date back to the cited publications.

However, we fully agree that earlier studies on the impact of cholesterol on protein adsorption would be certainly of interest to the reader and were not yet adequately covered in our original

submission. We have therefore added several related references (admittedly not all more recent than 2016 either) to earlier findings obtained for entirely different cholesterol-containing multicomponent interfaces; highlighting that they concordantly report on reduced bioadhesion characteristics in presence of cholesterol (ln. 119-126):

“These findings are in line with earlier reports on reduced bioadhesion at entirely different cholesterol-containing interfaces: Multicomponent lipid nanoparticles were reported to display reduced non-specific adsorption of certain plasma proteins – enabling the enhanced specific binding of immunoglobulins and complement proteins – when containing cholesterol^{21,22}. Moreover, depleting cholesterol in cell membranes was found to increase the adhesion energy between the membrane and the cytoskeleton while increasing cellular cholesterol levels had the opposite effect⁸. While these effects are obvious and functionally relevant, their origin has remained unclear.”

We are similarly grateful to the reviewer for making us aware of the relevance of residence time vs. bioadhesion analyses which can in fact extend the interpretation of our reported data, highlighting the efficacy of the proposed mechanism. We have extended the text accordingly and cited related key studies (ln. 143-146):

“Further analysis showed that a third-order kinetics controlled the underlying polarity adaptation process of the SCL interface to maximize hydrophobic interactions, which clearly differs from contact time dependencies observed in classical protein adsorption studies on surface-engineered biomaterials²⁴”

As stated above, we fully share the view that our findings are of importance clearly beyond the specific investigated system. In the revised manuscript, we have therefore expanded the text to better explain this in the light of recent reports displaying the key role of controlled bioadhesion (ln. 343-350):

“Given the widespread occurrence of cholesterol at various different types of tissue boundaries, our findings may shed new light on fundamentally important and ubiquitous interfacial processes of living matter. In particular, investigating whether cholesterol assemblies in lipid membranes can display fluctuating reorientations that entropically control bioadhesion will be a promising future research avenue. Beyond that, the development of cholesterol-inspired synthetic and scalable materials and systems to control bioadhesion is highly attractive but demands further analyses of the molecular requirements that govern the newly identified mechanism.”

3. Technical robustness. Does the manuscript have technical flaws which should prohibit its publication? If so, please provide details. Are these flaws resolvable with minor / major revisions?

Answer: The manuscript provides a considerable amount of data. The authors utilize complementary experimental techniques, which are well-established in the scientific community (e.g. quartz crystal microbalance, ellipsometry, FRAP, FTIR, wetting measurements, atomic force microscopy...). My perception is that it is technically well done, although the authors should clarify some issues regarding the experimental data.

The statistics look correct; however, I would like to see the fitting of the third order kinetics in the extended part. All the fitting procedures should be presented in detail.

We appreciate this very helpful comment and address the request in the revised manuscript by providing detailed descriptions of the fitting procedures (equations, methods) including the presentation of all relevant parameters (regression coefficients). These are now summarized in **Fig.2a, b** and in the **Extended Data Calculations Part 1 and 2** of the revised manuscript (ln. 958-1017).

Fig. 1 Cholesterol SCLs dynamically adapt to the polarity of their environment. (a, b) Contact time-dependent interaction forces (determined by AFM-based force spectroscopy) between (a) a hydrophobic colloidal probe ($\varnothing 10 \mu\text{m}$ silica bead modified with a hydrophobic silane) or (b) individual *Escherichia coli* cells (attached to a $\varnothing 10 \mu\text{m}$ silica bead) and cholesterol SCLs or thiocholesterol and thiocholonic acid SAMs. Experiments were performed at 37°C . In graphs (a) and (b), the mean \pm standard error of the mean are shown. Data were obtained from at least three independent experiments. The shown regressions are the best fit solutions for a third-order process indicating formation of bonds at the probe/cell-to-SCL interface following $\frac{d\sigma}{dt} = -k\sigma^3$ (for details see **Extended Data Calculations Part 1**). The interaction force was assumed to be directly correlated to the surface concentration of formed bonds σ . The coefficients of determination (R^2) for the fits are given in the figure.

„Part 1. Analysis of the time and temperature dependence of interaction forces between the lipid layer and the colloid particles or single bacterial cells

Different kinetic models were applied to analyze the AFM-based force spectroscopy data for a quantification of the interactions between cholesterol SCLs and colloidal probes or individual bacterial cells. The interaction force was assumed to be directly correlated to the surface concentration of the formed bonds σ . While first-order and second-order kinetics do not match the observed time dependence, third-order kinetics ($\frac{d\sigma}{dt} = -3k\sigma^3$) fit the data with a similar time constant, k , for both colloid particle and bacterial cell adhesion. Data were fitted with $F(t) = A \left(1 - \frac{1}{\sqrt{1+6kt}} \right) + B$ to account for limiting values of non-mobile SAM controls, ($F(t=0) = B$, $F(t \rightarrow \infty) = A + B$). The fitting of the force measurements shown in **Fig. 2 a,b** was performed using the analytic fit builder for nonlinear fits through an adopted hyperbolic function with a Levenberg Marquardt iteration algorithm in OriginLab 2021.“

„Part 2. Analysis of the temperature dependence of protein adsorption to lipid layers

The measured maximum mass of adsorbed lysozyme on a surface was estimated (asymptotic fit to concentration-dependent adsorption of lysozyme on gold at 20 °C; **Extended Data Calculations Fig. 1**) to be 443 ng/cm². All other measured amounts (**Fig. 2a**) were taken as fractions (q) of this maximum amount, with $0 < q < 1$ (0-100%), acquired by the method shown in **Extended Data Calculations Figure 1**. At adsorption equilibrium,⁵⁵ the ratio of the adsorption (k_A) and desorption (k_D) constants is related to the free energy of adsorption (ΔG) by:

$$K = \frac{k_A}{k_D} = e^{\frac{-\Delta G}{RT}} = \frac{q}{c^*(1-q)} \quad (1)$$

With the concentration c^* ($c^* = 6.8 \cdot 10^{-6}$, derived from the lysozyme concentration of 100 $\mu\text{g ml}^{-1}$, a molar mass of 14.7 kDa, and normalization to a standard concentration of 1 M), we calculated the corresponding values of ΔG (**Extended Data Calculations Table** and **Error! Reference source not found.**). Enthalpy (ΔH) and entropy (ΔS) of protein adsorption were assumed to be invariant across the investigated temperature range (15 – 40 °C). From these assumptions, we calculated the different slopes of ΔG by a linear fit according to:

$$\Delta G(T) = \Delta H - T \cdot \Delta S \quad (2)$$

Relative to the references (thiocholesterol SAM, thiocholonic acid SAM and gold), we derived the entropic repulsion barrier of orientationally fluctuating cholesterol molecules in cholesterol SCLs to be -197 (~ -200) $\text{J} \cdot \text{mol}^{-1} \cdot \text{K}^{-1}$ (i.e., the difference between $\Delta S(\text{reference and SAMs})$ and $\Delta S(\text{Cholesterol SCL})$).

Extended Data Calculations Fig. 1
Estimate of the partial adsorption of lysozyme at $c = 100 \mu\text{g ml}^{-1}$. The adsorption of lysozyme at concentrations of 50, 100, 200, 500 and 1000 $\mu\text{g ml}^{-1}$ on gold substrates was quantified by quartz crystal microbalance (QCM) measurements. A hyperbolic asymptotic fit ($f(x) = a(1 - 1/(1 + c^*x)^{1/d})$, $R^2 = 0.99$) was used to calculate the full coverage, resulting in a partial coverage of $\sim 93\%$ at 100 $\mu\text{g ml}^{-1}$. All other datasets of the QCM measurements were correlated to estimate their respective fractions (q) used to derive the temperature dependent values of ΔG using equation (1).

		Cholesterol SCL		Thiocholesterol SAM		Thiocholonic acid SAM		Au reference	
T [K]	ΔG	[kJ/mol]	SD	[kJ/mol]	SD	[kJ/(mol)]	SD	[kJ/mol]	SD
288		-30.6	1.9	-31.0	0.8	-31.5	1.3	-34.5	0.8
293		-30.0	1.5	-31.4	1.5	-32.8	1.3	-35.7	1.1
298		-30.8	0.7	-31.3	0.3	-34.3	2.1	-36.6	0.7
303		-30.6	1.8	-32.0	1.3	-33.4	0.5	-35.5	0.4
308		-29.5	0.7	-33.1	1.1	-33.9	3.6	-36.8	1.4
313		-28.7	1.1	-34.3	1.4	-35.0	2.1	-38.3	0.8
	Slope (=ΔS) [J/(mol*K)]	85	29	-112	35	-112	27	-113	49

Extended Data Calculations Table 1, Lysozyme. The data were obtained using Equation (1) and the derived slope of ΔG based on the calculated maximum protein coverage fitted from data shown in Extended Data Calculations Fig. 2.

Temperature-dependent adsorption of bovine serum albumin and fibrinogen to cholesterol SCLs, thiocholesterol SAMs, thiocholonic acid SAMs and gold were quantified by quartz crystal microbalance (QCM). Equivalent to the method used for lysozyme, the approximation of the maximum surface coverage leads to estimated values of ΔG according to Equation (1). A linear fit according to Equation (2) allowed the comparison of the slope of the energy balance with increasing temperature. The graphical comparison (**Extended Data Calculations Fig. 2**) reveals the difference in slope of cholesterol SCL samples in comparison to control samples (thiocholesterol and thiocholonic SAM), which we regard as an entropic barrier caused by the interfacial orientational fluctuations of SCLs, also statistically verified by an F-Test. The calculated entropic barriers of cholesterol SCL can be given as about -200 , -110 and $-70 \text{ J}\cdot\text{mol}^{-1}\cdot\text{K}^{-1}$ for lysozyme, bovine serum albumin and fibrinogen, respectively. For lysozyme and bovine serum albumin adsorption on cholesterol SCLs, the slope of ΔG is still positive, which means that the entropy contribution dominates the energy balance. In the case of fibrinogen adsorption, this entropic contribution is still present, but no longer dominant. We assume that the adsorption of this larger and softer molecule leads to a higher enthalpic gain by deformation and/or an entropic gain due to partial unfolding.⁵⁶ Although systematic errors in the QCM measurements and biased values of the maximum protein coverage could be sources for altered scaling of the absolute values in **Extended Data Calculations Fig. 2**, this would not change the characteristic difference evident between cholesterol SCLs and all other substrates.“

Extended Data Calculations Fig. 2 Statistical comparison of the fitted slopes of ΔG , i.e., ΔS . Fitted slopes of $\Delta G (= \Delta S)$ are shown with error bars indicating standard errors of the fit. A statistical F-test which compared the fitted datasets (OriginPro 2020) indicated a highly significant difference, with $p < 0.001$, for all substrates compared to cholesterol SCLs.

Other comments:

(i) The authors should write more generally why adhesion is important. They should connect other scientific and technological aspects of their work.

We agree and have further endorsed the introductory statement to read (ln. 24-33):

“Adhesion, the attachment of dissimilar phases, is a fundamental but extremely multifaceted phenomenon governing the key role of interfaces in both living and man-made matter. Life has evolved a plethora of powerful principles to control adhesion, some of which have been recapitulated in engineered materials. Prominent examples include the superhydrophobic leaves of the sacred lotus¹ and the omniphobic surfaces of Nepenthes pitcher plants.² While molecular and structural characteristics governing interfacial phenomena in nature are widely studied, the physical mechanisms underlying the control of bioadhesion – the interfacial accumulation of biopolymers and cells (including bacteria) – are not yet thoroughly understood. Filling these knowledge gaps is a critical prerequisite for the rational design of advanced biofunctional materials.”

Similarly, we have expanded the concluding paragraph to better point out the general relevance of our findings (ln. 343-350):

“Given the widespread occurrence of cholesterol at various different types of tissue boundaries, our findings may shed new light on fundamentally important and ubiquitous interfacial processes of living matter. In particular, investigating whether cholesterol assemblies in lipid membranes can display fluctuating reorientations that entropically control bioadhesion will be a promising future research avenue. Beyond that, the development of cholesterol-inspired synthetic and scalable materials and systems to control bioadhesion is highly attractive but demands further analyses of the molecular requirements that govern the newly identified mechanism.”

(ii) Page 4, “The resulting interaction forces...are found to increase continuously with contact time.” This sentence might imply that there is not a maximum value for the force, since you can increase the contact time as desire. Please, comment.

*We are grateful for highlighting this point and agree that the sentence could misleadingly give the impression that the adhesion force would increase steadily with contact time, which is of course not the case. The results displayed in **Fig.2** show a third-order rate dependence of the initial adhesion force increase, however, within the range marked out by the adhesion force values determined for the reference surfaces (thiocholenic acid SAM and thiocholesterol SAM), and approaching saturation after a period of about 20 sec.*

We have modified the respective sentence in the revised manuscript to clarify this statement accordingly (ln. 140-143):

*“The resulting interaction forces were quantified (Error! Reference source not found.**a, b**) and found to initially increase with the contact time, **approaching a constant value at about 20 sec that remains within the range of forces determined for the SAM reference surfaces (Fig. 1a, b).**”*

(iii) I guess, if the authors could analyse the AFM images to try to distinguish topographic patterns (e.g. extended Fig 7).

*We fully agree that the topographic features of the investigated surfaces deserve attention in the context of our analysis as morphological differences are well known to potentially influence adhesion.^{9,10} The topography of the AFM images was therefore comprehensively analyzed and reported as average roughness (**Extended Data Fig. 7b**). We further examined whether topographic differences of the compared surfaces would affect their bioadhesion characteristics. No significant effects nor correlations were found. For example, in the **Extended Data Fig. 7** mentioned above, the mixed layer of 10 wt.% cholesterol and 90 wt.% stearyl palmitate displayed a comparably lower bioadhesion than the pure cholesterol SCL, despite of the significantly higher roughness of the multicomponent layer. Topography differences between the compared SCL surfaces were therefore concluded to be negligible for the bioadhesive characteristics of the systems investigated in our study. In fact, this observation further underlines the importance of the discovered entropic repulsion force as it dominates the interfacial properties to such an extent that it can override the influence of morphological effects on bioadhesion. Moreover, our newly performed molecular dynamics simulations also support the conclusion that topography does not control bioadhesion in our*

system: Cholesterol SCL (**Extended Data Fig. 12f**) as well as stigmasterol reference SCL (exhibiting high levels of bioadhesion and no entropic repulsion) similarly show a low surface roughness of around $RMS = 0.13$ nm.

Fig. 2 Quantitative evidence of entropic repulsion by cholesterol SCLs and verification of the molecular mechanism. (e) RMSD (for cholesterol multilayers and stigmasterol controls) over three simulation trajectories of z-component of the position of all O-atoms at the interface layer as a function of time for frames every 0.01 μs with error bars stating the standard error of the mean.

(iv) The authors should show a phase diagram of the behaviour of lipid systems. How does their structure change when heated?

We agree that a possible influence of the phase behavior of the system on the studied effects should be excluded. While bulk cholesterol preparations were in fact reported to attain a crystalline state at room temperature⁶ and within the investigated temperature range (i.e., 15 – 40 °C), our ATR-FTIR measurements (**Extended Data Fig. 4**) provide evidence that the investigated cholesterol multilayer system is amorphous. No indications of phase transition effects were obtained in the reported time-dependent experiments either. Thus, we can reasonably assume that the analyzed phenomena are related to non-crystalline cholesterol layers. We have added the following statement to the revised manuscript to clarify this point (ln. 235-240):

"Indeed, protein adsorption to cholesterol SCLs was observed to decrease when the temperature was increased from 15 to 40 °C (i.e., in a temperature range where significant changes in protein conformation²⁵⁻²⁷ and phase transitions of cholesterol can be excluded¹⁸), in contrast to minor changes in similar experiments with thiocholesterol and thiocholonic acid SAMs (serving as orientationally fixed controls representing opposite interfacial orientations of cholesterol SCLs underneath the adsorbed protein)"

(v) In QCM-D experiments the crystal baseline can change with temperature. That is, in one cycle it could have hysteresis. The authors should show such control to see if this could influence the final outcome.

We appreciate this comment since it points out that our description of the temperature-dependent QCM experiments was not clear enough. If the temperature were to be changed during a QCM experiment, baseline changes would in fact make protein adsorption processes difficult to interpret. However, our reported QCM-D data (**Fig. 3a** and **Extended Data Fig. 10** of the revised manuscript) refer to independent experiments performed at different but

constant temperatures. To avoid misunderstanding, we have changed the description of the experiment in the methods section of the revised manuscript (ln. 438-440):

“Temperature-dependent measurements (in the range of 15 - 40°C) were conducted using the temperature module of the instrument, keeping the set temperature constant throughout the measurement.”

(vi) It would be interesting if the authors using ellipsometry and QCM-D measurements could distinguish between wet and dry mass. This would represent an improvement on the experimental data.

We agree that comparing protein adsorption data obtained by quartz microbalance and by ellipsometry can provide additional insights into the investigated systems. In fact, we scrutinized the results of our comprehensive QCM-based protein adsorption studies by a set of ellipsometric experiments on similar systems. The gradation of the adsorbed amount agreed very well. As expected, the absolute values of the adsorbed protein mass determined by the two methods displayed some deviation. However, since this can not only be caused by differences of the hydration state of the adsorbed protein layer but also by the differing transport conditions applied and by assumptions made in the models used for evaluating the data, we decided to restrict ourselves to the presentation of the QCM-D based data. The method detects the fully hydrated and therefore most relevant state of adsorbing protein layers, offers high throughput and thus better statistical validation of the measurements.

(vii) With regard to figure 12 (extended information). I wonder what topographical error the AFM is making. For a 20 nm apex, what is the error that can be made? Does the tip always scan the same position? Why do you choose a time of 20 seconds and not 10 to 30? Please explain.

*The repeat accuracy of the AFM measurements in the x-y axis is below 5nm, regardless of the measuring tip used. The topographic error/ repeatability of the scan is accordingly. The waiting time of 20s between two successive measurement cycles was chosen on the basis of the previous contact angle measurements (**Fig. 2c** and **Supplementary Movies** of the revised manuscript). These measurements had shown that oscillations of the three-phase contact line on cholesterol SCLs occur for about 15-20 sec, i.e., the reorientation of the interface occurs within this period. To minimize the influence of the AFM tip on the dynamics of the interfacial cholesterol molecules, a pause of 20 sec was inserted between two consecutive measurements.*

(viii) The authors should consider the contact area when measuring the interaction from particles and bacteria. The axis could be renamed as (force/radius). Furthermore, such contact area could be used to discussed FRAP results.

We agree that normalizing the interaction forces acquired in the colloid-based and single-cell force spectroscopy measurements to the contact area could provide further information about the investigated systems. However, the reported experiment was designed to prove the dynamic adaptation of the cholesterol SCLs interface to the polarity of the probes brought in contact. A comparison of absolute force values per surface area was not intended. Therefore, we did not estimate the contact areas nor discuss interaction forces per area.

We also like the idea to relate the contact area of the particles and bacteria to the FRAP experiments. However, after careful consideration, we concluded that contact areas with radii of about $1\mu\text{m}$ are too small for a meaningful discussion of diffusion phenomena of cholesterol over areas with radii of $5\mu\text{m}$ as estimated from FRAP experiments. Furthermore, the results of adhesion force measurements with the (fixed) SAM control systems (providing the upper and lower force boundary values) indicate that lateral mobility is not relevant for the investigated effect.

(ix) Would it make sense to do contact angle measurements as a function of the temperature? Please, comment.

In the current report, contact angle measurements solely served the purpose of probing the interfacial orientational mobility of the compared multilayer systems. With this approach, we consistently observed polarity adaptations in dependence on the lipid composition of the multilayers. Since phase transitions of the lipid multilayer systems were excluded for the temperature range investigated, related temperature-dependent variations in the wetting behaviour could be excluded as well. While we are aware of earlier thermodynamic interpretations of temperature-dependent contact angle measurements^{11,12} we consider them far less straightforward (and far less relevant for the characterization of an anti-bioadhesive repulsion) than our presented analysis based on temperature-dependent protein adsorption data and therefore have not explored that route.

References:

1. Ulman, A. Formation and Structure of Self-Assembled Monolayers. *Chem. Rev.* **96**, 1533–1554 (1996).
2. Vericat, C., Vela, M. E., Benitez, G., Carro, P. & Salvarezza, R. C. Self-assembled monolayers of thiols and dithiols on gold: new challenges for a well-known system. *Chem. Soc. Rev.* **39**, 1805–1834 (2010).
3. Li, L., Chen, S., Zheng, J., Ratner, B. D. & Jiang, S. Protein adsorption on oligo(ethylene glycol)-terminated alkanethiolate self-assembled monolayers: The molecular basis for nonfouling behavior. *J. Phys. Chem. B* **109**, 2934–2941 (2005).
4. Norde, W. Biocolloids and biosurfaces energy and entropy of protein adsorption. *J. Dispers. Sci. Technol.* **13**, 363–377 (1992).
5. Jeon, S. I., Lee, J. H., Andrade, J. D. & De Gennes, P. G. Protein—surface interactions in the presence of polyethylene oxide: I. Simplified theory. *J. Colloid Interface Sci.* **142**, 149–158 (1991).
6. Loomis, C. R., Shipley, G. G. & Small, D. M. The phase behavior of hydrated cholesterol. *J. Lipid Res.* **20**, 525–535 (1979).
7. Shieh, H. S., Hoard, L. G. & Nordman, C. E. Crystal structure of anhydrous cholesterol [39]. *Nature* vol. 267 287–289 (1977).
8. Sun, M. *et al.* The effect of cellular cholesterol on membrane-cytoskeleton adhesion. *J. Cell Sci.* **120**, 2223–2231 (2007).
9. Rechendorff, K., Hovgaard, M. B., Foss, M., Zhdanov, V. P. & Besenbacher, F. Enhancement of protein adsorption induced by surface roughness. *Langmuir* **22**, 10885–10888 (2006).
10. Rechendorff, K., Hovgaard, M. B., Foss, M. & Besenbacher, F. Influence of surface roughness on quartz crystal microbalance measurements in liquids. *J. Appl. Phys.* **101**, 114502 (2007).
11. Kumar, V. & Errington, J. R. Wetting behavior of water near nonpolar surfaces. *J. Phys. Chem. C* **117**, 23017–23026 (2013).
12. Yuk, S. H. & Jhon, M. S. Temperature dependence of the contact angle at the polymer-water interface. *J. Colloid Interface Sci.* **116**, 25–29 (1987).

Reviewer Reports on the Second Revision:

Referees' comments:

Referee #1 (Remarks to the Author):

I have read this interesting manuscript again and have read with attention the reply to the referees. I maintain the original enthusiasm in the content of this paper but unfortunately, also the skepticism that I had has not changed.

I will start with a key point the authors must be able to tell with statistical rigor that the delta G scales linearly with temperature. I asked for statistical analysis, I get in response an R^2 of 0.69 with no comment on whether the error was correctly propagated in the calculation of such a quantity (e.g. were only the means considered or also the standard deviation, or were all points measured taken in account and for the determination of such points how was experimental error accounted for?). I also see two new proteins measured and no R^2 reported.

The second point I understand what the authors have done with SAMs but they are using a SAM that has 100% coverage of cholesterol as a control for their sample that for example has 10% how is this even acceptable? The authors have to perform proper controls with 1) SAMs at varying coverage choosing wisely the second component to fairly mimic their measurements and 2) crosslinked versions of their samples.

At present, I cannot be confident that their message is correct scientifically.

Referee #2 (Remarks to the Author):

The authors have done further work to address the questions raised by the reviewers. New data and explanations in the rebuttal provide more context for the thermodynamic basis of reduction in cell fouling. There is now better support for the Chol entropic contribution to cell fouling. I sense that the authors consider the protein and cell fouling results to be in alignment, but actually I do not believe they are. Cell fouling resistance is much better than protein fouling resistance; in fact, the level of protein resistance observed by the model systems is modest. This is not necessarily a criticism of the work but rather an observation. I recommend the authors reframe the manuscript to be more specific to understanding the mechanism of Collembola cellular antifouling.

1. I continue to be unimpressed by the protein fouling resistance. It is not necessary for me to see that the investigated cholesterol-based model systems compete with, or even outcompete, existing antifouling technologies. However, protein antifouling performance of the best current protein resistant coatings is often well below 10 ng/cm^2 , which provides a useful comparison. The SCL allows protein adsorption at the level of $\sim 150 \text{ ng/cm}^2$ (albumin), $\sim 400 \text{ ng/cm}^2$ (serum) (Fig 1) and even as high as $\sim 750 \text{ ng/cm}^2$ (fibrinogen, Extended Data Fig 10). The fibrinogen adsorption on SCL looks to be reduced only by roughly a factor of 2. I don't believe anyone working on protein antifouling coatings would consider this fouling resistant. On this basis I specifically object to comments such as in the abstract, "...strongly restricts protein adsorption...".

2. Given that SCL does not excel at resisting protein adsorption, it is important for the authors to be more precise in their use of the term "bioadhesion", which is currently too ambiguous. More accurately, good resistance to 'cellular bioadhesion' has been demonstrated. As an example of a recommended change, ending the title as "...cellular bioadhesion in Collembola cuticle" would be more accurate. Any implication that the results suggest a more general (molecular) resistance to bioadhesion is unwarranted.

3. I have no issue with using a normalization approach to compare SCL and control surfaces, and it is obvious that cell fouling is reduced on SCL compared to controls. The authors have provided a spreadsheet with the raw data, although the average values were not included. I had to calculate the averages myself, finding that cell adhesion on SCL was approximately 9.6% and 4% of the control for E coli and Staph epi, respectively. Because the linear plot is difficult for the reader to get a good feeling for these values, I recommend the authors either include the average values in

the text or in the figure legend or in the supporting information.

4. In summary, I think overall the work is novel, interesting and technically solid. But there is a disconnect between protein and cell fouling performance of SCL, which requires more precise terminology when discussing bioadhesion. Since protein resistance is much less effective than cell resistance, the manuscript should be reframed as a study focused solely on understanding the cell fouling resistance mechanism of Collembola.

Referee #3 (Remarks to the Author):

The authors have addressed most of my concerns and suggestions with regard to the original manuscript in their rebuttal. However, there are still a few small issues that need to be resolved.

1. The authors have extended and improved the data in Fig. 3a to illustrate the linear relationship between ΔG and temperature. However, the coefficients of determination (R^2) are not high enough (the largest R^2 value is only 0.81), which might mean that the linear relationship is not significant enough. Does this mean that there are other influencing factors besides the entropy repulsion mechanism?

2. For our second question, what we wanted to know is whether the surface morphology of the prepared SAM is sufficiently uniform and defect-free. The authors demonstrated the orientation of the surface groups by ATR-FTIR and the thickness of the SCLs was measured by ellipsometry. However, in Extended Data Fig. 7c, the surface roughness of obtained SCLs is very large. Whether this roughness of the sample surface plays a role in the anti-bioadhesion.

Referee #4 (Remarks to the Author):

The revised manuscript has increased its quality. Although I cannot speak for the other referees, I would consider this version for publication in Nature (or if Nature Physics). The authors have addressed all my questions in detail. I am satisfied with their arguments. The main question is simple but has a complex experimental proof. Therefore some aspects related to the mechanism at molecular scale are still be open for discussion (which I consider a good thing). In the revised version (and the answer to the referees) the authors present new experimental data which have complemented with computer simulations. In general, they have combined the right available experimental techniques to tackle the main hypothesis.

In particular, they have addressed my demands of showing the fittings of the data.

1. Key results.

Answer:

The work is very interesting and will be of importance for researches investigating adhesion phenomena at micro and nanoscale. For sure it will influence new experimental work in different fields. The main idea is simple and therefore it can be attractive for other researches. The authors propose the idea that entropic repulsion induced by molecular orientational fluctuations could be crucial for designing new anti-bioadhesive structures.

2. Originality and significance (novelty).

Answer: Entropic repulsion at interfaces is not a new idea. However, the idea (the system) proposed by the authors can be considered novel. I find it inspiring. In the revised version (and the answers to the referees) they quote adequate references.

3. Data and methodology.

Answer: The paper is technically robust. The experiments are well planned and well performed. They authors use an appropriate palette of experimental techniques to tackle different aspects of the main problem.

4. Statistics.

Answer:
The authors use the right statistics.

5. Conclusions.

Answer:
The revised manuscript presents new results and arguments that make the conclusions reliable.

6. Improvements.

Answer:
Not necessary at this stage.

7. References.

Answer:
The revision has improved this aspect.

8. Clarity and context.

Answer:
The new text has improved the clarity of the whole paper. The abstract reflects the importance and the main results of the paper. The introduction is very informative and the conclusions are well written.

Author Rebuttals to Second Revision:

Referees' comments

Reviewer #1:

I have read this interesting manuscript again and have read with attention the reply to the referees. I maintain the original enthusiasm in the content of this paper but unfortunately, also the skepticism that I had has not changed.

We very much appreciate the reviewer's continued enthusiasm, as well as her/his further helpful critical comments. We have carefully considered the specific points mentioned and addressed them in the revision of our manuscript as subsequently detailed.

I will start with a key point the authors must be able to tell with statistical rigor that the delta G scales linearly with temperature. I asked for statistical analysis, I get in response an R² of 0.69 with no comment on whether the error was correctly propagated in the calculation of such a quantity (e.g. were only the means considered or also the standard deviation, or were all points measured taken in account and for the determination of such points how was experimental error accounted for?). I also see two new proteins measured and no R² reported.

We are grateful the reviewer points out that further detail on the applied statistical analysis of the linear fits of the protein adsorption data should be provided. We have revised the manuscript accordingly to contain the respective information, including the error propagation, in the section Extended Data Calculations Part 2.

The section now comprises all measured data points including the respective standard deviations. The more detailed analysis of the standard error of the slope of the linear fits of the temperature dependence of ΔG clearly confirms that the slope of ΔG for the adsorption of lysozyme to cholesterol SCLs is significantly different from the slope of ΔG for the adsorption of lysozyme to the controls (as already shown by the F-test analysis in the Extended Data Calculation Fig. 2), and that this difference corresponds to an entropic adsorption barrier of ΔS of $-200 \pm 60 \text{ J mol}^{-1} \text{ K}^{-1}$, which is the main conclusion of our analysis. The extended statistical analysis further shows that the error of ΔS propagating from the standard error of the slopes allows for a definite determination of ΔS .

We have adjusted the respective sections of the revised manuscript to provide this information:

(line 103-114, Supplementary Information)

"With this derivation of ΔG , we calculated the different slopes of $\Delta G(T)$ (providing ΔS of the protein adsorption) by a linear fit according to Equation (2) (assuming changes in ΔH within this narrow temperature range to be neglectable):

$$\Delta G(T) = \Delta H - T \cdot \Delta S \quad (2)$$

With this approach, we determined the entropic repulsion barrier caused by orientationally fluctuating cholesterol in the topmost part of cholesterol SCLs to be

$-200 \pm 60 \text{ J}\cdot\text{mol}^{-1}\cdot\text{K}^{-1}$ (i.e., the difference between ΔS of the adsorption to SAMs or gold and to cholesterol SCLs). The statistical error (standard error) of the linear fits (see **Extended Data Calculation Table 1**) was used to calculate the standard error of the determination of the entropic repulsion barrier. A statistical F-test also indicated the significantly different slope of $\Delta G(T)$ for the adsorption to cholesterol SCLs (**Extended Data Calculation Figure 2**)."

The section **Extended Data Calculations Part 2** of the revised manuscript now similarly provides the details of the statistical analysis for the determination of the temperature dependence of ΔG for the adsorption of bovine serum albumin and fibrinogen, including the respective ΔG values, their errors, and the fitted slopes ($=\Delta S$) and their errors in the newly added **Extended Data Calculation Table 2 & 3**.

		Cholesterol SCL		Thiocholesterol SAM		Thiocholonic acid SAM		Au reference	
T [K]	ΔG	[kJ/mol]	SE	[kJ/mol]	SE	[kJ/(mol)]	SE	[kJ/mol]	SE
288		-33.4	1.4	-36.0	1.7	-35.8	1.3	-42.1	10.6
293		-32.2	1.5	-35.9	1.7	-36.5	2.2	-37.9	1.2
298		-32.4	0.8	-37.0	0.9	-38.1	1.3	-38.0	1.2
303		-32.4	1.7	-36.9	1.7	-36.1	1.2	-45.3	7.8
308		-32.4	0.9	-38.6	1.7	-37.3	0.4	-43.8	2.1
313		-31.0	1.6	-37.5	1.1	-37.5	0.6	-42.7	5.3
	Slope ($=\Delta S$) [J/(mol*K)]	48	26	-69	28	-51	33	-345	103

Extended Data Calculations Table 2 ΔG and ΔS for bovine serum albumin adsorption.

		Cholesterol SCL		Thiocholesterol SAM		Thiocholonic acid SAM		Au reference	
T [K]	ΔG	[kJ/mol]	SE	[kJ/mol]	SE	[kJ/(mol)]	SE	[kJ/mol]	SE
288		-37.5	0.5	-40.4	0.8	-39.0	0.1	-42.0	2.5
293		-37.4	1.3	-39.9	1.1	-40.2	0.5	-43.4	3.9
298		-38.1	1.4	-41.4	1.2	-40.2	0.3	-43.5	4.5
303		-38.0	1.3	-42.4	2.1	-41.2	0.3	-43.3	3.4
308		-38.1	0.4	-43,04	1.2	-41.4	0.1	-44.7	3.4
313		-38.8	0.3	-42,90	1.1	-41.6	0.3	-46.0	2.8
	Slope ($=\Delta S$) [J/(mol*K)]	-49	10	-120	23	-119	6	-144	30

Extended Data Calculations Table 3 ΔG and ΔS for fibrinogen adsorption.

In addition, we have adapted the corresponding paragraph:

(line 120-131, Supplementary Information)

"The temperature dependent adsorption of bovine serum albumin and fibrinogen to cholesterol SCLs, thiocholesterol SAMs, thiocholonic acid SAMs and gold was quantified based on QCM experiments (**Extended Data Fig. 5a,b**). Equivalent to the method used for lysozyme, the approximation of a maximal surface coverage provides ΔG values according to Equation (1) (**Extended Data Calculation Table 2 & 3**). The linear fit of $\Delta G(T)$ according to Equation (2) allowed again the comparison of the

*entropic contributions to the free energy balance of the adsorption. **Extended Data Calculations** Fig. 2 displays the differences between the slopes of $\Delta G(T)$ for cholesterol SCL and thiocholesterol or thiocholenic SAMs, which can be used to calculate ΔS of the entropic repulsion barrier to cholesterol SCLs. A statistical F-Test of the slopes clearly indicates the significantly different slope of cholesterol SCL. The calculated entropic barriers of cholesterol SCL against the adsorption of bovine serum albumin and fibrinogen were calculated to -110 ± 60 and $-70 \pm 30 \text{ J}\cdot\text{mol}^{-1}\cdot\text{K}^{-1}$, respectively.”*

The second point I understand what the authors have done with SAMs but they are using a SAM that has 100% coverage of cholesterol as a control for their sample that for example has 10% how is this even acceptable? The authors have to perform proper controls with 1) SAMs at varying coverage choosing wisely the second component to fairly mimic their measurements and 2) crosslinked versions of their samples. At present, I cannot be confident that their message is correct scientifically.

We respectfully disagree that our presented analysis requires the investigation of multicomponent SAMs as the reviewer suggests. As explained in the point-by-point response to the reviewer's comments on our original submission, thiocholesterol and thiocholenic acid SAMs were used in our study to explore how restricted molecular mobility deteriorates the anti-bioadhesive properties of cholesterol independent of its orientation. The results demonstrate that fixed cholesterol layers of either orientation lack the anti-bioadhesive characteristics of cholesterol SCLs. The additional presence of another component without anti-bioadhesive properties can be reasonably excluded to alter the restricted molecular mobility and the absence of anti-bioadhesive properties of thiocholesterol and thiocholenic acid SAMs. Therefore, we fail to see how the investigation of multicomponent SAMs could contribute to the further elucidation of the repulsive characteristics of cholesterol-containing multicomponent SCLs. (Apart from the fact that the exact composition of the topmost part of the multicomponent SCL, which would have to be replicated by multicomponent SAMs, is unknown.) Of note, multicomponent SCLs were not included in our study to further elucidate the antiadhesive capacity of cholesterol assemblies but to relate our findings on cholesterol SCLs to naturally occurring systems with lower cholesterol contents. The results impressively demonstrated the effectiveness and robustness of the newly discovered anti-bioadhesive mechanism even for layer systems with low cholesterol contents.

Reviewer #2:

The authors have done further work to address the questions raised by the reviewers. New data and explanations in the rebuttal provide more context for the thermodynamic basis of reduction in cell fouling. There is now better support for the Chol entropic contribution to cell fouling. I sense that the authors consider the protein and cell fouling results to be in alignment, but actually I do not believe they are. Cell fouling resistance is much better than protein fouling resistance; in fact, the level of protein resistance observed by the model systems is modest. This is not necessarily a criticism of the work but rather an observation. I recommend the authors reframe the manuscript to be more specific to understanding the mechanism of *Collembola* cellular antifouling.

We are pleased to see the reviewer appreciates the extended thermodynamic analysis presented by our revised manuscript and are grateful for her/his additional comments on the delineation of the repulsive effect of cholesterol-containing SCLs on proteins and bacterial cells, which we specifically address below.

1. I continue to be unimpressed by the protein fouling resistance. It is not necessary for me to see that the investigated cholesterol-based model systems compete with, or even outcompete, existing antifouling technologies. However, protein antifouling performance of the best current protein resistant coatings is often well below 10 ng/cm², which provides a useful comparison. The SCL allows protein adsorption at the level of ~150 ng/cm² (albumin), ~400 ng/cm² (serum) (Fig 1) and even as high as ~750 ng/cm² (fibrinogen, Extended Data Fig 10). The fibrinogen adsorption on SCL looks to be reduced only by roughly a factor of 2. I don't believe anyone working on protein antifouling coatings would consider this fouling resistant. On this basis I specifically object to comments such as in the abstract, "...strongly restricts protein adsorption...".

As explained in the point-by-point response to the reviewer's comments on our original submission, we agree that the adsorbed amount of protein to cholesterol-containing SCLs is not as low as reported for some of the fully elaborated anti-fouling technologies based on other design principles. We reiterate that we do not claim that our cholesterol-based model systems developed to investigate the hitherto unknown phenomenon would immediately outcompete existing anti-fouling technologies and that further work will be needed to enable the technological translation of the obtained findings.

However, we would like to emphasize that our reported data clearly show that protein adsorption to cholesterol-containing SCLs is significantly reduced compared to layers of compositionally related compounds, which is in remarkable agreement with the observed resistance to bacterial cell adhesion. Of note, more effective protein-resistant surfaces (the reviewer is referring to) are based on entirely different design concepts compared to the lipid-based layers investigated in our study: While polymeric brush-like and/or ionized (zwitterionic) coatings have in fact been reported to result in highly protein-repellent surfaces, we are not aware of earlier reports on the discovered capability of assemblies of low molecular weight lipids. This important distinction was also explicitly pointed out by reviewer #3 in his/her comments on our original submission. In fact, it seems likely that both the previously known polymer- and /or charge-based repulsion effects and our newly discovered orientational

entropy-based repulsive mechanism coexist in living nature and complementarily restrict bioadhesion.

Nonetheless, we fully agree with reviewer #2 that the degree of anti-bioadhesive efficacy should be unambiguously stated. Accordingly, we have further modified three sentences on the protein resistance of our reported systems in the revised manuscript to avoid misperception by the reader:

(line 12-14, Abstract)

*“We discovered that entropic repulsion caused by interfacial orientational fluctuations of cholesterol layers **strongly** restricts protein adsorption and bacterial adhesion.”*

(line 101-103)

*“The results of single protein adsorption experiments were **impressively** confirmed when applying physiologically relevant protein mixtures (i.e., 10 vol.% fetal bovine serum) (**Error! Reference source not found.i**).”*

(line 156-158)

*“The **striking** anti-bioadhesive properties of cholesterol-containing SCLs were shown above to correlate with a dynamic adaptation of the interfacial orientation of cholesterol in response to changes in the environmental polarity.”*

2. Given that SCL does not excel at resisting protein adsorption, it is important for the authors to be more precise in their use of the term ‘bioadhesion’, which is currently too ambiguous. More accurately, good resistance to ‘cellular bioadhesion’ has been demonstrated. As an example of a recommended change, ending the title as “...cellular bioadhesion in Collembola cuticle” would be more accurate. Any implication that the results suggest a more general (molecular) resistance to bioadhesion is unwarranted.

To better delineate the degree of anti-bioadhesive efficacy we have modified the statement on the efficacy of the protein resistance of our reported systems in three sentences of the revised manuscript as indicated above.

Since our reported data clearly show significantly lower protein adsorption to cholesterol-containing SCLs compared to compositionally related systems, and since the bacterial adhesion data confirm the generally accepted mechanistic connection of molecular and cellular bioadhesion, we find the proposed restriction of our statement on bioadhesion to “cellular bioadhesion” inappropriately narrow.

3. I have no issue with using a normalization approach to compare SCL and control surfaces, and it is obvious that cell fouling is reduced on SCL compared to controls. The authors have provided a spreadsheet with the raw data, although the average values were not included. I had to calculate the averages myself, finding that cell adhesion on SCL was approximately 9.6% and 4% of the control for E coli and Staph epi, respectively. Because the linear plot is difficult for the reader to get a good feeling for these values, I recommend the authors either include the average values in the text or in the figure legend or in the supporting information.

We in fact decided against providing the numerical bacterial adhesion average values in the main document. However, we would like to point out that the data presented in all figures are shared according to the guidelines of the journal. Given the large number of bacterial adhesion data reported in our manuscript, we very much prefer not to include additional information in the text or legends accompanying the figures, as this may affect the manuscript's readability.

4. In summary, I think overall the work is novel, interesting and technically solid. But there is a disconnect between protein and cell fouling performance of SCL, which requires more precise terminology when discussing bioadhesion. Since protein resistance is much less effective than cell resistance, the manuscript should be reframed as a study focused solely on understanding the cell fouling resistance mechanism of Collembola.

We are pleased about the very positive evaluation of our study. We agree that the anti-bioadhesive efficacy of the investigated cholesterol-containing SCLs should be described unambiguously and have further amended the manuscript accordingly. However, in view of the significantly lower protein adsorption to cholesterol-containing SCLs when compared to compositionally related systems, and considering the experimentally confirmed connection of molecular and cellular bioadhesion (in line with the generally accepted knowledge in the field and comments of reviewer #1 on our original submission), we would not find it appropriate to reframe the manuscript "as a study focused solely on understanding the cell fouling resistance mechanism of Collembola". In fact, we consider the newly discovered antiadhesive mechanism to be a universal interfacial phenomenon and would like to point out that the other reviewers share this view.

Referee #3:

The authors have addressed most of my concerns and suggestions with regard to the original manuscript in their rebuttal. However, there are still a few small issues that need to be resolved.

We are very pleased to read the reviewer finds his/her concerns and suggestions adequately addressed and respond to the few additional points mentioned below.

1. The authors have extended and improved the data in Fig. 3a to illustrate the linear relationship between ΔG and temperature. However, the coefficients of determination (R^2) are not high enough (the largest R^2 value is only 0.81), which might mean that the linear relationship is not significant enough. Does this mean that there are other influencing factors besides the entropy repulsion mechanism?

As pointed out above (please see answer to the comments of reviewer #1), we agree that further details of our statistical analysis for calculation of ΔG and ΔS were required and provide them now in the Extended Data Calculations Part 2 section of our revised manuscript. Although the R^2 values of the linear fits of the temperature-dependent protein adsorption data are indeed not very high, it should be taken into account here that the underlying experiments on multiple samples are typically associated with a rather large scatter. However, the extended statistical analysis shows that despite the scatter, the slopes of the linear fits for cholesterol SCL are significantly different from the compared controls. Thus, the calculation of the entropy threshold for adsorption based on this can therefore clearly be regarded as reliable. The values determined (entropic barrier ΔS of $-200 \pm 60 \text{ J mol}^{-1} \text{ K}^{-1}$ for the adsorption of lysozyme to cholesterol-based SCLs) indicate that the uncovered entropic repulsion mechanism dominates the investigated system.

2. For our second question, what we wanted to know is whether the surface morphology of the prepared SAM is sufficiently uniform and defect-free. The authors demonstrated the orientation of the surface groups by ATR-FTIR and the thickness of the SCLs was measured by ellipsometry. However, in Extended Data Fig. 7c, the surface roughness of obtained SCLs is very large. Whether this roughness of the sample surface plays a role in the anti-bioadhesion.

We regret and apologize that our response to the respective comment on our original submission was not sufficiently clear. This might be caused by the fact, that our manuscript provides roughness data of multiple systems, namely of SAMs, cholesterol SCLs, fatty acid SCLs and cholesterol-containing multicomponent SCLs. The roughness analysis was in all cases based on AFM measurements.

*As to the SAM systems, we were able to confirm very smooth and nearly defect-free surfaces and homogeneous protein adsorption. (please see the corresponding figure from our response to the reviewer's comments on the original submission below: **Response Letter Figure 1**).*

For fatty acid SCLs and cholesterol-containing multicomponent SCLs we in fact report elevated roughness, which the reviewer is referring to. However, as explained in the point-by-point response to the reviewer's comments on our original submission (please see the respective answer to reviewer #4 given below), the identified anti-bioadhesive characteristics of

cholesterol-containing multicomponent SCLs are clearly independent of the detected variations in roughness: Smooth cholesterol SCL and rough cholesterol-containing multicomponent SCL similar display anti-bioadhesive properties.

From the point-by-point response to the comments of reviewer#4 on our original submission:

“For example, in the **Extended Data Fig. 3**, the mixed layer of 10 wt.% cholesterol and 90 wt.% stearyl palmitate displayed a comparably lower bioadhesion than the pure cholesterol SCL, despite of the significantly higher roughness of the multicomponent layer, i.e., the discovered entropic repulsion force dominates the interfacial properties to an extent that it can override the influence of morphological effects on bioadhesion. Moreover, our newly performed molecular dynamics simulations also support the conclusion that topography does not control bioadhesion in our system: Cholesterol SCL (**Extended Data Fig. 6f**) as well as stigmasterol reference SCL (exhibiting high levels of bioadhesion and no entropic repulsion) similarly show a low surface roughness of around $RMS = 0.13 \text{ nm}$.”

Response Letter Figure 1: Fluorescently labelled bovine serum albumin homogeneously adsorbs to thiocholesterol SAMs. $\sigma_{\text{Fluorescence}}$ is the standard deviation of the fluorescence signal of an image, indicating the homogeneity of surface coverage.

Reviewer #4:

The revised manuscript has increased its quality. Although I cannot speak for the other referees, I would consider this version for publication in Nature (or if Nature Physics). The authors have addressed all my questions in detail. I am satisfied with their arguments. The main question is simple but has a complex experimental proof. Therefore some aspects related to the mechanism at molecular scale are still be open for discussion (which I consider a good thing). In the revised version (and the answer to the referees) the authors present new experimental data which have complemented with computer simulations. In general, they have combined the right available experimental techniques to tackle the main hypothesis. In particular, they have addressed my demands of showing the fittings of the data.

We are very grateful for the insightful comments of the reviewer and pleased to read that he/she finds our revision of the manuscript appropriate.

Reviewer Reports on the Third Revision:

Referees' comments:

Referee #1 (Remarks to the Author):

Unfortunately, my opinion on this paper has not changed throughout this process. The concept is interesting and I like it a lot. The data presented do not build confidence in me that the concept is correct. I have doubts on the statistical analysis and I have more serious concerns on the lack of proper controls. I guess at this point it will be an editorial decision whether this manuscript is accepted or not. I remain enthusiastic on the concept and highly skeptical on whether is concept is supported by data.

Referee #2 (Remarks to the Author):

The authors have removed several superlatives describing the level of protein adsorption (strongly, impressively, striking) and this is an improvement. What remains unchanged are the differences of opinion between this reviewer and the authors on the protein fouling performance. The data in my view only show a modest reduction in protein adsorption. A statistically significant reduction is not the same as a meaningful reduction. I do not believe a factor of two reduction in protein adsorption is a meaningful reduction in terms of biofouling.

Referee #3 (Remarks to the Author):

The standard errors of the energy barriers calculated by the authors are close to half of the mean values (for example, -200 ± 60 J/(mol*K), -110 ± 60 J/(mol*K) and -70 ± 30 J/(mol*K)). This data seems to be not solid enough to support the main hypothesis. I wonder why such a large error range exists under the relatively ideal experimental conditions (for example, low roughness as well as single-component 100% coverage surface) given by the authors.

Author Rebuttals to Third Revision:

December 08, 2022

Point-by-point response to the reviewer's comments

on Jens Friedrichs et al. *Entropic repulsion of cholesterol-containing layers counteracts bioadhesion*.

Reviewer #1:

Unfortunately, my opinion on this paper has not changed throughout this process. The concept is interesting and I like it a lot. The data presented do not build confidence in me that the concept is correct. I have doubts on the statistical analysis and I have more serious concerns on the lack of proper controls. I guess at this point it will be an editorial decision whether this manuscript is accepted or not. I remain enthusiastic on the concept and highly skeptical on whether is concept is supported by data.

We regret that the reviewer remains skeptical despite our substantially extended validation of the proposed mechanism. Since it is difficult for us to glean the exact reasons for the persisting concerns from the reviewer's rather general verdict, we explain below how we understand the respective statements before responding:

First, doubts are raised about the statistical analysis without specifying them. Since the revised manuscript contains the complete statistical analysis of all reported data, including the error progression (addressing the reviewer's previous comments), it is not clear what the comment is referring to. If the doubts would refer to the experimental uncertainty of the protein adsorption data used to quantify the entropic adhesion barrier, we refer to our response to Reviewer #3 below. We would like to recall that the quantification of the entropic repulsion by first-principles thermodynamics provides significant results for three different globular proteins (lines 201 – 211 and lines 231 – 235 of the manuscript) and is in agreement with an independent estimation of the effect from geometrical considerations, relating the cross-sectional area of protein-facing cholesterol at the SCL interface to the area of orientationally free cholesterol at the SCL-solution interface (Fig. 3b, lines 215 – 230 of the manuscript and Extended Data Calculations Part 3). Beyond that, the entropic repulsion mechanism was independently proven by molecular dynamics (MD) simulations (lines 236 – 249 of the manuscript) and by AFM-based force mapping experiments (lines 250 – 259 of the manuscript).

Second, the reviewer points out to be even more seriously concerned about “the lack of proper controls” in our study. We assume this refers to their following earlier comments:

1st review:

“...if I understand correctly the authors used SAMs solely composed of cholesterol as controls for wax/cholesterol spin-coated sample where cholesterol was 10%, how can this be acceptable?”

2nd review (of resubmitted manuscript):

„... they are using a SAM that has 100% coverage of cholesterol as a control for their sample that for example has 10% how is this even acceptable? The authors have to perform proper controls with 1) SAMs at varying coverage choosing wisely the second component to fairly mimic their measurements and 2) crosslinked versions of their samples.”

In the responses to these earlier comments, we have already clarified the role of SAM controls in our study. As those explanations were obviously not sufficient to convince the reviewer, we further expand our response here:

Bioadhesion data of monocomponent cholesterol SCLs in comparison to cholesterol SAM controls (i.e., thiocholesterol and thiocholenic acid SAMs) revealed the decisive role of orientational mobility for the antiadhesive characteristics (lines 88 – 92 of the manuscript, Fig. 1 f, g and Extended Data Fig. 2). Cholesterol SAMs were also used as controls in the quantification of the entropic repulsion from protein adsorption data (lines 193 – 201 of the manuscript, Fig. 3a, Extended Data Fig. 5 and Extended Data Calculations Part 2). Thus, SAM controls were vital to unraveling and quantifying the entropic repulsion of orientationally unrestricted cholesterol assemblies.

Bioadhesion experiments with multicomponent SCLs were based on these findings and performed to explore the antiadhesive effect of orientationally unconstrained cholesterol in the presence of bioadhesion-enhancing lipid components (using stearyl palmitate as an example). The results show that multicomponent SCLs containing only minor fractions of cholesterol in addition to the bioadhesion-enhancing stearyl palmitate effectively restrict bioadhesion (lines 95 – 105 of the manuscript, Fig. 1 h, I and Extended Data Fig. 3). The controls required to interpret these bioadhesion experiments are pure cholesterol SCLs and pure stearyl palmitate SCLs, which are both included (samples designated as 0/100 and 100/0).

Admittedly, the graphical presentation of the data plots in Fig. 1 could have been misunderstood to assume that the mono- and multicomponent SCLs would similarly rely on orientationally fixed SAM controls. However, multicomponent SAM controls (or crosslinked SCLs), as requested by the reviewer, would be of no further use in interpreting bioadhesion data of multicomponent SCLs, because we have already clearly shown before that orientation-fixed cholesterol lacks any repulsive capacity.

To better avoid misperception, we have modified Fig. 1 of the revised manuscript to distinguish more clearly mono- and multilayer data and to unambiguously display the composition of the compared systems graphically:

Fig. 1 Layers of *Collembola* cuticular lipids and their bioadhesion properties. **(a)** Image of *Tetrodontophora bielanensis* (*T. bielanensis*), an exemplary *Collembola* spp. Scale bar = 1 mm. **(b)** Scanning electron microscopy image of a *T. bielanensis* cuticula. Scale bar = 500 nm. **(c)** Cross-section schematic of the cuticula, showing a layered structure consisting of a chitin-rich inner skeleton covered by a protein-rich layer. A thin lipid-rich envelope covers the protein-rich layer. Scale bar = 200 nm. **(d)** Summary of the detected lipids in the outer cuticula layer of *T. bielanensis*. **(e)** Layers of *Collembola* cuticular lipids: Spin-Coated Lipid Multilayers (SCLs) containing cholesterol allow for orientational adaptation of the topmost lipids to the polarity of the environment. ATR-FTIR [Supplementary Fig. 2] and dynamic contact angle measurements (Fig. 2c and Extended Data Fig. 9a) indicate highly ordered cholesterol molecules, with the hydrocarbon tail of the outer cholesterol layer initially oriented towards the interface, while the hydroxyl groups are oriented inward. Self-Assembled Monolayers (SAMs) chemisorbed to gold via thiol groups, with either the polar or nonpolar sides of cholesterol oriented to the interface, served as references in selected experiments. **(f, i)** Adsorbed amount of protein (lysozyme or fetal bovine serum) on **(f)** monocomponent layers of *Collembola* cuticular lipids and **(i)** multicomponent SCLs

of stearyl palmitate and cholesterol, determined by quartz crystal microbalance measurements. **(g, h)** Normalized adherent cells of *Staphylococcus epidermidis* on **(g) monocomponent** layers of *Collembola* cuticular lipids and **(h) multicomponent SCLs** of stearyl palmitate and cholesterol. Data are normalized to the average adherent cell density on a silica (SiO₂) substrate. **In h, i, pure stearyl palmitate SCLs (100/0) and pure cholesterol SCLs (0/100) served as negative and positive controls, respectively.** Graphs **(f-i)** show the mean + standard deviation. Data were obtained from at least three independent experiments. Ns ($p > 0.05$), **** ($p < 0.0001$), unpaired t-test.

Reviewer #2:

The authors have removed several superlatives describing the level of protein adsorption (strongly, impressively, striking) and this is an improvement. What remains unchanged are the differences of opinion between this reviewer and the authors on the protein fouling performance. The data in my view only show a modest reduction in protein adsorption. A statistically significant reduction is not the same as a meaningful reduction. I do not believe a factor of two reduction in protein adsorption is a meaningful reduction in terms of biofouling.

We are pleased that Reviewer #2 finds our revision improved.

Referring to the remaining disagreement on the scoring of protein adsorption data, we would like to emphasize again that our study was targeting a previously unknown anti-adhesive mechanism of non-covalently assembled lipid systems but not its technological implementation. In particular, we are not claiming that the detected protein repulsion effect would immediately outperform established anti-adhesive coating technologies. Instead, we clearly state in the manuscript that the perspective technological use of the discovered design principle will require further development.

However, we would like to point out that the investigated anti-adhesive system may offer valuable advantages over the widely investigated hydrated polymer brushes as the latter are known to be of limited stability: Small-molecule-based, non-covalent assemblies can be reasonably considered to be more robust and potentially self-renewing. Also, brush-like polymer structures might act synergistically with anti-adhesive lipid systems in living nature, which deserves further exploration per se and as a potential template for the two-tier design of engineered anti-adhesive surfaces.

Thus, while we understand the perspective taken by the reviewer, we consider the reported anti-bioadhesive effects of orientationally unconstrained cholesterol layers meaningful as they allowed for the identification and quantification of a previously unknown entropic repulsion mechanism.

Reviewer #3:

The standard errors of the energy barriers calculated by the authors are close to half of the mean values (for example, -200 ± 60 J/(mol*K), -110 ± 60 J/(mol*K) and -70 ± 30 J/(mol*K)). This data seems to be not solid enough to support the main hypothesis. I wonder why such a large error range exists under the relatively ideal experimental conditions (for example, low roughness as well as single-component 100% coverage surface) given by the authors.

We appreciate the critical comment by the reviewer, which showed the necessity better to explain the characteristics and conditions of the applied analysis.

In fact, our approach relies on analyzing bioadhesion-related experimental data (i.e., protein adsorption data directly representing a measure of bioadhesion) by first-principles thermodynamics (i.e., avoiding customized models using multiple fitting parameters). The magnitude of the error in the determination of the entropic barrier by this analysis depends on the choice of the protein but not on the quality and reproducibility of the layered surfaces, which is rather ideal (as the reviewer correctly mentioned):

Lysozyme, as a small and compact globular protein, displays rather uniform adsorption characteristics. It is, therefore, best suited to quantify the entropic repulsion barrier from adsorption data at different temperatures and over multiple sample types. The analysis of this system is associated with a standard error of less than one-third of the absolute value and in impressive agreement with theoretical estimates. This primary validation of our proposed mechanism (lines 201 – 212 of the manuscript, Extended Data Fig. 5c and Extended Data Calculations Part 2) certainly cannot be denied.

The data obtained for the adsorption of the much larger and conformationally less stable globular proteins albumin and fibrinogen show greater scatter (lines 231 – 235 of the manuscript). However, the analysis similarly provides evidence of a significant entropic repulsion. The adsorbed amounts of large and 'soft' globular proteins are well-known to be influenced by the coexistence of multiple interaction modes and conformational changes of the protein.¹⁻⁶ Accordingly, the entropic repulsion obtained from the adsorption data of the different proteins varies as adsorption-induced conformational changes differently contribute to the free enthalpy of adsorption.⁷⁻¹¹

Together, the significant entropic repulsion barrier obtained for all investigated systems underscores the robustness of the analysis and the general relevance of the discovered mechanism.

To clarify the above points, we have further revised the Supplementary Information provided with the manuscript, elaborating more on the statistical analysis and error propagation of our experimental data:

(lines 103 – 125 of Supplementary Information)

“With this derivation of ΔG , we calculated the different slopes of $\Delta G(T)$ (providing ΔS of the protein adsorption) by a linear fit according to Equation (2):

$$\Delta G(T) = \Delta H - T \cdot \Delta S \quad (2)$$

Within this first-principle approach, the enthalpy (ΔH) and the entropy (ΔS) of protein adsorption were assumed to be invariant across the investigated temperature range (15 – 40 °C). An entropic repulsion barrier of $-200 \pm 60 \text{ J} \cdot \text{mol}^{-1} \cdot \text{K}^{-1}$ for the adsorption of lysozyme to cholesterol SCLs was determined from the difference between ΔS of the adsorption to cholesterol SAMs and SCLs. This analysis provides the entropic repulsion of proteins at cholesterol SCLs despite additional entropic contributions associated with conformational changes and interactions with water molecules. The statistical uncertainty (standard error) of the linear fits of $\Delta G(T)$ (see Extended Data Calculation Table 1) was used to calculate the standard error of the determined entropic repulsion (sum of absolute values of the standard error of ΔS_{chol} and $\Delta S_{\text{Au/SAM}}$). A statistical F-test (OriginPro 2020) indicated the significantly different slope of $\Delta G(T)$ for the adsorption of lysozyme to cholesterol SCLs (Extended Data Calculation Fig. 2).

The quantification of the entropic repulsion from adsorption data of larger, globular proteins is associated with higher statistical uncertainty. This results from a greater scatter in the adsorbed amounts due to their different interaction modes and/or adsorption-induced conformational changes.

However, the fact that first-principle thermodynamics provides a dominant contribution of entropic repulsion to the overall free energy balance demonstrates the importance of the discovered effect.”

(lines 148 – 154 of Supplementary Information)

“With this full error propagation, a significant and quantifiable repulsive entropic contribution was obtained for the adsorption of the large and conformationally ‘soft’ proteins bovine serum albumin and fibrinogen (see Extended Data Calculations Fig. 2). As the slope of ΔG is positive for lysozyme and bovine serum albumin, the entropic repulsion dominates the energy balance. In fibrinogen adsorption, this entropic repulsion is still effective but no longer dominant. Accordingly, the adsorption of this by far largest and softest protein is associated with a higher gain of enthalpy by deformation and/or a gain of entropy due to conformational changes.”

We would like to recall that the above-discussed quantification of the entropic repulsion by first-principles thermodynamics is also in remarkable agreement with an independent estimation based on geometrical considerations, relating the cross-sectional area of protein-facing cholesterol at the SCL interface to the area of orientationally free cholesterol at the SCL-solution interface (Fig. 3b and Extended Data Calculations Part 3), that is experimentally supported by time-dependent interaction force data (lines 216–229 of the manuscript).

Beyond that, we would like to emphasize that the relevance of molecular orientational fluctuations for the investigated system proving the entropic repulsion mechanism was independently demonstrated by molecular dynamics (MD) simulations (lines 235–248 of the manuscript) and by AFM-based force mapping experiments (lines 249–258 of the manuscript).

References

3. Jia, P. *et al.* Probing the adjustments of macromolecules during their surface adsorption. *ACS Appl. Mater. Interfaces* **7**, 6422–6429 (2015).
4. Haynes, C. A. & Norde, W. Structures and stabilities of adsorbed proteins. *J. Colloid Interface Sci.* **169**, 313–328 (1995).
5. Norde, W. & Lyklema, J. Why proteins prefer interfaces. **2**, 183–202 (2012).
6. Roach, P., Farrar, D. & Perry, C. C. Interpretation of protein adsorption: Surface-induced conformational changes. *J. Am. Chem. Soc.* **127**, 8168–8173 (2005).
7. Norde, W. Adsorption of proteins at solid-liquid interfaces. *Cells Mater.* **5**, 97–112 (1995).
8. Rabe, M., Verdes, D. & Seeger, S. Understanding protein adsorption phenomena at solid surfaces. *Adv. Colloid Interface Sci.* **162**, 87–106 (2011).
9. Kurnik, M. *et al.* Quantitative measurements of protein–surface interaction thermodynamics. *Proc. Natl. Acad. Sci. U. S. A.* **115**, 8352–8357 (2018).
10. Ben-Tal, N., Honig, B., Bagdassarian, C. K. & Ben-Shaul, A. Association entropy in adsorption processes. *Biophys. J.* **79**, 1180–1187 (2000).
11. Fang, F. & Szleifer, I. Kinetics and thermodynamics of protein adsorption: A generalized molecular theoretical approach. *Biophys. J.* **80**, 2568–2589 (2001).

Reviewer Reports on the Fourth Revision:

Referees' comments:

Referee #3 (Remarks to the Author):

The revision and the response letter have sufficiently addressed my questions and concerns. Therefore, I recommend to accept the paper for publication.

Author Rebuttals to Fourth Revision:

Referee comments

Reviewer #3:

The revision and the response letter have sufficiently addressed my questions and concerns. Therefore, I recommend to accept the paper for publication.

We greatly appreciate the reviewer's effort in evaluating our revised manuscript and are very pleased with the positive comment.